# Carbonyl Sulfide: Comparing a Mechanistic Representation of the Vegetation Uptake in a Land Surface Model and the Leaf Relative Uptake Approach

Fabienne Maignan[1], Camille Abadie[1], Marine Remaud[1], Linda M. J. Kooijmans[2], Kukka-Maaria Kohonen[3], Róisín Commane[4], Richard Wehr[5], J. Elliott Campbell[6], Sauveur Belviso[1], Stephen A. Montzka[7], Nina Raoult[1], Ulli Seibt[8], Yoichi P. Shiga[9], Nicolas Vuichard[1], Mary E. Whelan[10], and Philippe Peylin[1]

[1]Laboratoire des Sciences du Climat et de l'Environnement, LSCE/IPSL, CEA-CNRS-UVSQ, Université Paris-Saclay, Gif-sur-Yvette, France
[2]Meteorology and Air Quality, Wageningen University and Research, Wageningen, The Netherlands
[3]Institute for Atmospheric and Earth System Research (INAR)/Physics, Faculty of Science, University of Helsinki, Helsinki, Finland
[4]Dept. Earth & Environmental Sciences, Lamont-Doherty Earth Observatory of Columbia University, New York, NY 10964, USA
[5]Department of Ecology and Evolutionary Biology, University of Arizona, Tucson, USA
[6]Sierra Nevada Research Institute, University of California, Merced, California 95343, USA
[7]NOAA Global Monitoring Laboratory, Boulder, Colorado, USA
[8]Dept of Atmospheric & Oceanic Sciences, University of California Los Angeles, California 90095, USA
[9]Universities Space Research Association, Mountain View, CA, USA
[10]Department of Environmental Sciences, Rutgers University, New Brunswick, NJ 08901, USA

*Correspondence to*: Fabienne Maignan (fabienne.maignan@lsce.ipsl.fr)

**Abstract**. Land surface modelers need measurable proxies to constrain the quantity of carbon dioxide ($CO_2$) assimilated by continental plants through photosynthesis, known as Gross Primary Production (GPP). Carbonyl sulfide (COS), which is taken up by leaves through their stomata and then hydrolysed by photosynthetic enzymes, is a candidate GPP proxy. A former study with the ORCHIDEE land surface model used a fixed ratio of COS uptake to $CO_2$ uptake normalized to respective ambient concentrations for each vegetation type (Leaf Relative Uptake, LRU), to compute vegetation COS fluxes from GPP. The LRU approach is known to have limited accuracy since the LRU ratio changes with variables such as Photosynthetically Active Radiation (PAR): while $CO_2$ uptake slows under low light, COS uptake is not light limited. However, the LRU approach has been popular for COS-GPP proxy studies because of its ease of application and apparent low contribution to uncertainty for regional scale applications. In this study we refined the COS-GPP relationship and implemented in ORCHIDEE a mechanistic model that describes COS uptake by continental vegetation. We compared the simulated COS fluxes against measured hourly COS fluxes at two sites, and studied the model behaviour and links with environmental drivers. We performed simulations at global scale, and estimated the global COS uptake by vegetation to be -756 Gg S yr$^{-1}$, in the middle range of former studies (-490 to -1335 Gg S yr$^{-1}$). Based on monthly mean fluxes simulated by the mechanistic approach in ORCHIDEE, we derived new LRU values for the different vegetation types, ranging between 0.92 and 1.72, close to recently published averages for observed values of 1.21 for C4 and 1.68 for C3 plants. We transported the COS using the monthly vegetation COS fluxes derived from both the mechanistic and the LRU approaches, and evaluated the simulated COS concentrations at NOAA sites. Although the mechanistic approach was more appropriate when comparing to high-temporal-resolution COS flux measurements, both approaches gave similar results when transporting with monthly COS fluxes and evaluating COS concentrations at stations. In our study, uncertainties between these two approaches are of second importance

as compared to the uncertainties in the COS global budget, which are currently a limiting factor to the potential of COS concentrations to constrain GPP simulated by land surface models on the global scale.

## 1 Introduction

Humanity has to face the urgency of climate change if it hopes to limit adverse future impacts (Allen et al., 2018; IPCC, 2019a, 2019b). In order to make reliable predictions of future climate, scientists have built powerful numerical Earth System Models (ESMs), where they continuously integrate gained knowledge on a multitude of climate-related and climate-interacting processes. The carbon cycle is at the heart of the present global warming, caused by anthropogenic $CO_2$ emissions (Ciais et al., 2013). In the global carbon budget, the land component

shows the largest uncertainty (Le Quéré et al., 2018; Bloom et al., 2016). Land Surface Models (LSMs) struggle to accurately represent the large spatial and temporal variability of the $CO_2$ gross and net fluxes (Anav et al., 2015). $CO_2$ is first assimilated through plant photosynthesis, before being respired by the ecosystem. The quantity of assimilated carbon is called Gross Primary Productivity (GPP). All other carbon fluxes and stocks derive from this first gross assimilation flux. To help reduce uncertainties in the estimated GPP, LSMs can benefit from knowledge

obtained through local eddy covariance measurements of the net ecosystem-atmosphere $CO_2$ exchange (Friend et al., 2007; Kuppel et al., 2014).

GPP proxies are also used, such as Solar-Induced Fluorescence (Norton et al., 2019; Bacour et al., 2019), isotopic composition of atmospheric $CO_2$ ($\delta^{18}O$: Farquhar et al., 1993; Welp et al., 2011; $\delta^{13}C$: Peters et al., 2018) and Carbonyl Sulfide (COS) atmospheric concentrations (Hilton et al., 2015). Using atmospheric COS measurements

as a tracer for terrestrial photosynthesis was first suggested by Sandoval-Soto et al. (2005) and Montzka et al. (2007), and Campbell et al. (2008) provided quantitative evidence using airborne observations of COS and $CO_2$ concentrations and an atmospheric transport model. COS is an atmospheric trace gas that has a molecular structure very similar to $CO_2$ and is likewise taken up by plants through stomates. COS is hydrolysed within the leaf, this reaction being catalysed by the enzyme Carbonic Anhydrase (CA). This reaction is light-independent (Protoschill-

Krebs et al., 1996; Goldan et al., 1998) and, because of the high catalytic efficiency of this enzyme (Ogawa et al., 2013; Ogée et al., 2016; Protoschill-Krebs et al., 1996), COS hydrolysis inside the leaf seems therefore to be limited by COS supply driven by changes in stomatal conductance (Goldan et al., 1988; Sandoval-Soto et al., 2005; Seibt et al., 2010; Stimler et al., 2010). Leaves' uptake of COS and $CO_2$ are thus very similar, but leaves do not produce COS (Protoschill-Krebs et al., 1996; Notni et al., 2007), whereas they emit $CO_2$ through respiration.

That is why vegetation COS fluxes could be used as a proxy for GPP. It is however to be noted that Gimeno et al. (2017) reported COS emissions by bryophytes during daytime.

The approach generally adopted to constrain GPP with COS relies on the determination of a Leaf Relative Uptake (LRU), which is the ratio of COS to $CO_2$ uptake normalized by their atmospheric concentrations (Sandoval-Soto et al., 2005):

$$LRU = \frac{F_{COS}}{GPP} \frac{[CO_2]_a}{[COS]_a} \qquad (1)$$

where $F_{COS}$ is the flux of COS uptake (pmol COS m$^{-2}$ s$^{-1}$), $GPP$ is the gross flux of $CO_2$ assimilation (µmol $CO_2$ m$^{-2}$ s$^{-1}$), $[COS]_a$ is the atmospheric COS mixing ratio (pmol COS mol$^{-1}$, ppt), and $[CO_2]_a$ is the atmospheric $CO_2$ mixing ratio (µmol $CO_2$ mol$^{-1}$, ppm).

LRU can be estimated experimentally, and then used as a scaling factor for estimating GPP, if $F_{COS}$, $[COS]_a$ and $[CO_2]_a$ are available. Measurements can be made at leaf level using branch chambers (Seibt et al., 2010; Kooijmans et al., 2019); LRU can also be estimated at ecosystem level: eddy-covariance flux towers measure the ecosystem total COS flux (Kohonen et al., 2020), removing the soil contribution gives access to the vegetation part (Wehr et al., 2017). Soil can absorb and emit COS (Whelan et al., 2016; Kitz et al., 2020), the magnitude of their flux being generally much lower than that of vegetation fluxes (Berkelhammer et al., 2014; Maseyk et al., 2014; Wehr et al., 2017; Whelan et al., 2018). Epiphytes (lichen, mosses) could also have a significant contribution to the ecosystem COS budget (Kuhn and Kesselmeier, 2000; Rastogi et al., 2018).

However, LRU does not appear constant under some environmental conditions. For example, the fixation of carbon from $CO_2$ relies on light-dependent reactions, unlike the uptake of COS by the CA enzyme, which is light-independent (Stimler et al., 2011). Because of these different responses of COS and $CO_2$ uptake in leaves, LRU varies with light conditions, and decreases sharply with PAR increase (Stimler et al., 2010 ; Maseyk et al., 2014; Commane et al., 2015; Wehr et al., 2017; Yang et al., 2018). Consequently, LRU values are smaller at midday or in seasons with high incoming light (Kooijmans et al., 2019). Moreover, COS assimilation continues at night as stomatal conductance to gas transfer does not drop to zero, whereas $CO_2$ uptake by plants stops, leading to an infinite value of LRU. Note however that stomates mostly close at night, so the COS uptake at night is smaller than the COS uptake during the day. The diel (i.e. 24-hourly) variation of LRU with light may however be only of second order importance as GPP is very low at low light, and Yang et al. (2018) found that considering sub-daily variations of LRU when computing daily mean GPP values had no importance. It has also been shown that LRU varies between plant species (Stimler et al., 2011), which is why different LRU values were estimated for different vegetation types (Seibt et al., 2010; Whelan et al., 2018). The variability of LRU with plant type and over a day and season (inferred by changes in light-conditions) should therefore be carefully accounted for when COS concentrations or flux measurements are used to estimate GPP at the ecosystem and larger scales. We also have to acknowledge that there are still factors that are not accounted for if discrepancies between GPP and COS-based estimations are larger than their estimated respective uncertainties.

Before being able to use COS observations to constrain the simulated GPP, Land Surface Models (LSMs) first need to have an accurate model to simulate vegetation COS fluxes. In a former study, Launois et al. (2015b) simply defined the COS uptake by vegetation as the $CO_2$ gross uptake simulated by LSMs, scaled with a constant LRU value for each large vegetation class. The goal of this study is to now simulate the uptake of atmospheric COS by continental vegetation in a more complex and realistic way using a mechanistic approach within an LSM, and apply this model to evidence the shortcomings or pertinence of the LRU concept, depending on the studied scales. To this end:

 i) We used the state-of-the art ORCHIDEE LSM (Krinner et al., 2015), and implemented in it the vegetation COS uptake model of Berry et al. (2013) to simulate the COS fluxes absorbed at the leaf and canopy levels by the continental vegetation.

 ii) We evaluated the simulated COS fluxes against measurements at two forest sites, namely the Harvard Forest, United States (Wehr et al., 2017), and Hyytiälä, Finland (Kooijmans et al., 2019; Kohonen et al., 2020; Sun et al., 2018a). We studied the high-frequency behaviour of the modelled conductances over the season and the dependency of the LRU on the environmental and structural conditions.

iii) We compared the simulated mechanistic COS fluxes at global scale to former estimates; we studied LRU values estimated from monthly fluxes, that are pertinent for atmospheric studies, and compared them to monthly means of high-frequency LRU values.

iv) The mechanistic and LRU simulated COS fluxes were used with the atmospheric transport model LMDz (Hourdin et al. 2006), to provide atmospheric COS concentrations that were evaluated against measurements at sites of the NOAA network.

## 2 Models, Data, and Methodology

### 2.1 Implementation of plant COS uptake in the ORCHIDEE LSM to simulate COS vegetation fluxes

### 2.1.1 The ORCHIDEE LSM

ORCHIDEE is an LSM developed mainly at Institut Pierre Simon Laplace (IPSL), that computes the water, carbon and energy balances at the interface between land surfaces and atmosphere (Krinner et al., 2005). Fast processes including hydrology, photosynthesis and energy balance are run at a half-hourly timestep, while other slower processes such as carbon allocation and mortality are simulated at a daily timestep. The sub-grid variability for vegetation is represented using fractions of Plant Functional Types (PFTs), grouping plants with similar morphologies and behaviours growing under similar climatic conditions. Photosynthesis follows the Yin and Struik (2009) approach, bringing improvements to the standard Farquhar et al. (1980) model for C3 plants, the Collatz et al. (1992) model for C4 plants, and the Ball et al. (1987) model for the stomatal conductance. A main novelty is the introduction of a mesophyll conductance linking the $CO_2$ concentration at the carboxylation sites, $C_c$, to the $CO_2$ intracellular concentration, $C_i$. For each PFT, the reference value for the maximum photosynthetic capacity at 25°C, $V_{max,25}$, is derived from literature survey, observation databases, possibly later calibrated using FLUXNET observations (e.g. Kuppel et al., 2012). To compute the maximum photosynthetic capacity at leaf level, $V_{max}$, the reference value is multiplied at a daily time step by the relative photosynthetic efficiency of leaves based on the mean leaf age following Ishida et al. (1999) (see equation A12 and Figure A12 in Krinner et al., 2005). Leaves are very efficient when they are young and stay so till they approach their pre-defined leaf lifespan. The temperature-dependence of the maximum photosynthetic capacity follows Medlyn et al. (2002) and Kattge and Knorr (2007). A water stress function varying between 0 and 1 depending on soil moisture and root profile (de Rosnay and Polcher, 1998) is applied on maximum photosynthetic capacity and conductances. The canopy is discretized in several layers of growing thickness, the number depending on the actual Leaf Area Index (LAI). All the incoming light is considered to be diffuse, and no distinction is made between sun and shaded leaves. The light is attenuated through the canopy following a simple Beer-Lambert absorption law. The $CO_2$ assimilation, the stomatal conductance and the intercellular $CO_2$ concentration $C_i$ are computed per LAI layer, provided LAI is higher than 0.01 and the monthly mean air temperature is higher than -4°C. The $CO_2$ assimilation and the stomatal conductance are further summed-up over all layers to compute GPP and the total conductance at canopy level. The scaling to the grid cell is made using means weighted by the Plant Functional Types fractions. Phenology is fully prognostic with PFT-specific phenological models as described in Botta et al. (2000) and MacBean et al. (2015). ORCHIDEE can be run from the site scale to the global scale, coupled with an atmospheric general circulation model, or in off-line mode forced by meteorological fields. In this study, we prescribed the vegetation distribution

for site simulations and used yearly PFT maps derived from the ESA Climate Change Initiative (CCI) land cover products for global simulations (Poulter et al., 2015). The soil type is derived from the Zobler map (Zobler, 1986). To account for the $CO_2$ fertilization effect, we considered global means of $[CO_2]_a$ with yearly varying values, as provided by the TRENDY model inter-comparison project (Sitch et al., 2015). The impact of not taking into account the spatial and temporal variations of $[CO_2]_a$ on GPP has been studied in Lee et al. (2020); while this simplification has indeed no impact at global yearly scale for GPP, this may be less true at site and seasonal scales. We used the recent ORCHIDEE version fine-tuned for the Climate Model Intercomparison Project (CMIP) 6 exercise (Peylin et al., *in prep*.), forced by micro-meteorology fields at FLUXNET sites or by 2-degree CRUNCEP reanalyses at global scale (https://rda.ucar.edu/datasets/ds314.3/).

**2.1.2 The Berry model for plant COS uptake**

We implemented in the ORCHIDEE LSM the mechanistic model of plant COS uptake based on Berry et al. (2013). In this model, COS follows a diffusive law from the atmosphere to the leaf interior, where it is consumed by CA in the chloroplasts. The uptake from the atmosphere is assumed unidirectional, reflecting the fact that COS is generally not produced by plants. The model distinguishes three conductances along the COS path between the atmosphere and the leaf interior: (1) the boundary layer conductance ($g_{B\_COS}$) to gas transfer between the leaf surface and the atmosphere, (2) the stomatal conductance ($g_{S\_COS}$), and (3) the internal conductance ($g_{I\_COS}$). Internal conductance combines the mesophyll conductance and the CA activity into a single equivalent conductance.

The stomatal and boundary layer conductances are associated with factors describing diffusion of COS relative to that of water vapour (1.94 and 1.56, respectively, Stimler et al., 2010). In the chloroplast, the COS hydrolysis is catalysed by the enzyme CA, following first order kinetics. COS uptake depends on the amount of CA and its relative location to intercellular air spaces, which brings in the mesophyll conductance. These two factors have been shown to scale with the maximum reaction rate of the Rubisco enzyme, $V_{max}$ (µmol m$^{-2}$ s$^{-1}$) (Badger and Price, 1994; Evans et al., 1994). The mesophyll conductance and the first-rate constant are then regrouped into a single equivalent internal conductance, proportional to $V_{max}$:

$$g_{I\_COS} = \alpha * V_{max} \qquad (2)$$

The parameter $\alpha$ takes two values depending on the plant photosynthetic pathway (C3 or C4). These values were determined experimentally by Berry et al. (2013), who estimated an $\alpha = 0.0012$ for C3 and an $\alpha = 0.013$ for C4 species. We thus have the final equation:

$$F_{COS} = [COS]_a * g_{T\_COS} = [COS]_a * \left[ \frac{1.0}{g_{B\_cos}} + \frac{1.0}{g_{S\_cos}} + \frac{1.0}{g_{I\_cos}} \right]^{-1}$$

$$= [COS]_a * \left[ \frac{1.56}{g_{B\_W}} + \frac{1.94}{g_{S\_W}} + \frac{1.0}{g_{I\_cos}} \right]^{-1} \qquad (3)$$

where $F_{COS}$ is the flux of COS uptake (pmol COS m$^{-2}$ s$^{-1}$), $[COS]_a$ is the background atmospheric COS mixing ratio considered here as a constant (500 ppt), $g_{T\_COS}$, $g_{B\_COS}$, $g_{S\_COS}$, and $g_{I\_COS}$ are respectively the total, boundary layer, stomatal, and internal conductances to COS (mol COS m$^{-2}$ s$^{-1}$), and $g_{B\_W}$ and $g_{S\_W}$ are respectively the boundary layer and stomatal conductances to water vapour (mol H$_2$O m$^{-2}$ s$^{-1}$). Note that in this work $[COS]_a$ is held constant when computing the COS fluxes, contrary to Berry et al. (2013) and Campbell et al. (2017), where $[COS]_a$ is dynamic and taken from the previous time step's PCTM (Parameterized Chemical

Transport Model) value. The uncertainty introduced by this simplification is evaluated in the Discussion section. The vegetation COS flux and related conductances are computed for each LAI layer, and then summed-up to get total values at canopy level. Unless specified otherwise, fluxes, conductances and LRU are further presented and discussed at canopy level.

### 2.1.3 Minimal conductances

As plant $CO_2$ uptake only occurs under certain conditions such as with sufficient light, temperature, and water, $CO_2$ assimilation is not calculated in ORCHIDEE when these conditions are not fulfilled. Therefore, the stomatal conductance to $CO_2$ that is needed to obtain the stomatal conductance to COS is not always computed in ORCHIDEE. However, some studies have shown incomplete stomatal closure at night (Dawson et al., 2007; Lombardozzi et al., 2017; Kooijmans et al., 2019), leading to nighttime COS plant uptake (Berry et al., 2013; Kooijmans et al., 2017). Therefore, we had to define a minimal stomatal conductance to COS under these particular conditions when there is no $CO_2$ assimilation. The minimal conductance to $CO_2$ used in ORCHIDEE is based on the residual stomatal conductance if the irradiance approaches zero, represented as the $g_0$ offset in the stomatal conductance models (see equations (15) for C3 and (25) for C4 plants in Yin and and Struik, 2009). In the absence of water stress, $g_0$ takes a constant value for C3 (0.00625 mol $CO_2$ $m^{-2}$ $s^{-1}$) and C4 (0.01875 mol $CO_2$ $m^{-2}$ $s^{-1}$) plants. This constant is multiplied by a water-stress function to compute the minimal conductance. This minimal conductance to $CO_2$ was then applied under conditions when there is no $CO_2$ assimilation, multiplied by the ratio to convert the conductance to $CO_2$ into a conductance to COS. We thus model COS assimilation even at night, for all PFTs, and in winter for evergreen species, depending on water stress conditions.

### 2.1.4 Simulations protocol

All simulations were preceded by a "spin-up" phase to get to an equilibrium state where the considered carbon pools and fluxes are stable with no residual trends in the absence of any disturbances (climate, land use change, $CO_2$ atmospheric concentrations) (e.g. Wei et al., 2014). A few decades are enough to equilibrate above-ground biomass and GPP. As we transport not only COS, but also $CO_2$ (see Sect. 2.4 below), we need a longer spin-up where all carbon pools including those in the soil are stable and the net $CO_2$ fluxes oscillate around zero. Equilibrating the ecosystem photosynthesis with its respiration takes a long time as the slowest soil carbon pool has a residence time on the order of one thousand years. The ORCHIDEE model has a built-in spin-up procedure to accelerate the convergence towards this equilibrium state, using a pseudo-analytical iterative estimation of the targeted carbon pools, based on Lardy et al. (2011). For global simulations, we first performed a 340-year spin-up phase with non-varying pre-industrial atmospheric $CO_2$ concentration and vegetation map, cycling over the same 10 years of meteorological forcing files, where the final relative variation of the global slowest soil carbon pool was less than 5%. Starting from this equilibrium state, a transient state simulation was then run applying climate change, land use change and increasing $CO_2$ atmospheric concentrations, and COS and GPP fluxes were calculated from 1860 to 2017. We performed site simulations at the Harvard Forest (United States) and Hyytiälä (Finland) FLUXNET sites (see below). For the two sites, we first performed a spin-up simulation cycling over the available years of the FLUXNET forcing files, for around 340 years, using a constant atmospheric $CO_2$ concentration corresponding to the first year of the FLUXNET forcing file. We then performed the transient simulations over the available FLUXNET years, for each site, with a varying $CO_2$ atmospheric concentration.

**2.2 Evaluation of vegetation COS fluxes at two FLUXNET sites**

Vegetation COS fluxes can be measured using branch chambers or estimated using the difference between measurements of ecosystem and soil fluxes. Such measurements were available at the Hyytiälä (Finland) and Harvard Forest (United States) FLUXNET sites. The Hyytiälä site (61.85°N, 24.29°E) is a boreal evergreen needleleaf forest dominated by Scots pine (*Pinus sylvestris*). Branch measurements of COS fluxes were made in a Scots pine tree from March to July 2017 using gas-exchange chambers (Kooijmans et al., 2019); fluxes were derived from mole fraction changes when the chambers were closed once every hour. Measurements were made with an Aerodyne Quantum Cascade Laser Spectrometer (QCLS) and were calibrated against reference standards (Kooijmans et al., 2016). Fluxes from empty chambers were regularly measured to be able to correct for gas exchange by the chamber and tubing material (Kooijmans et al., 2019). We also used the Hyytiälä COS ecosystem fluxes (Kohonen et al., 2020); eddy covariance fluxes were measured during years 2013-2017 at 23 m height, approximately 6 m above the canopy height. Flux data were processed, quality screened and gap-filled according to recommendations by Kohonen et al. (2020). Soil fluxes were also available for year 2015 (Sun et al., 2018a), we thus derived the COS vegetation fluxes at canopy scale for that year from the difference between ecosystem and soil fluxes. Soil fluxes were generally low compared to plant uptake.

The Harvard Forest site (42.54°N, 72.17°W) is a temperate deciduous broadleaf forest with mainly red oak (*Quercus rubra*), red maple (*Acer rubrum*) and hemlock (*Tsuga canadensis*). Ecosystem COS eddy flux measurements were carried out from a tower from May to October, in 2012 and 2013, using an Aerodyne QCLS and calibrated using gas cylinders. They were further split into vegetation and soil components, using soil chamber $CO_2$ measurements and a sub-canopy flux-gradient approach (Wehr et al., 2017).

The simulated COS fluxes were evaluated against measurements using the Root Mean Square Deviation:

$$RMSD = \sqrt{\frac{\sum_{n=1}^{N}\left(F_{COS}^{Obs}(n) - F_{COS}^{Mod}(n)\right)^2}{N}} \tag{4}$$

where $N$ is the number of considered observations, $F_{COS}^{Obs}(n)$ is the nth observed COS flux and $F_{COS}^{Mod}(n)$ is the nth modelled COS flux, and the relative RMSD:

$$rRMSD = \frac{RMSD}{\overline{F_{COS}^{Obs}}} \tag{5}$$

which is the RMSD divided by the mean value of observations.

We also computed the bias, standard deviations and correlation coefficient:

$$bias = \overline{F_{COS}^{Mod}} - \overline{F_{COS}^{Obs}} \tag{6}$$

$$SD^{Mod} = \sqrt{\frac{\sum_{n=1}^{N}\left(F_{COS}^{Mod}(n) - \overline{F_{COS}^{Mod}}\right)^2}{N}}$$

$$SD^{Obs} = \sqrt{\frac{\sum_{n=1}^{N}\left(F_{COS}^{Obs}(n) - \overline{F_{COS}^{Obs}}\right)^2}{N}} \tag{7}$$

$$r = \frac{\sum_{n=1}^{N}\left(F_{COS}^{Obs}(n) - \overline{F_{COS}^{Obs}}\right) \cdot \left(F_{COS}^{Mod}(n) - \overline{F_{COS}^{Mod}}\right)}{N \cdot SD^{Obs} \cdot SD^{Mod}} \tag{8}$$

We used partial correlations to identify the main drivers of the modelled conductances. Given the high non-linearity of the equations linking the conductances to their predictors, we also used Random Forests (RF) to simulate ORCHIDEE results, and applied a permutation technique on these RF models to rank predictors

(Breiman, 2001). RF are well adapted for non-linear problems, they were for example used to rank variables of importance for soil COS fluxes in Spielman et al. (2020).

**2.3 Global scale flux estimates and Comparisons with the LRU approach**

We compared our estimate for plant COS uptake at global scale to former studies, with a focus on the LRU approach. We also applied the LRU approach to derive new estimates of global plant COS uptake for comparison,

using a monthly climatology of our modelled GPP fluxes over the 2000-2009 period, a constant atmospheric concentration of 500 ppt for COS and global yearly values for $CO_2$ (from 368 ppm for year 2000 to 386 ppm for year 2009). We considered two sets of constant PFT-dependent LRU values. The first set (LRU_Seibt) was taken from Seibt et al. (2010), based on the observed LRU values displayed in their Table 3 (intermediate column). The second set (LRU_Whelan) used constant values for C3 (1.68) and C4 (1.21) plants where the values are an average

over different field and laboratory measurements as assembled by Whelan et al. (2018). Both sets are listed in Table 1.

Reciprocally, we derived LRU values using equation (1) applied to the monthly climatology of our modelled COS and GPP fluxes over the 2000-2009 period, these will be further called LRU_MonthlyFluxes values. LRU_MonthlyFluxes values were computed for all strictly positive GPP values. For each PFT, we studied the

275 spatio-temporal distribution of LRU_MonthlyFluxes values among grid cells where the PFT was present. We also compared these LRU_MonthlyFluxes values computed from a climatology of monthly fluxes, to the climatology of monthly mean LRU values, directly computed from the original half-hourly LRU values, and further called Monthly_LRU. Given the non-linearity of the problem, we expect LRU_MonthlyFluxes to be different from Monthly_LRU values. Considering that the objective of the LRU approach was to estimate COS fluxes from GPP

using a constant value per PFT, the optimal LRU value for each PFT was obtained by linearly regressing monthly COS fluxes against monthly GPP fluxes multiplied by the ratio of the mean COS to $CO_2$ concentrations, with no offset, thus:

$$LRU\_Opt = \frac{\sum_{n=1}^{N} F_{COS}^{Mod}(n) GPP^{Mod}(n) \frac{[COS(n)]_a}{[CO_2(n)]_a}}{\sum_{n=1}^{N} \left( F_{COS}^{Mod}(n) \right)^2} \tag{9}$$

with N the number of grid cell-month simulated fluxes where the PFT is present in the monthly climatology.

We compared this new set of optimal PFT-dependent LRU values against LRU_Seibt and LRU_Whelan.

We finally used the $LRU\_Opt$ values to re-compute the monthly mean COS fluxes from our modelled monthly mean GPP, and compared with the mechanistic COS flux calculation. The differences, due to the non-linearity of the COS flux calculation, provide some information on the use of a simplified approach based on mean LRU values.

**Table 1: Table of LRU per PFT. First column: median and optimal LRU values calculated from the simulated mechanistic COS and GPP fluxes. Middle columns: calculated from Seibt et al. (2010) for the ORCHIDEE PFT classification. Last column: from Whelan et al. (2018)**

| PFT | | ORCHIDEE | | Seibt | Whelan |
|---|---|---|---|---|---|
| Long name | Abbreviation | Median | Optimal | | |
| 1 – Bare soil | Bare | 0.00 | 0.00 | 0.00 | 0.00 |
| 2 – Tropical Broad-leaved Evergreen Forest | TroBroEver | 1.56 | 1.72 | 3.09 | 1.68 |

| | | | | | |
|---|---|---|---|---|---|
| 3 – Tropical Broad-leaved Raingreen Forest | TroBroRain | 1.48 | 1.62 | 3.38 | 1.68 |
| 4 – Temperate Needleleaf Evergreen Forest | TempNeedleEver | 1.17 | 1.39 | 1.89 | 1.68 |
| 5 – Temperate Broad-leaved Evergreen Forest | TempBroEver | 0.86 | 1.06 | 3.60 | 1.68 |
| 6 – Temperate Broad-leaved Summergreen Forest | TempBroSum | 1.06 | 1.31 | 3.60 | 1.68 |
| 7 – Boreal Needleleaf Evergreen Forest | BorNeedleEver | 0.82 | 0.95 | 1.89 | 1.68 |
| 8 – Boreal Broad-leaved Summergreen Forest | BorBroSum | 0.84 | 1.03 | 1.94 | 1.68 |
| 9 – Boreal Needleleaf Summergreen Forest | BorNeedleSum | 0.76 | 0.92 | 1.89 | 1.68 |
| 10 – Temperate C3 Grass | TempC3grass | 1.01 | 1.18 | 2.53 | 1.68 |
| 11 – C4 Grass | C4grass | 1.38 | 1.45 | 2.00 | 1.21 |
| 12 – C3 Agriculture | C3crops | 1.21 | 1.37 | 2.26 | 1.68 |
| 13 – C4 Agriculture | C4crops | 1.75 | 1.72 | 2.00 | 1.21 |
| 14- Tropical C3 grass | TropC3grass | 1.40 | 1.52 | 2.39 | 1.68 |
| 15- Boreal C3 grass | BorC3grass | 0.87 | 0.97 | 2.02 | 1.68 |

### 2.4 Simulations of COS concentrations and Evaluation at NOAA air sampling sites

The vegetation COS fluxes, as well as all other sources and sinks of the global COS budget, based on their latest estimates, are transported with an atmospheric transport model, so that we are able to simulate 3D COS atmospheric concentrations and compare them to the NOAA surface measurements.

### 2.4.1 The atmospheric transport model LMDz

In order to simulate COS and $CO_2$ concentrations in the atmosphere, we used the version of the atmospheric component LMDz of the Institut Pierre Simon Laplace Coupled Model (IPSL-CM) (Dufresne et al., 2013) which has been contributing to the CMIP6 exercise. To reduce the computation time, we used its off-line mode: precomputed air mass fluxes provided by the full version of LMDz are used to transport the different tracers (Hourdin et al., 2006). This version is further called LMDz6 and is described in Remaud et al. (2018) and references therein for the transport of $CO_2$. The horizontal winds are nudged towards ECMWF meteorological analyses (ERA-5, https://www.ecmwf.int/en/forecasts/datasets/archive-datasets/reanalysis-datasets/era5) to realistically account for large scale advection. The tropospheric OH oxidation of COS is calculated from OH monthly data that are produced from a first simulation done with the INCA tropospheric photochemistry scheme (Folberth et al., 2006; Hauglustaine et al., 2004, 2014). The photolysis reaction of COS in the stratosphere is not considered: the lifetime of COS in the stratosphere is 64 years (Barkley et al., 2008). The model is set up at a horizontal resolution of 3.8° x 1.9° (96 grid cells in longitude and latitude) with 39 hybrid sigma-pressure levels reaching an altitude up to about 75 km, corresponding to a vertical resolution of about 200-300 m in the planetary boundary layer. The model timestep is 30 minutes and the output concentrations are 3-hourly averaged.

### 2.4.2 Atmospheric simulations: sampling methods and data processing

We ran the LMDz6 version of the atmospheric transport model described above for the years 2000 to 2009. The prescribed COS and $CO_2$ fluxes used as model inputs are presented in Table 2 and Table 3. The GPP estimated by

ORCHIDEE (148.1 Gt C yr$^{-1}$) is on the high range among the model estimates (Anav et al., 2015), with a corresponding high respiration (145.7 Gt C yr$^{-1}$) to ensure a realistic net ecosystem exchange (Friedlingstein et al., 2019). However, other high GPP estimates can be found in the literature such as Welp et al. (2011) that suggest a range of 150 to 175 based on $\delta^{18}O$ data. Likewise, Joiner et al. (2018) have proposed a new GPP product, based on satellite data and calibrated on FLUXNET sites, with an estimate around 140 Gt C yr$^{-1}$ for 2007.

The fluxes are given as a lower boundary condition of the atmospheric transport model (LMDz), which then simulates the transport of COS and $CO_2$ by the atmospheric flow. The atmospheric COS seasonal variations are likely to be dominated by the seasonal exchange with the terrestrial vegetation, while the mean mole fractions result from all sources and sinks of COS, some of which are still largely unknown (e.g. ocean fluxes, Whelan et al., 2018). In this study, we only focus on the seasonal cycle and do not attempt to simulate the annual mean value,

we thus started from a null initial state. The atmospheric transport is almost linear with respect to the fluxes: the linearity is a property of the atmospheric transport, though it is violated in LMDz because of the presence of slope limiters in the advection scheme. Overall, since all the other LMDz components are linear, LMDz transport is generally considered linear with fluxes (Hourdin and Talagrand, 2006). Relying on this relationship, we first transported each flux separately, and then added all the simulated concentrations in the end, for each species.

For all COS and $CO_2$ observations, the model output was sampled at the nearest grid point and vertical level to each station, and was extracted at the exact hour when each flask sample had been taken. For each station, the curve-fitting procedure developed by the NOAA Climate Monitoring and Diagnostic Laboratory (NOAA/CMDL) (Thoning, 1989) was applied to modelled and observed COS and $CO_2$ time series to extract a smooth detrended seasonal cycle. We first fitted a function including a second-order polynomial term and 4 harmonic terms, and

then applied to the residuals a low pass filter with either 80 or 667 days as short-term and long-term cut-off values, respectively. The detrended seasonal cycle is defined as the smooth curve (full function plus short-term residuals) minus the trend curve (polynomial plus long-term residuals).

**Table 2: Prescribed COS surface fluxes used as model input. Mean magnitudes of different types of fluxes are given for the period 2000-2009**

\*A bug has been discovered in the parameterization of direct COS emissions in the NEMO PISCES ocean model: the hydrolysis rate was three times too low, resulting in an artificial build-up of COS in seawaters. As a correction, we divided by three the total amount of oceanic COS fluxes within a year, assuming that the bug does not affect the spatial pattern of direct emissions of COS.

| Type of COS flux | Temporal resolution | Total (Gg S yr$^{-1}$) | Data Source |
|---|---|---|---|
| Anthropogenic | Monthly, interannual | 337.3 | Zumkehr et al. (2018) |
| Biomass burning | Monthly, interannual | 56.3 | Stinecipher et al. (2019) |
| Soil | Monthly, climatological | -409.0 | Launois et al. (2015b) |
| Ocean | Monthly, climatological | 444.7 | Kettle (2002) for indirect oceanic emissions (via CS2 and DMS oxydation), and Launois et al. (2015a) for direct oceanic emissions. The direct emissions are rescaled to be equal to 200 Gg S yr$^{-1}$ |

| | | | (*). |
|---|---|---|---|
| Vegetation uptake | Monthly, interannual | See Table 1. | This work, including mechanistic and LRU approaches (Seibt et al., 2010; Whelan et al., 2018). |


**Table 3: Prescribed $CO_2$ surface fluxes used as model input. Mean magnitudes of different types of fluxes are given for the period 2000-2009**

| Type of $CO_2$ flux | Temporal resolution | Total (Gt C $yr^{-1}$) | Data Source |
|---|---|---|---|
| Fossil fuel | Monthly, interannual | 7.7 | ECJRC/PBL EDGAR version 4.2 |
| Biomass burning | Monthly, interannual | 1.9 | GFED 4.1s |
| Respiration (including the land use emissions and wood harvest) | Monthly, interannual | 145.7 | ORCHIDEE |
| Ocean | Monthly, climatological | -1.3 | Landschützer et al. (2015) |
| GPP | Monthly, interannual | -148.1 | ORCHIDEE |

**2.4.3 COS and $CO_2$ concentrations at the NOAA/Global Monitoring Laboratory (GML) surface sites**

We used the NOAA/GML measurements of both $CO_2$ and COS at 10 sites located on both hemispheres, listed in
Table 4.

**Table 4: List of air sampling sites selected for evaluation of COS and $CO_2$ concentrations**

| Site | Short name | Coordinates | Elevation (m above sea level) | Comment |
|---|---|---|---|---|
| South Pole, Antarctica, United States | SPO | 90.0°S, 24.8° W | 2810 | |
| Cape Grim, Australia | CGO | 40.4°S, 144.6°W | 164 | inlet is 70 m aboveground |
| Tutuila, American Samoa | SMO | 14.2°S, 170.6°W | 77 | |
| Cape Kumukahi, United States | KUM | 19.5°N, 154.8°W | 3 | |
| Mauna Loa, United States | MLO | 19.5°N, 155.6°W | 3397 | |
| Niwot Ridge, United States | NWR | 40.0°N, 105.54°W | 3475 | |
| Wisconsin, United States | LEF | 45.9°N, 90.3°W | 868 | inlet is 396 m aboveground on a tall tower |
| Mace Head, Ireland | MHD | 53.3°N, 9.9°W | 18 | |
| Barrow, United States | BRW | 71.3°N, 155.6°W | 8 | |
| Alert, Canada | ALT | 82.5°N, 62.3°W | 195 | |

The samples are collected as pair flasks one to five times a month since 2000 and are then analysed in the
NOAA/GML's Boulder laboratories with gas chromatography and mass spectrometry detection. The

measurements are retained only if the difference between the pair flasks is less than 6.3 ppt for COS. These measurements can be downloaded from the ftp sites ftp://ftp.cmdl.noaa.gov/hats/carbonsulfide/ and, for $CO_2$, at ftp://ftp.cmdl.noaa.gov/ccg/co2.

### 2.4.4 Evaluation metrics

To evaluate and compare the performances of the mechanistic and LRU approaches at different NOAA surface sites, we used the normalised standard deviation (NSD) and the Pearson correlation coefficient (R). NSD is calculated as the ratio between the standard deviation of the simulated concentrations and the observed concentrations at the NOAA surface sites. NSD and R values closer to 1 indicate a better accuracy of the model.

# 3 Results

## 3.1 Site scale COS fluxes, conductances and LRU

### 3.1.1 COS fluxes

#### 3.1.1.1 Daily cycle

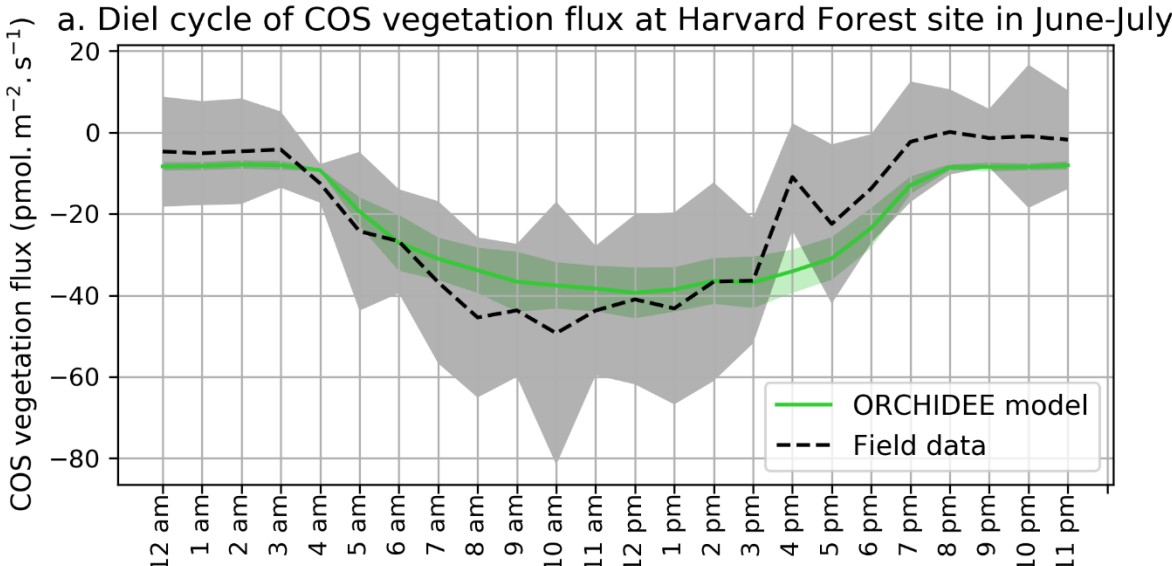

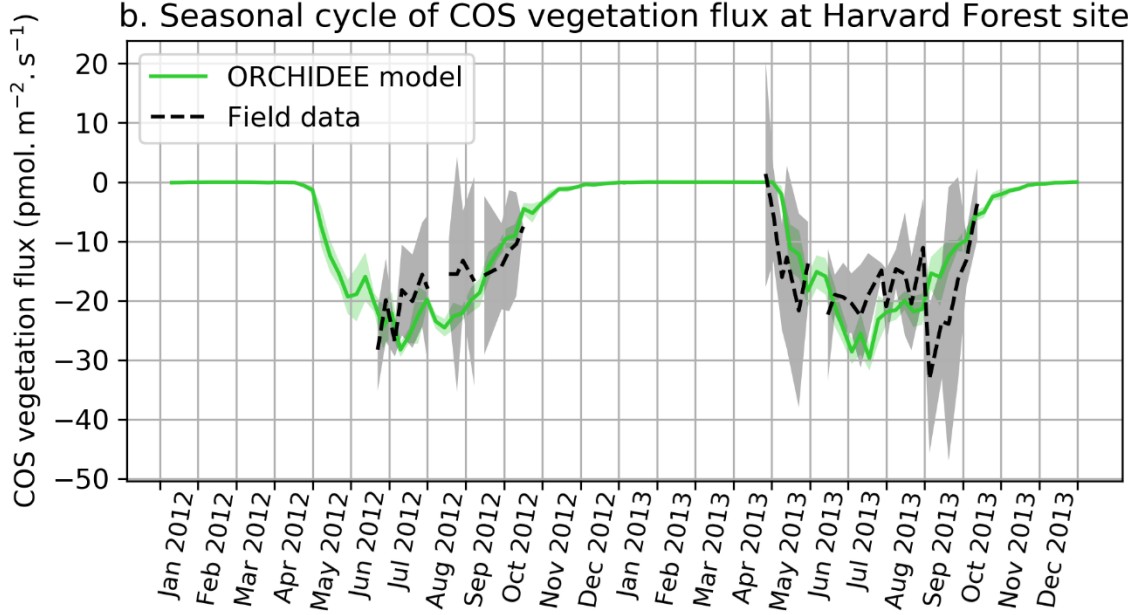

**Figure 1: a. Mean diel cycle of observed vegetation COS flux (Wehr et al., 2017) and modelled COS vegetation flux in June and July 2012 and 2013, at Harvard Forest, using an atmospheric convention where an uptake of COS by the ecosystem is negative. The shaded areas above and below each curve represent one standard-deviation of the considered hourly values over the June-July period. b. Mean seasonal cycle of simulated and observed weekly average vegetation COS flux in 2012 and 2013, at Harvard Forest. The shaded areas above and below each curve represent one standard-deviation of the daily means within the considered week. We imposed to have at least observations on two different days to compute the corresponding weekly mean.**

COS assimilation is minimum at night (between 8 PM and 4 AM) for observed and simulated fluxes (Figure 1a).

During night, uptake of modelled COS flux is around -8 pmol m$^{-2}$ s$^{-1}$ while field observations vary between -5

and 0 pmol m$^{-2}$ s$^{-1}$. In the morning, both simulated and observed uptakes increase. However, while the simulation shows a maximum assimilation of -38 pmol m$^{-2}$ s$^{-1}$ at noon, the maximum assimilation for observations is reached at 10 AM with a flux of -49 pmol m$^{-2}$ s$^{-1}$. Observed fluxes have thus a greater daily amplitude than simulated fluxes, and are a little ahead of the simulation, but this shift does not seem significant given the large variability of observations, as represented by the one standard-deviation in Figure 1a. RMSD for this mean diel cycle is 8.0 pmol m$^{-2}$ s$^{-1}$, and relative RMSD is 35%. The bias is -1.7 pmol m$^{-2}$ s$^{-1}$, the standard deviations are 17.5 pmol m$^{-2}$ s$^{-1}$ for the observations and 12.8 pmol m$^{-2}$ s$^{-1}$ for the simulated fluxes, and the correlation coefficient is 0.91. A similar study at the Hyytiälä site over July-September in year 2015 (Figure B1a) yields a similar underestimation of the amplitude of the mean diel cycle, with an RMSD of 4.0 pmol m$^{-2}$·s$^{-1}$ and a relative RMSD of 36%; the bias is 2.4 pmol m$^{-2}$·s$^{-1}$, the standard-deviations are 5.5 pmol m$^{-2}$·s$^{-1}$ for the observations and 2.7 pmol m$^{-2}$·s$^{-1}$ for the simulated fluxes, and the correlation coefficient is 0.93.

### 3.1.1.2 Seasonal cycle

The simulated weekly seasonal vegetation COS uptake roughly follows the same trend as the observed one ($r$=0.53, Figure 1b). COS uptake increases in spring when the vegetation growing season starts and decreases in autumn at the end of the forest activity period. Simulated and observed fluxes also take similar values over the two years. There are however differences: in 2013 the start of the season is simulated about two weeks too late in May instead of late April, and measured fluxes peak in May-June and August-September, while the modelled fluxes peak in July. We notice that the amplitude of observed COS flux variations is larger than the one of modelled fluxes. Kohonen et al. (2020) have quantified the relative uncertainty of weekly-averaged ecosystem COS fluxes at 40%, which is coherent with the large standard-deviation computed for field data (Figure 1b). RMSD for the seasonal cycle is 7.0 pmol m$^{-2}$ s$^{-1}$, and the relative RMSD is 41%. The bias is low (-0.3 pmol m$^{-2}$ s$^{-1}$), the standard deviations are similar: 6.6 pmol m$^{-2}$ s$^{-1}$ for the observations and 7.7 pmol m$^{-2}$ s$^{-1}$ for the simulated fluxes. At the Hyytiälä site in year 2015 (Figure B1b), the RMSD for the seasonal cycle is 2.4 pmol m$^{-2}$ s$^{-1}$, and the relative RMSD is 25%; the bias is low too (0.2 pmol m$^{-2}$ s$^{-1}$) and the standard deviations are also close: 3.6 pmol m$^{-2}$ s$^{-1}$ for the observations and 3.5 pmol m$^{-2}$ s$^{-1}$ for the simulated fluxes, the correlation coefficient is 0.78.

### 3.1.1.3 Nighttime fluxes

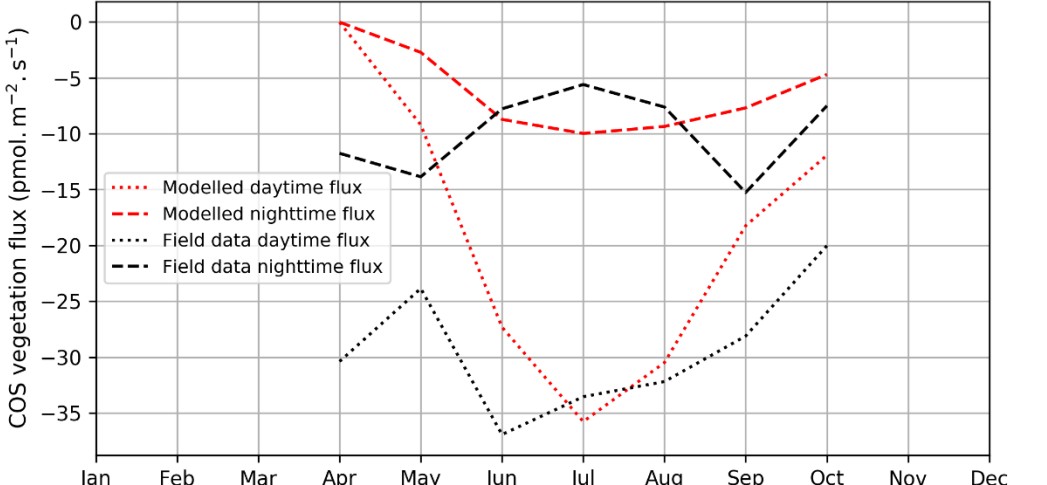

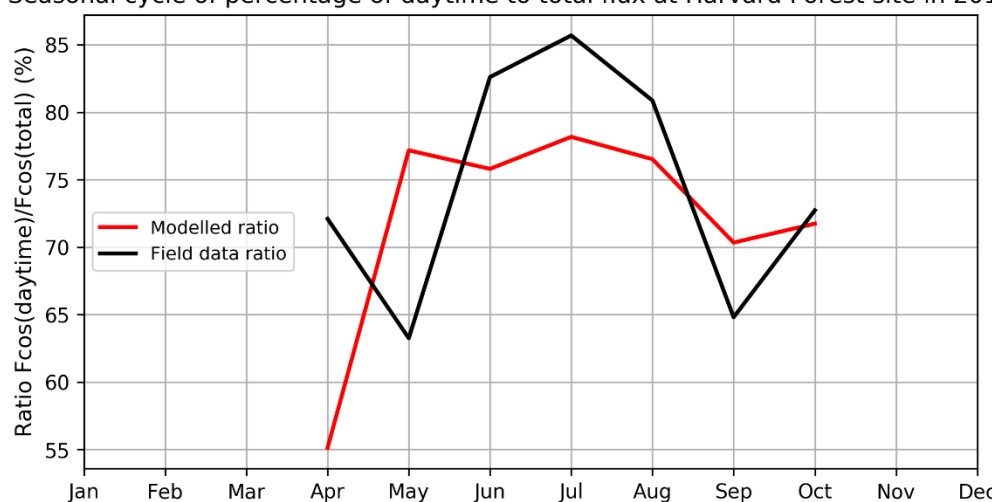

**Figure 2: a. Seasonal cycle of: a. daytime (dotted curve) and nighttime (dashed curve) for observed (black) and modelled (red) vegetation COS fluxes, b. percentage of the daytime to the total flux (solid curve), at Harvard Forest in 2012-2013**

Figure 2 compares mean daytime and nighttime observed and modelled vegetation COS fluxes and the percentage of the daytime to the total flux, computed for each month over 2012 and 2013 at the Harvard Forest site. We selected an arbitrary PAR threshold of 50 µmol m$^{-2}$ s$^{-1}$ to split between daytime and nighttime fluxes. We see that the modelled nighttime flux varies across the growing season, with a maximum uptake of -10 pmol m$^{-2}$ s$^{-1}$ reached in July and a lower absorption in the enclosing colder months. This seasonal variation can be explained by the seasonal change in LAI and the conductances dependency on $T_{air}$, which increases in summer. The observed nighttime fluxes are of the same magnitude but present an opposite seasonal cycle with lower uptake at the summer peak, albeit variations are within the one-standard deviation represented in Figure 1a. The modelled nighttime fluxes account from 22 % of the total COS uptake at the peak of the growing season to 45% in April at the very beginning. The observed ones exhibit slightly lower values, between 14 and 37%. At Hyytiälä, the modelled nighttime ratio is also slightly higher (between 30 and 34%) than the observed one (between 20 and 25%, Figure B2). These ratios are in line with other studies: Maseyk et al. (2014) reported a ratio of 29 ± 5% over a wheat field

in Oklahoma, and Sun et al. (2018b) one of 23% for the San Joaquin Freshwater Marsh site in California. The results may vary given the definitions adopted for nighttime and daytime periods.

### 3.1.2 Modelled conductances

To investigate the importance of each conductance in vegetation COS uptake we compared the three simulated conductances: leaf boundary layer, stomatal and internal, studying their variability and their drivers at the diel and seasonal scales. The boundary layer conductance to COS is higher than the two other conductances by a median factor larger than 25 (see Table A1 for more detailed statistics). As a high conductance value is equivalent to a low resistance to COS transfer, we focused only on the stomatal ($g_{S\_COS}$) and internal ($g_{I\_COS}$) conductances, which are the two most limiting factors to plant COS uptake.

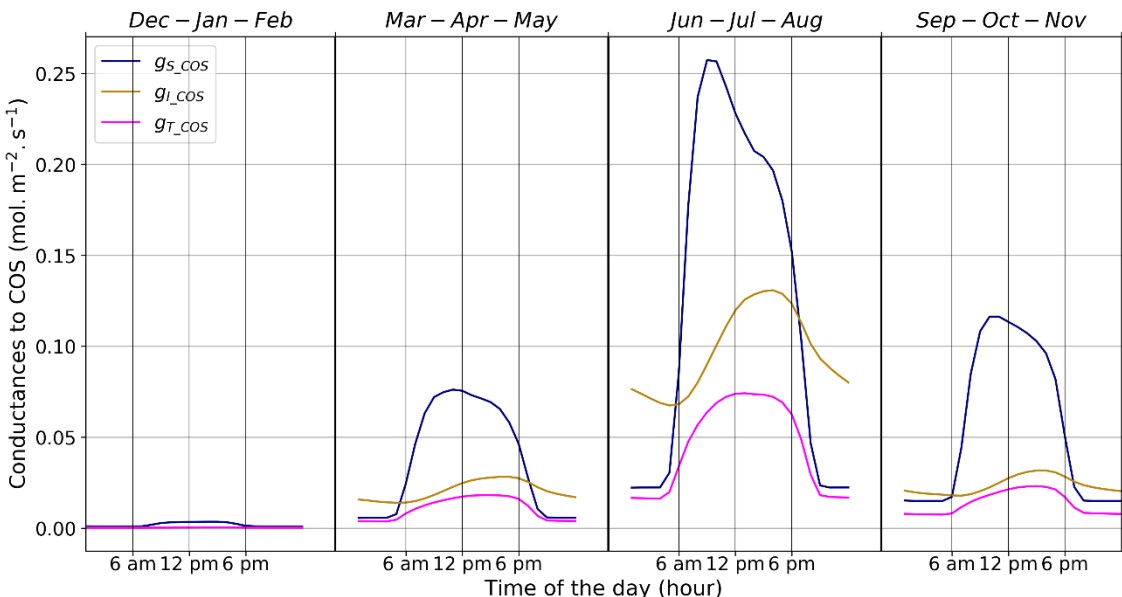

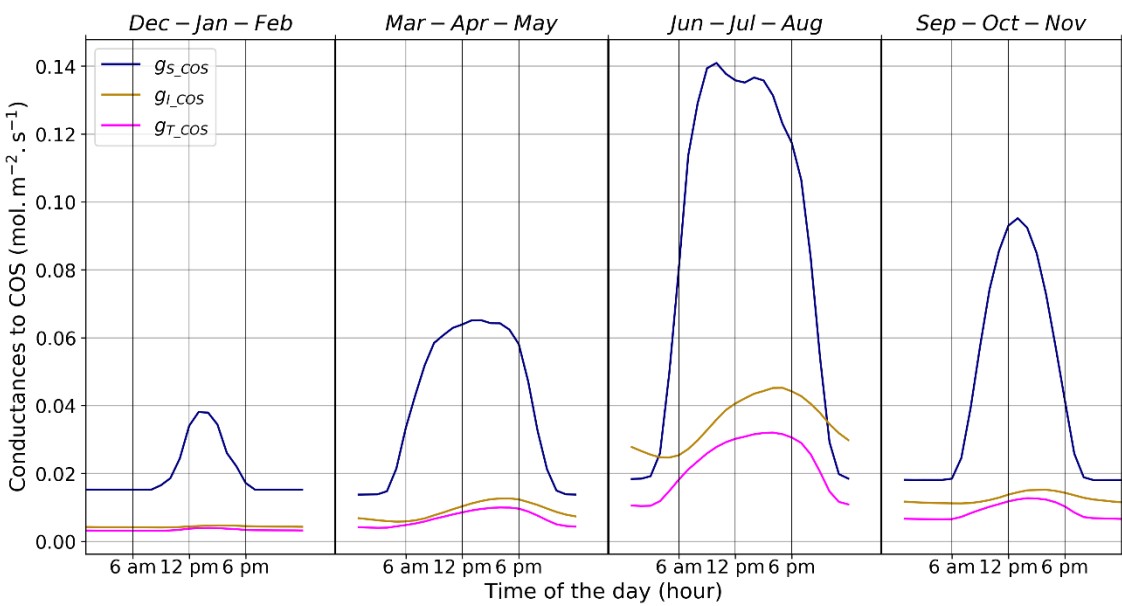

**Figure 3: Mean diel cycles of simulated conductances for each season at Harvard Forest in 2012 (a) and Hyytiälä in 2017 (b). The area reference for the units is m² ground area.**

Figure 3 presents the mean diel cycles of the simulated total, stomatal and internal conductances for each season, computed over 2012 at Harvard Forest and 2017 at Hyytiälä. For practicality, we shifted the month of December before the month of January of the same year to compute the winter mean. The seasonal variations are similar at both sites. The conductances, as well as the amplitude of their diurnal cycle, increase from winter to summer and decline in autumn. Harvard Forest is predominantly a deciduous forest and winter values of the conductances are zero at this site as there are no leaves in that season. Hyytiälä on the other hand is an evergreen pine forest, such that daytime stomatal conductance in winter does not become zero. The stomatal conductance is peaking between 9am and 1pm, depending on site and season, while the internal conductance is peaking later in the afternoon. The total conductance is in general limited by the internal conductance. The stomatal conductance is limiting roughly between 6pm and 6am from spring to autumn at Harvard, and only in June-July-August roughly between 9pm and 9 am at Hyytiälä.

These results are consistent with the results obtained at branch level by Kooijmans et al. (2019), who found that the COS flux was limited by the internal conductance in the early season, and later during daytime, while the effect of the stomatal conductance was larger at night. For the Harvard Forest site, Wehr et al. (2017) computed the stomatal conductance using both a water flux method and a COS flux method, and obtained a close agreement between two different methods; the mesophyll conductance is modelled using an experimental temperature response, and the biochemical conductance, representing CA activity, is modelled using a simple parameter (0.055 mol m$^{-2}$ s$^{-1}$), both scale with LAI to get canopy estimates. Wehr et al. (2017) found similar maximum values around 0.27 mol m$^{-2}$ s$^{-1}$ during daytime, from May to October, for the stomatal conductance and for the biochemical conductance (their Figure 4); adding the slightly larger mesophyll conductance (peaking around 1.0 mol m$^{-2}$ s$^{-1}$) to the biochemical conductance would thus also lead to a more limiting role of the internal conductance (peaking around 0.21 mol m$^{-2}$ s$^{-1}$) during daytime, albeit not as strong as for the modelled one (peaking around 0.13 mol m$^{-2}$ s$^{-1}$); the simulated stomatal conductance exhibits minimum and maximum values similar to the observation-based ones, but peaks more sharply in the morning.

To better understand the conductances behaviour, we studied the relative importance of their drivers. These include environmental variables directly or indirectly involved in their modelling: air surface temperature ($T_{air}$), photosynthetically active radiation ($PAR$), vapour pressure deficit ($VPD$) and soil moisture ($SM$), as well as LAI, as leaf-level conductances are summed over LAI layers to provide canopy-level conductances. Partial correlations are computed for all half-hourly values of the variables associated to LRU values between 0 and 8, and are provided in Table A2. We also used half-hourly ORCHIDEE outputs associated to LRU values between 0 and 8 to train Random Forests models for conductances at the two sites, taking into account the same five predictors. A random predictor was also added to check that the variable importance was correctly estimated. All RF models have an accuracy of at least 96%. Figures B3 and B4 present the relative ranking of the five predictors for the two conductances and the two sites. The ranking is different between the two methods (partial correlation versus RF), but they agree that at both sites the main driver for the internal conductance is air temperature and the main driver for the stomatal conductance is $PAR$.

As expected, $g_{I\_COS}$ mainly depends on $T_{air}$. This is explained by the fact that $g_{I\_COS}$ is proportional to $V_{max}$, which represents the Rubisco activity for CO$_2$; $V_{max}$ is assumed to be a measure for the mesophyll diffusion and for the CA activity for COS, which are the components of the internal conductance (Berry et al., 2013). $V_{max}$ depends on $T_{air}$, considered here as a proxy of the leaf temperature (Yin and Struik, 2009). This strong link

explains why $g_{I\_COS}$ is more limiting in winter, as $T_{air}$ is low with thus lower enzyme activities, and, as soon as $T_{air}$ rises in spring, $g_{I\_COS}$ becomes less limiting, especially at night. $PAR$ is the most important variable for the stomatal conductance at the two sites. Due to the way of how $g_{S\_COS}$ is simulated according to Yin and Struik (2009), there is a linear relationship with the $CO_2$ assimilation, which depends mainly on $PAR$.

### 3.1.3 LRU variability

LRU decreases as a function of PAR, as initially observed by Stimler et al. (2010). Kooijmans et al. (2019) made measurements in two branch chambers installed at the top of the canopy in two Scots pine trees in Hyytiälä. They plotted the response of LRU to light, as quantified by PAR. To compare the ORCHIDEE model behaviour to these field data, we determined an LRU using our modelled COS and GPP fluxes, considering a constant atmospheric concentration of 500 ppt for COS and global yearly values for $CO_2$.

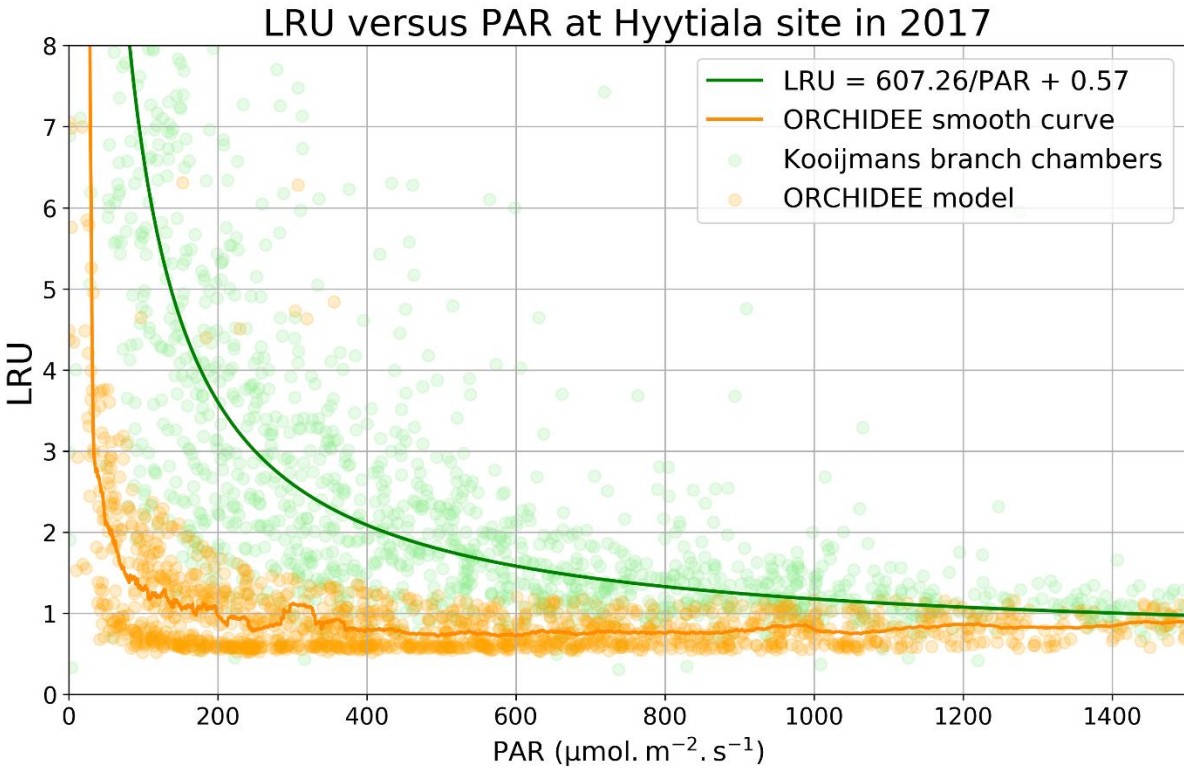

**Figure 4: LRU against PAR (Hyytiälä) for ORCHIDEE outputs and measurements (hourly data measured between 18 May and 13 July, Kooijmans et al., 2019). The light green circles represent average LRU values for chambers 1 and 2, light orange circles represent modelled LRU values. A moving average with a window of 50 points leads to the orange smooth curve for the model. The green line represents the function LRU=607.26/PAR + 0.57 from Figure S6 of the Kooijmans et al. (2019) supplement. To focus on LRU behaviour when PAR decreases, we plotted LRU response to PAR for PAR < 1500 μmol m⁻² s⁻¹.**

LRU increases with low PAR values for both branch chambers and for the model, and converge towards a constant value for high PAR values (Figure 4). This demonstrates that assuming a constant value for LRU, and not considering an increase in LRU under low light conditions, will result in erroneous estimation of COS fluxes. The increasing LRU can be explained by the light-dependence of the photosynthesis reaction contrary to the CA activity that is light-independent. Consequently, $CO_2$ fluxes tend to zero when PAR decreases while COS is still taken up in the dark, leading in theory to infinite values of LRU. The drop of LRU when PAR increases is however

much sharper in the model that in the observations. It is to be noted that we compare here LRU values estimated from measurements at branch level to modelled LRU estimated at canopy level. We conducted a similar modelling study considering only the top of canopy level and the associated COS and GPP fluxes, yielding similar results (not shown). This can be linked to the fact that the version of ORCHIDEE we use considers all the incoming light to be diffuse, and does not distinguish between sun and shaded leaves. We thus have similar LRU values at all canopy levels.

Following the model developed in Seibt et al. (2010, their equation (8)), the LRU explicitly depends on only two variables: the $g_{S\_COS}$ to $g_{I\_COS}$ ratio, and the ratio of the $CO_2$ intracellular concentration, $C_i$, to $[CO_2]_a$ (equally named $C_a$) ratio. The modelled daily mean values for the $C_i/C_a$ ratio computed at the two sites vary between 0.68 and 1.00 (Figure B5). These variations are in agreement with Prentice et al. (2014) who state that the $C_i/C_a$ ratio is pretty stable with only ± 30% variations. These values are on the upper part of the range reported in Seibt et al. (2010, their Table 2); following their Figure 3, for a given $C_i/C_a$ ratio a larger $g_{S\_COS}$ to $g_{I\_COS}$ ratio implies a lower LRU, in consistence with our results.

We also performed a predictor ranking for LRU, as was done previously with conductances. The predictors rank similarly for the two sites, as shown in Figure B6, the main factors explaining the variability of the simulated LRU at a half-hourly time step are $PAR$, $T_{air}$ and LAI.

### 3.2 Global scale plant COS fluxes and Study of LRU values

### 3.2.1 Comparison of plant COS uptake sink estimates

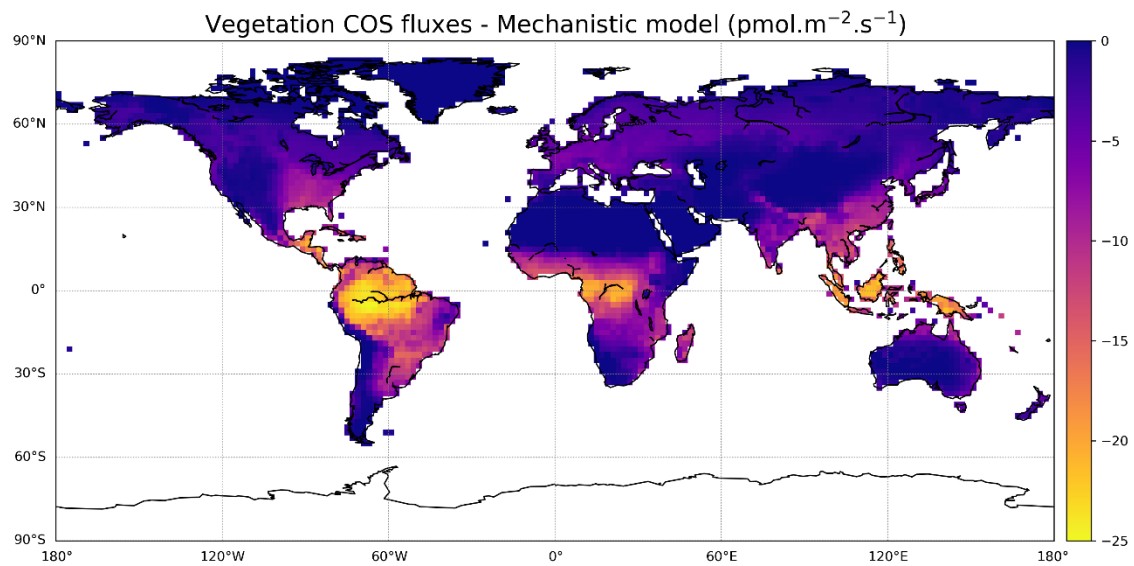

**Figure 5: Map of average vegetation COS fluxes over the 2000-2009 period, from the mechanistic model as implemented in ORCHIDEE**

The mechanistic approach simulated in the ORCHIDEE model gives a plant COS uptake of -756 Gg S yr[-1] over the 2000-2009 period. COS fluxes are the strongest in South America, Central Africa and Southeast Asia (Figure 5), as expected as these regions are also the most productive ones for GPP.

**Table 5: Overview of COS plant uptake per year (Gg S yr[-1])**

| | Kettle et al. (2002) | Montzka et al. (2007) | Suntharalingam et al. (2008) | Berry et al. (2013) | Launois et al. (2015b) | | | This study |
| --- | --- | --- | --- | --- | --- | --- | --- | --- |
| | | | | | ORC. | LPJ | CLM4 | |
| Period study | circa 1990-2000 | 2000-2005 | 2001-2005 | 2002-2005 | 2006-2009 | | | 2000-2009 |
| Uptake by plants | -238 (±30) | -730 to -1500 | -490 (-460 to -530) | -738 | -1335 | -1069 | -930 | -756 |

The more recent studies (Montzka et al., 2007; Suntharalingam et al., 2008; Berry et al., 2013; Launois et al., 2015b) show a higher global plant sink than the one initially found by Kettle et al. (2002) (Table 5). Kettle et al.
(2002) used an LRU-like approach, based on NPP and on the NDVI temporal evolution, and already acknowledged their estimate was assumed to be a lower bound one. Estimates from plant chambers and atmospheric measurements (Sandoval et al., 2005; Montzka et al., 2007; Campbell et al., 2008) confirmed that the COS plant sink should be twofold to fivefold larger than estimated in Kettle et al. (2002). Suntharalingam et al. (2008) also found a low estimate of -490 Gg S yr⁻¹, using 3D modelling of COS atmospheric concentrations, constrained by
surface site observations. We note that our estimate is similar to the -738 Gg S yr⁻¹ found by Berry et al. (2013), which was implemented in the Simple Biosphere (SiB) 3 LSM. The reason for this similarity can be that, on top of using the same mechanistic model for vegetation COS uptake, the leaf photosynthesis and stomatal conductance in both LSMs are derived from the same classical models from Farquhar et al. (1980), Collatz et al. (1992) and Ball et al. (1987).
Launois et al. (2015b) adopted an LRU approach, using constant LRU values for large MODIS vegetation classes, adapted from Seibt et al. (2010). Based on these values and a set of global GPP estimates from three LSMs (ORCHIDEE, LPJ, CLM4), the authors derived the corresponding global vegetation COS uptakes reported in Table 5. The selection of the LSM itself thus introduces an uncertainty on the global vegetation COS uptake of around 40% in this case.
Applying the LRU values derived from Seibt et al. (2010) (Table 1) to the global GPP simulated in this study leads to the highest plant COS uptake with -1343.3 Gg S yr⁻¹. Seibt et al. (2010) report LRU values for different internal conductance limitations. The LRU values that we used here represent a small limitation of internal conductance to the total COS uptake (the ratio of stomatal to internal conductances is 0.1). A smaller global COS uptake can be expected when the LRU values with a more limiting effect of the internal conductance are used. Applying the
LRU values derived from Whelan et al. (2018) (Table 1) leads to an intermediate estimate of -808.3 Gg S yr⁻¹, which is closer to the global uptake obtained with the mechanistic model. This analysis shows that the choice for certain LRU values introduces an uncertainty on the global vegetation COS uptake (around 70% in this case), and highlights the importance of deriving accurate PFT-dependent LRU values.

**3.2.2 Dynamics of simulated LRU values**

The PFT distributions of the LRU values, both those computed using equation (1) applied to the monthly climatology of mechanistic COS and GPP fluxes over the 2000-2009 period (LRU_MonthlyFluxes), and the climatological monthly means computed directly from the original half-hourly values (Monthly_LRU), do not support the idea of a constant PFT-dependent LRU value (Figure 6).

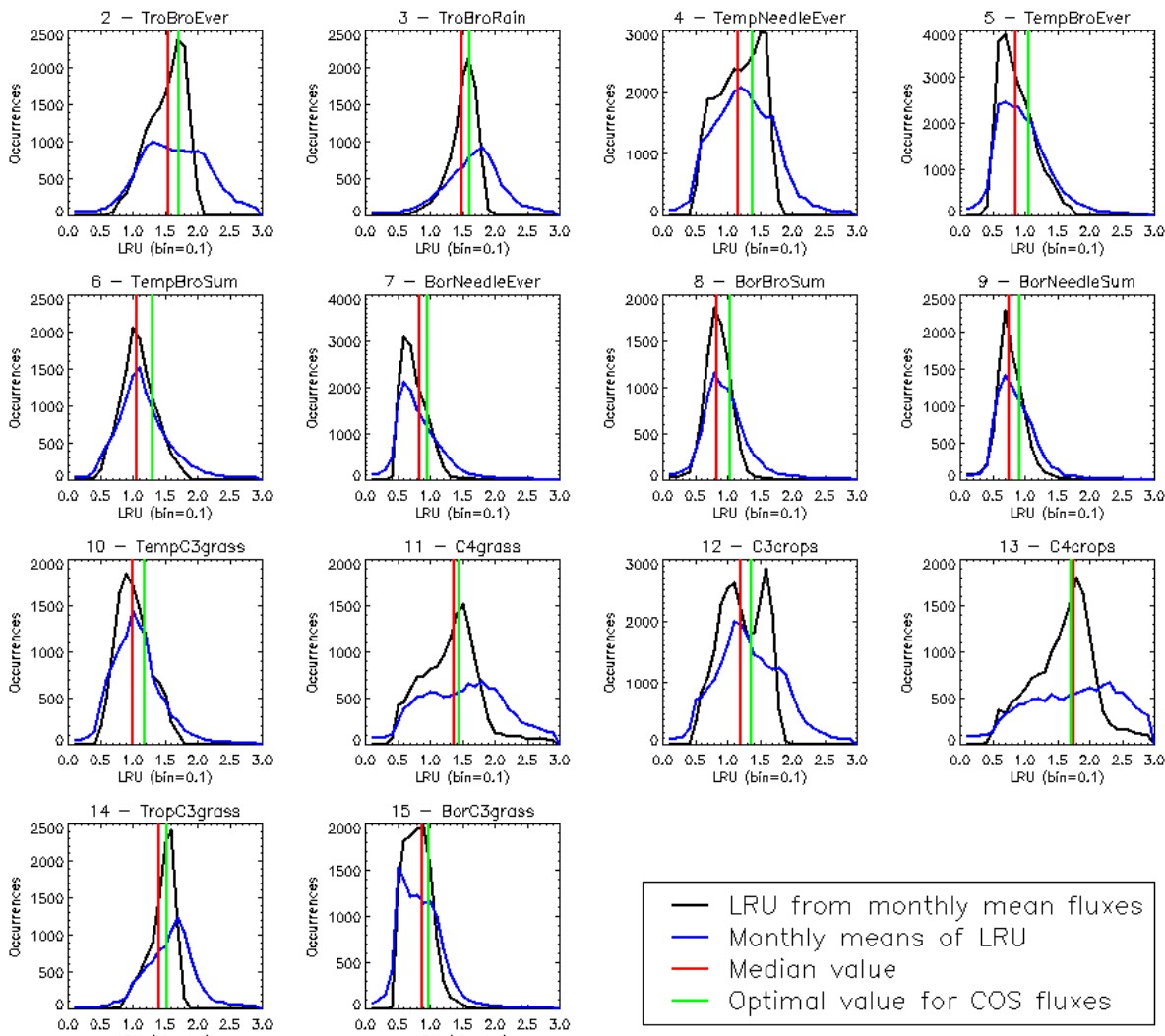

Figure 6: Distributions of the LRU values computed from the mechanistic approach over the 2000-2009 period. Each subplot represents one of the 14 vegetated PFTs used in ORCHIDEE, considering all grid cells where the PFT is present. The x-axis represents the LRU value between 0 and 3, with 0.1 bins. The y-axis represents the occurrences. For each PFT, the black distribution is computed using a monthly climatology of simulated COS and GPP fluxes (LRU_MonthlyFluxes), the blue distribution is computed using the monthly climatology of LRU values estimated at the original half-hourly time step (Monthly_LRU), the red vertical bar represents the median LRU value for LRU_MonthlyFluxes, the green vertical bar represents the LRU optimal value that minimizes the error between plant COS uptakes estimated at a monthly time step by the mechanistic approach and the LRU approach, for all pixels of the considered PFT (see names and abbreviations in Table 1).

The distributions are usually not gaussian; nor are they all unimodal, as is the case for PFT 12 C3 Agriculture. The distributions for C4 PFTs (PFT 11 C4 Grass and PFT 13 C4 Agriculture) exhibit a large spread. The median values are represented by vertical red bars in Figure 6 and listed in Table 1. The optimal values ($LRU\_Opt$) obtained by linearly regressing monthly COS fluxes against monthly GPP fluxes multiplied by the ratio of the mean COS to $CO_2$ concentrations (see Figure C1) are represented by vertical green bars and also listed in Table 1. They are usually higher than the median values, with a mean difference of 12.1%. Using either monthly means or yearly means of fluxes gives very similar optimal LRU values, the mean difference being only -0.2%.

The LRU values from monthly fluxes (LRU_MonthlyFluxes) tend to be lower than the monthly means of the LRU computed at a half-hourly time step (Monthly_LRU). This is visible in Figure 6 where the blue distributions yield larger LRU values, and in the bi-dimensional histogram of LRU_MonthlyFluxes against Monthly_LRU (Figure





C2). The bias is -0.2 and the correlation is 0.67. This shows that LRU is scale dependent. The values to be
considered should be coherent with their usage. For example, the optimal values we computed are lower than
values estimated from measurements, but they are adapted to make the link with atmospheric COS studies.

$LRU\_Opt$ values are much smaller than LRU_Seibt values for all PFTs, roughly by a factor 2. They are closer to
the LRU_Whelan values, being smaller for all C3 PFTs except the Tropical Broad-leaved Evergreen Forests, and
higher for C4 PFTs (Table 1). In the $LRU\_Opt$ set, the most productive PFTs (tropical forests and C4 crops) have
the highest values around 1.7, while the less productive PFTs (boreal forests and grasses) have the lowest values
around 0.9. To the contrary, in the LRU_Seibt set, temperate broad-leaved forests have the highest values (3.6)
while needleleaf forests have the smallest value around 1.9.

Another way to understand the distribution of LRU values is to look directly at the scatter plots of monthly COS
fluxes against GPP fluxes, multiplied by the ratio of COS to $CO_2$ concentrations (Figure C1). For most PFTs, it is
in fact obvious that the relationship shows non-linear features, disagreeing with the classical linear LRU model.
Based on these findings, we fitted a simple exponential model as:

$$F_{COS} = a\left(e^{bGPP\frac{[COS]_a}{[CO_2]_a}} - 1\right)$$

with two parameters a and b. However, given the large spread of the data around the model, the Akaike criterion
is always favourable to the LRU linear model, so we won't investigate further with this exponential model, more
specific research is needed here in order to bridge this data gap. Still, it is important to note that the larger COS
fluxes will in general be underestimated using a linear LRU approach. It also appears that in certain PFTs (4, 5, 7)
small COS fluxes will be underestimated.


We computed mean annual vegetation COS fluxes using our modelled GPP and this new $LRU\_Opt$ set of values
and compared them to the mechanistic COS fluxes (Figure 7a).

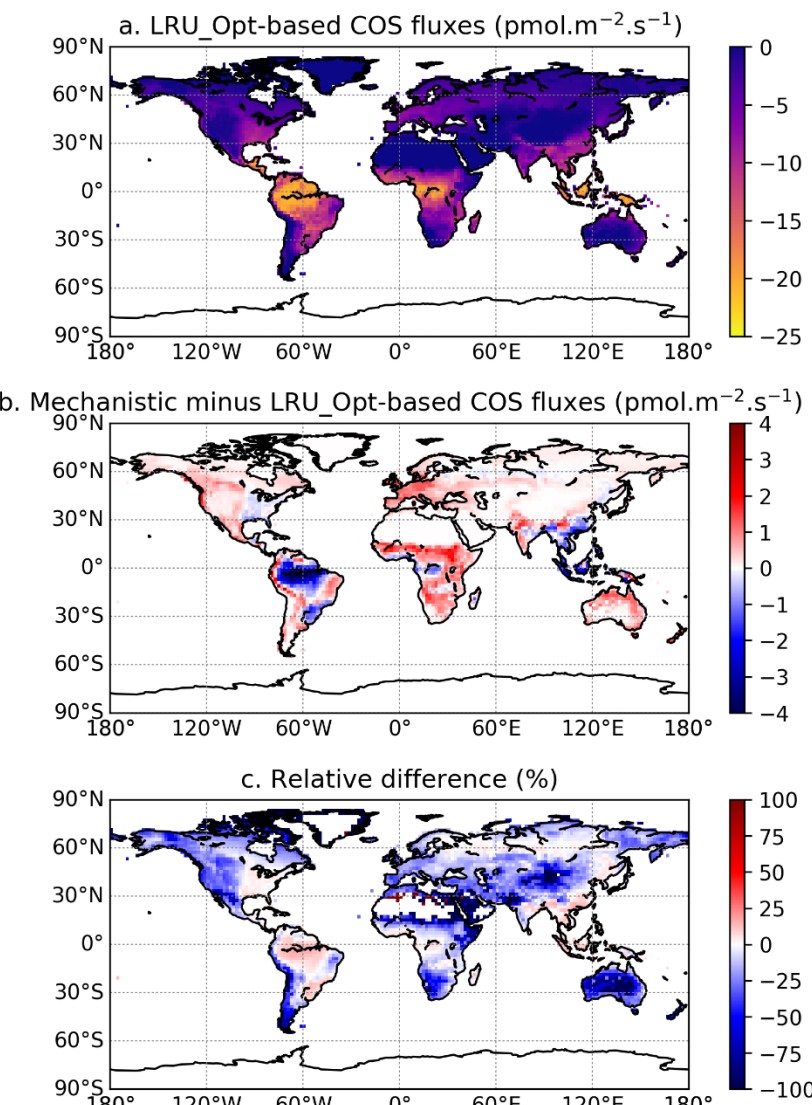

**Figure 7: a. Mean annual vegetation COS fluxes for the 2000-2009 period fluxes computed using a linear LRU approach with optimal values for each PFT. b. Differences between mechanistic and LRU-based fluxes. c. Relative difference (%)**

The maps of differences between the mechanistic and $LRU\_Opt$-based COS fluxes (Figure 7b), and relative differences (Figure 7c), provide evidence for the spatial errors introduced by considering a constant LRU value. The differences are always lower than 4 pmol m$^{-2}$ s$^{-1}$ in absolute values, and are mainly positive, with the main exception over the Amazon region where the mechanistic approach shows a larger uptake than the linear LRU approach. The difference between the global estimates of the two approaches is less than 2%; we could still improve the linear regression determining the LRU optimal value by weighting grid-cell fluxes with the corresponding surface of the PFT.

We also compared the mean seasonal cycles of the COS vegetation flux over the 2000-2009 period, for the mechanistic approach and the $LRU\_Opt$-based approach, for each PFT (Figure C3). The seasonal cycles are very similar; for PFT 13 C4 Agriculture, the $LRU\_Opt$-based cycle is slightly in advance as compared to the mechanistic cycle.

### 3.3 Simulating atmospheric COS concentration at surface stations

We transported the global COS and $CO_2$ fluxes (i.e. the ones obtained from the ORCHIDEE model plus the additional components of each cycle, listed in Table 2 and Table 3) with the LMDz6 atmospheric transport model as described in Sect. 2.4.2. We analysed COS concentrations derived from simulated COS fluxes obtained with the mechanistic and LRU approaches in regards with observed COS concentrations from the NOAA at a few selected sites.

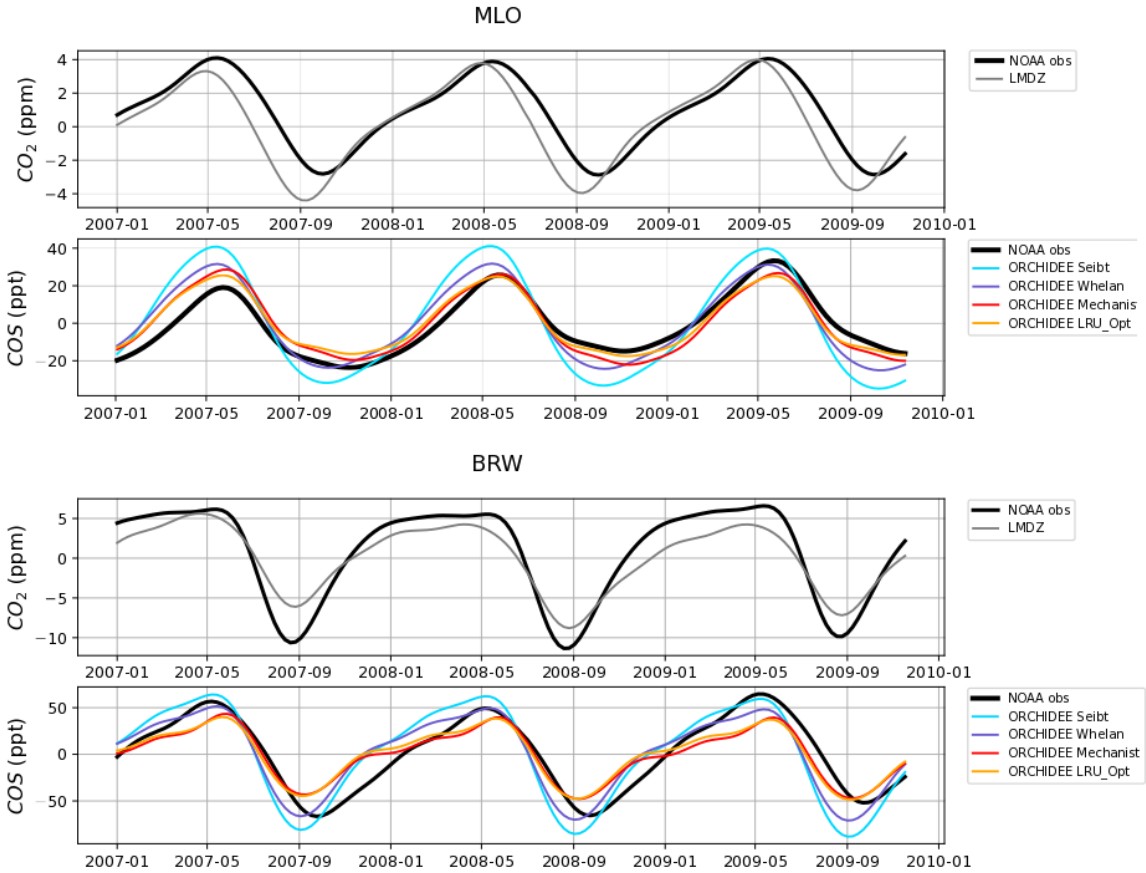

**Figure 8: Detrended temporal evolutions of simulated and observed $CO_2$ and COS concentrations at two selected sites, for the mechanistic (ORCHIDEE Mechanist) and LRU approaches (ORCHIDEE Seibt, ORCHIDEE Whelan, ORCHIDEE LRU_Opt), simulated with LMDz6 transport between 2007 and 2009. The ORCHIDEE LRU_Opt line (orange) corresponds to the concentrations simulated using the optimal LRU values derived from the mechanistic model. Top: Mauna Loa station (MLO, Hawaii), bottom: Barrow station (BRW, Alaska). The curves have been detrended beforehand and filtered to remove the synoptic variability (see Sect. 2.2.4)**

Figure 8 shows the detrended temporal evolution of $CO_2$ and COS concentrations for the mechanistic and LRU approaches at Barrow (BRW, Alaska) and Mauna Loa (MLO, Hawaii). The MLO site samples air masses coming from all over the northern hemisphere (Conway et al., 1994). $CO_2$ seasonal amplitude at BRW reflects the contributions of surface fluxes from high latitude ecosystems (Peylin et al., 1999), but also from regions further south due to atmospheric transport (Parazoo et al., 2011; Graven et al., 2013). These two stations have been used to detect large-scale changes in ecosystem functioning (Graven et al., 2013; Commane et al., 2017). In spite of their importance, LMDz driven by the ORCHIDEE vegetation fluxes has difficulties in representing their seasonal cycles. For instance, at MLO, the simulated seasonal amplitude of $CO_2$ is overestimated and precedes the observations by one month.

For COS, the simulated concentrations match relatively well the observed seasonal variations and seem to be more in phase with the observations than for $CO_2$. Such a feature could indicate that the phase issues with $CO_2$ is not primarily driven by GPP issues but by the other $CO_2$ flux components. The mechanistic model and its LRU optimal equivalent better reproduce the observed one-month lag between the COS and the $CO_2$ simulation at MLO (i.e. the minimum COS lags the one of $CO_2$) than the other LRU approaches with values from Whelan et al. (2018) and Seibt et al. (2010). The simulations differ more in the amplitude than in the phase of their seasonal cycles. The mechanistic approach simulates an amplitude lower than the LRU ones. At MLO for example, the lower amplitude of the mechanistic model is in better agreement with the observations. At BRW, its seasonal amplitude is also lower but is now underestimated. The COS concentration at this station from the mechanistic approach varies between +30 ppt and -50 ppt while it varies between +50 ppt (respectively +37) and -71 ppt (respectively -50) for the simulation based on Seibt et al. (2010) (respectively Whelan et al., 2018). This is a direct consequence of lower COS fluxes with the mechanistic model compared to the fluxes based on Seibt and Whelan LRU approaches. At both the MLO and BRW sites, the difference between the mechanistic model and its LRU optimal equivalent after being transported is lower than 8 ppt, within the range of the observations uncertainty.

**Table 6: Normalized standard deviations (NSDs) of the simulated concentrations by the observed concentrations. Within brackets are the Pearson correlation coefficients (R) between simulated and observed COS concentrations for the mechanistic and LRU approaches, calculated between 2004 and 2009 at 10 NOAA stations. For each station, NSD and R closest to one are in bold and farthest ones are in italic. The time-series have been detrended beforehand and filtered to remove the synoptic variability (see Sect. 2.2.4).**

|  | SPO | CGO | SMO | KUM | MLO | NWR | LEF | MHD | BRW | ALT |
|---|---|---|---|---|---|---|---|---|---|---|
| ORCHIDEE Seibt | *1.15* | *0.67* | **0.58** | 1.32 | *1.65* | *2.12* | *2.17* | *1.52* | 1.25 | 1.16 |
|  | *(0.96)* | *(0.5)* | (-0.47) | *(0.92)* | *(0.89)* | *(0.50)* | *(0.92)* | **(0.96)** | *(0.90)* | **(0.95)** |
| ORCHIDEE Whelan | **1.00** | 0.83 | 0.40 | **1.03** | 1.23 | 1.50 | 1.67 | 1.26 | **1.00** | **0.92** |
|  | (0.97) | (0.91) | (0.1) | (0.93) | (0.90) | (0.52) | (0.93) | (0.94) | *(0.90)* | *(0.94)* |
| ORCHIDEE mechanist | 1.10 | **1.01** | *0.35* | 0.90 | **1.05** | 1.26 | **1.34** | 1.09 | 0.69 | *0.64* |
|  | (0.97) | **(0.97)** | **(0.4)** | **(0.95)** | **(0.92)** | **(0.63)** | **(0.94)** | *(0.85)* | **(0.91)** | **(0.95)** |
| ORCHIDEE LRU_Opt | 1.02 | 0.98 | 0.34 | *0.85* | 0.94 | **1.21** | **1.34** | **1.04** | *0.68* | *0.64* |
|  | **(0.98)** | **(0.97)** | *(-0.5)* | (0.94) | **(0.92)** | *(0.50)* | **(0.94)** | (0.88) | **(0.91)** | **(0.95)** |

Table 6 presents the NSDs and Pearson correlation coefficients between simulated and observed COS concentrations for the mechanistic and LRU approaches. We see that the simulation with Seibt et al. (2010) intermediate LRU values overestimates the seasonal standard deviation and has the lowest accuracy for most stations. It is difficult to tell whether the mechanistic model is better than the LRU approach based on Whelan values. While the mechanistic approach captures known features of the temporal dynamics of the COS to $CO_2$ flux ratio, it underestimates the simulated concentrations at Alert (ALT, Canada) and Barrow (BRW, United States). It should be noted that, due to other sources of errors (in particular transport and oceanic emissions), the comparison presented here should be taken as a sensitivity study of COS seasonal cycle to the vegetation scheme rather than a complete validation of one approach.

**4 Discussion**

**4.1 How can we use COS fluxes and the mechanistic COS model to improve the simulated GPP?**

The mechanistic model links vegetation COS uptake and GPP fluxes through the stomatal conductance model, which includes the minimal conductance as an offset, and the common use of the carboxylation rate of Rubisco, $V_{max}$, in the internal conductance formulation for COS, and in the Rubisco-limited rate of assimilation for $CO_2$. The downside is the introduction of the somewhat uncertain $\alpha$ parameter that relates the COS internal conductance to $V_{max}$. Using COS flux measurements to optimize the parameters of the stomatal and internal conductances would thus in principle benefit the simulated GPP. This optimization may be done based on appropriate data assimilation techniques; for example, Kuppel et al. (2012) optimized key parameters of the ORCHIDEE model related to several processes including photosynthesis (see their Table 2), by assimilating eddy-covariance flux data over multiple sites. The approach relies on a Bayesian framework where a cost function including uncertainties on observations, model and parameters is minimized (Tarantola, 1987). The results obtained in this study pave the way for a similar approach using COS fluxes to optimise key parameters controlling GPP; they can be used to define an optimal set up for the a priori errors and the error correlations in a Bayesian framework. We acknowledge however the scarcity of available measurements for the time being, with no samples for most biomes, a few sites with less than one year of data, and only Hyytiälä allowing for interannual variability studies.

**4.1.1 First step: Improving the mechanistic modelling of vegetation COS fluxes**

Without any calibration, the chosen mechanistic model was able to reproduce observed vegetation COS fluxes at the Harvard Forest and Hyytiälä sites with relative RMSDs on the order of 40%. Regarding conductances, differences are also seen between the diel cycles of simulated and observation-based conductances from (Wehr et al., 2017). Diel variations in atmospheric $[COS]_a$, not accounted for in our model, cannot explain these differences, as they would only affect $F_{COS}$ but not the conductances. These discrepancies advocate for the assimilation of COS fluxes to optimize the parameters related to the internal and stomatal conductances. In our modelling framework, the internal conductance is assumed to be the product of $V_{max}$ by the $\alpha$ parameter. This parameter has been calibrated by Berry et al. (2013) using gas exchange measurements of COS and $CO_2$ uptake (Stimler et al., 2010; Stimler et al., 2012).. As this $\alpha$ parameter seems much more uncertain as compared to the relatively well known $V_{max}$, we should first try to optimize $\alpha$ keeping $V_{max}$ fixed.

**4.1.2 Exploiting the alternative dominant role between stomatal and internal conductances**

Without being perfect, the mechanistic model could reproduce some expected behaviours, such as the limiting role of the internal conductance in winter and then during daytime in the growing season, in relation to the control of CA activity and mesophyll diffusion by temperature, as also depicted in Kooijmans et al. (2019). Determining the limiting conductances to COS uptake depending on the time of day provides useful information, as it can be used to better target which model parameters to optimize, using data assimilation approaches. Thus, observations made in the morning and early afternoon could be used to better constrain the $\alpha$ parameter when the internal conductance is limiting COS fluxes, at least as modelled on the C3 species of the two sites, and we could investigate whether the $\alpha$ parameter should be further quantified per PFT rather than simply per photosynthetic pathway. It is to be noted that for C4 species, the internal conductance is larger than for C3 species by a factor ten, so that stomatal

conductance is limiting, and it could be difficult and useless to try optimizing internal conductance using the $\alpha$ parameter. We have to acknowledge the large uncertainty regarding the modelling of the internal conductance. In parallel to optimizing the parameters of the internal conductance, an improvement could thus also be to replace it by the two factors it represents, i.e. the mesophyll conductance and CA activity. A model for the mesophyll

conductance is already implemented in ORCHIDEE, with a simple parameter depending on temperature through a multiplication by a modified Arrhenius function following Medlyn and al. (2002) and Yin & Struik (2009). The impact of mesophyll conductance on photosynthesis and water use efficiency is now more studied (e.g. Buckley and Warren, 2014), even if its modelling remains challenging too: the temperature response has notably been reported as highly variable between plant species (von Caemmerer and Evans, 2015), which would imply having

PFT-dependent parameters. Regarding measurements, [13]C discrimination of the isotopic composition of $CO_2$ exchanges allows for an estimation of the mesophyll conductance (Stangl et al., 2019). Concerning CA activity, we could test the simple model using a constant value presented in Wehr et al. (2017). Measuring CA activity can be done at a coarse frequency, using different techniques (Henry, 1991).

### 4.1.3 Exploiting nighttime conductances

Recent studies have shown that nighttime field measurements of stomatal conductances often exhibit larger values than the ones used in models (Caird et al., 2007; Phillips et al., 2010). In the ORCHIDEE model, minimum stomatal conductances to $CO_2$, $g_0$, take two different values: 6.25 mmol m$^{-2}$ s$^{-1}$ for C3 species and 18.75 mmol m$^{-2}$ s$^{-1}$ for C4 species. However, Lombardozzi et al. (2017), using data from literature, found that observed nighttime conductances to $CO_2$ range from 0 mmol m$^{-2}$ s$^{-1}$ to 450 mmol m$^{-2}$ s$^{-1}$ with an overall mean value of 78 mmol m$^{-2}$

s$^{-1}$. Moreover, they defined a mean value for each PFT (see Table A3) while the ORCHIDEE model uses one value for all C3 species and another one for all C4 species. Using higher nighttime stomatal conductances in models has the impact of increasing plant transpiration and reducing available soil moisture, which alters water and carbon budgets, especially in semi-arid regions (Lombardozzi et al., 2017). Lower $VPD$ values at night, that could limit the impact of higher nighttime stomatal conductances, follow however an increasing trend (Sadok and Jagadish,

2020). A better representation of these minimal conductances in the model could then improve the constraint of gas exchange between the atmosphere and the terrestrial biosphere. It is to be noted that Barnard and Bauerle (2013) found, based on sensitivity analyses, that $g_0$ was the parameter having the largest influence on their modelled transpiration estimates. They also stress that $g_0$ should maybe be seen as an asymptotic minimal value, rather than an offset. During nighttime, the stomatal conductance limits COS uptake. In the model, the nocturnal

stomatal conductance to COS is calculated from the above-mentioned minimum stomatal conductance values. For now, the absolute vegetation COS fluxes at night are slightly overestimated as compared to observed fluxes (updated Figure 1a for Harvard and Figure B1a for Hyytiälä), thus hinting to overestimated nighttime stomatal conductances. Therefore, nighttime observations of COS fluxes could be used to optimize the minimum stomatal conductance values for each PFT.

We thus see that COS fluxes could be used, through standard data assimilation techniques, to optimize the model parameters related to conductances, thus contributing to the improvement of the GPP. However, many more COS flux measurements are needed over a large variety of biomes, first to assert the validity of the mechanistic COS model at global scale, and second to be assimilated in order to improve simulated conductances and GPP estimates.

## 4.2 The mechanistic versus LRU approach

The mechanistic model is able to reproduce the high temporal frequency LRU variations observed at sites. It is thus legitimate to consider this approach as more accurate than the classical linear LRU approach that uses a time-constant LRU value per PFT to estimate COS fluxes from GPP. Furthermore we have shown that computing LRU values using equation (1) applied to from monthly mean fluxes yields values lower than computing monthly means of high-frequency LRU values (Figure 6). This may explain why the LRU values we have thus estimated from monthly mean fluxes show generally lower values than the ones derived from measurements, although these cover a large range from 0.7 to 6.2 (Seibt et al., 2010; Whelan et al., 2018). More recently, Spielman et al. (2019) estimated LRU values from ecosystem and soil measurements: 0.89 for an agricultural soybean field, 1.02 for a temperate C3 grassland, 2.22 for a temperate beech forest and 2.27 for a Mediterranean savanna ecosystem; our corresponding PFTs respectively give: 1.37 (C3crops), 1.18 (TempC3grass), 1.31 (TempBroSum) and 1.06 (TempBroEver), with thus higher estimates for herbaceous plants and lower ones for trees. It is difficult to say whether in situ and laboratory measurements are too sparse and not representative enough of the variability of plants and environmental conditions across the globe to have a reasonable confidence in their derived mean or median LRU values, or if we can use these LRU values to falsify the modelled COS and/or GPP fluxes. We may also add that LRU values derived from measurements performed in leaf chamber measurements, that are well-ventilated and thus associated with large leaf boundary layer conductances, may not be representative of the real-world transfer processes, where the boundary layer conductances vary with wind speed, temporally and within canopy depth (Wohlfahrt et al., 2012).

Without any calibration, the mechanistic approach performs similarly to LRU approaches based on monthly mean fluxes, when COS is transported using all known COS fluxes as inputs, and COS concentrations are evaluated at stations of the NOAA network. We have now a much finer representation of the COS fluxes as, at every timestep, the model integrates the plant's response to environmental conditions in the calculation of the internal and stomatal conductances, unlike in the LRU approach which uses constant values for each PFT.

In order to quantify the first order uncertainty on $F_{COS}$ related to the fact that we have used a constant $[COS]_a$ in our implementation of the Berry model, we computed an alternative $F'_{COS}$, using the LRU approach based on a climatology of hemispheric monthly means of COS atmospheric concentrations (Montzka et al., 2007), the optimal LRU we derived in this study (given in Table 1), average yearly values for $CO_2$ atmospheric concentrations, and a climatological seasonal cycle of simulated monthly GPP per PFT. Over the 2000-2009 period, the mean difference between the mean seasonal COS fluxes computed with this method ($F'_{COS}$) and the ones simulated with the mechanistic model ($F_{COS}$) amounts to -7.9% over the Northern hemisphere. As expected, the seasonal amplitude of COS fluxes is dampened as $[COS]_a$ decreases with vegetation growth. We thus have to improve our methodology to consider a varying $[COS]_a$ as was done in Berry et al. (2013), either inside the ORCHIDEE model, or as a post-processing. This requires devising some trade-off between the high-frequency timestep of ORCHIDEE and the cost of running the transport model. However, it is to be noted that there is no impact on the derived LRU values as the LRU does not depend on the considered $[COS]_a$, as long as the same one is considered for the computation of the COS fluxes in the mechanistic model (Eq. (3)) and for the computation of the LRU (Eq. (1)) (i.e. whether fixed or varying monthly).

However, there is currently a larger uncertainty on other COS fluxes in the global COS budget, which have an important impact on simulated COS concentrations (Ma et al., 2020) and their relative seasonal changes. For example, if we use another estimation of the direct oceanic fluxes (Lennartz et al., 2017), that shows a seasonal cycle whose amplitude is comparable to the one from the vegetation in high latitudes, this results in an overestimated seasonal cycle at all sites, with the mechanistic approach having the most realistic seasonal amplitude (see Appendix D1 and Figure D1). An additional sensitivity test was performed to assess the impact of indirect oceanic emissions via DMS oxidation on simulated seasonal cycles as the importance of these fluxes in the global COS budget is still debated (Whelan et al., 2018). Whereas the impact on northern sites is negligible, the removal of indirect oceanic emissions via the DMS of Kettle et al. (2002) decreases the seasonal amplitude of southern sites (CGO and SPO) in the same proportion in all experiments (see Appendix D2 and Table D2). Transport errors also add uncertainties on the simulated concentrations, especially at continental elevated sites (Remaud et al., 2018). Plus, given the present discrepancies between the GPP estimates of different land surface models, it can be argued that using a mechanistic model instead of an LRU approach when comparing COS concentrations seems to be of a second order importance (Campbell et al., 2017; Hilton et al., 2017). We nevertheless note in this study that we found an uncertainty on the global vegetation COS uptake of 40% when considering three different LSMs (Launois et al., 205b), to be compared to an uncertainty of 70% when considering three LRU datasets.

Setting aside the uncertainty for the moment, how could we use atmospheric COS concentrations to constrain GPP? A first optimization was performed with the ORCHIDEE model in Launois et al. (2015b), who optimized a single scaling parameter applied on the vegetation COS fluxes simulated with the LRU approach, thus equivalent to a scaling factor applied on the GPP or the LRU. They assimilated the atmospheric COS concentrations measured at the NOAA air sampling stations, using the LMDz transport model (Hourdin et al., 2006) and a Bayesian framework as in Kuppel et al. (2012). The optimization reduced in absolute value the estimated global vegetation COS uptake from -1335 Gg S yr$^{-1}$ to -708 Gg S yr$^{-1}$, more in line with this work's estimate based on a mechanistic modelling of vegetation COS uptake. A mid-term perspective is to go beyond a single scaling parameter, and to optimize a set of ORCHIDEE parameters using both atmospheric COS and $CO_2$ data. Such an approach has been used in several studies with $CO_2$ data only (e.g. Rayner et al., 2005; Peylin et al., 2016). However, compared to $CO_2$, the spatial coverage of COS surface observations is still too sparse to accurately constrain the GPP and therefore ORCHIDEE parameters (Ma et al., 2020). There is some hope that new satellite retrievals of COS column content, such as with the IASI (Infrared Atmospheric Sounder Interferometer) instrument, could have enough accuracy to better constrain the surface fluxes (Serio et al., 2020).

**5 Conclusions and Outlooks**

We have implemented inside the ORCHIDEE land surface model the mechanistic model of Berry et al. (2013) for COS uptake by the continental vegetation. Modelled COS fluxes were compared at site scale against measurements at the Harvard temperate deciduous broadleaf forest (USA) and at the Hyytiälä Scots pine forest (Finland), yielding relative RMSDs of around 40% at both diel and seasonal scales. We found that the mechanistic model yields a lower and thus more limiting internal conductance as compared to former works (Seibt et al., 2010; Wehr et al.,

2017). The next step is to perform a sensitivity analysis (Morris, 1991; Sobol, 2001) and to optimize the most sensitive parameters related to the modelled fluxes and conductances, to get a better agreement with observations. Our global estimate of COS uptake by continental vegetation of -756 Gg S yr[-1] is in the lower range of former studies. An important finding is that the LRU computed from monthly values of the COS and GPP fluxes yield

values lower than monthly means of high-frequency LRU values. This has consequences for atmospheric studies where COS concentrations integrate influences from fluxes at large spatial and temporal scales.

Using appropriate LRU values, we transported the monthly mean COS fluxes from the mechanistic and LRU approaches using the LMDz6 model. The evaluation of the modelled COS atmospheric concentrations against observations at stations of the NOAA network yields comparable results for both approaches.

As a general conclusion and for the moment, we can say that the mechanistic model is particularly valuable when studying small time or spatial scales using COS fluxes, while for global analyses using COS concentrations, both the mechanistic and LRU approaches give similar results. The fact that the global COS budget has so many components with a large uncertainty (Whelan et al., 2018) limits the use of COS concentrations as a constraint for GPP in land surface models on the global scale, for the present time.

A further development will be to refine the estimation for COS soil fluxes and to implement inside ORCHIDEE a mechanistic model for soil COS fluxes (Ogée et al., 2016; Sun et al., 2015). Having both the vegetation and soil contributions, we will also be able to assimilate ecosystem COS fluxes to optimize COS-related parameters such as $\alpha$ in the internal conductance formulation from the Berry et al. (2013) model for vegetation uptake, and those related to the stomatal conductance (Wehr et al., 2017; Berkelhammer et al., 2020). We will also later look at the

complementary constraints on GPP brought by COS and Solar-Induced Fluorescence, another GPP proxy (Bacour et al., 2019; Whelan et al., 2020).

**Appendices**

**Appendix A. Additional tables related to conductances**

**Table A1: Ratios of modelled boundary conductance to stomatal conductance and internal conductance, respectively,**
**at the two studied sites, computed over year 2012 at Harvard Forest and 2017 at Hyytiälä**

|  | Harvard Forest | | Hyytiälä | |
| --- | --- | --- | --- | --- |
| Ratio | Boundary to stomatal | Boundary to internal | Boundary to stomatal | Boundary to internal |
| Median | **28** | **69** | **46** | **228** |
| Minimum | 9 | 20 | 17 | 48 |
| Maximum | 188 | 1523 | 232 | 9304 |

**Table A2: Partial correlations linking stomatal and internal conductances to photosynthetically active radiation ($PAR$), air temperature ($T_{air}$), vapour pressure deficit ($VPD$), soil moisture ($SM$) and leaf area index (LAI), computed at a half-hourly time step over year 2012 at the Harvard Forest site and 2017 at the Hyytiälä site**

| Conductance | Site | $PAR$ | $T_{air}$ | $VPD$ | $SM$ | LAI |
| --- | --- | --- | --- | --- | --- | --- |
| $g_{s\_cos}$ | Harvard | **0.66** | 0.46 | -0.61 | -0.04 | 0.33 |
|  | Hyytiälä | **0.59** | 0.49 | -0.47 | -0.03 | 0.25 |
|  | Harvard | -0.06 | **0.68** | 0.30 | -0.27 | 0.15 |

| | | | | | |
|---|---|---|---|---|---|
| $g_{I\_COS}$ | Hyytiälä | -0.13 | **0.74** | 0.65 | 0.32 | 0.49 |


**Table A3: Minimum stomatal conductance to $CO_2$ (mmol m$^{-2}$ s$^{-1}$) for each PFT in Lombardozzi et al. (2017) and ORCHIDEE. No value is given for C4 crops in Lombardozzi et al. (2017).**

| | Mean minimum conductance in Lombardozzi et al. (2017) | Minimum conductance in ORCHIDEE |
|---|---|---|
| 1 - Bare soil | 0 | 0 |
| 2 - Tropical Broad-leaved Evergreen Forest | 90.488 | 6.25 |
| 3 - Tropical Broad-leaved Raingreen Forest | 109.744 | 6.25 |
| 4 - Temperate Needleleaf Evergreen Forest | 16.896 | 6.25 |
| 5 - Temperate Broad-leaved Evergreen Forest | 34.017 | 6.25 |
| 6 - Temperate Broad-leaved Summergreen Forest | 72.637 | 6.25 |
| 7 - Boreal Needleleaf Evergreen Forest | 8 | 6.25 |
| 8 - Boreal Broad-leaved Summergreen Forest | 50 | 6.25 |
| 9 - Boreal Needleleaf Summergreen Forest | 29 | 6.25 |
| 10 - C3 Grass | 157.988 | 6.25 |
| 11 - C4 Grass | 93.933 | 18.75 |
| 12 - C3 Agriculture | 60.629 | 6.25 |
| 13 - C4 Agriculture | x | 18.75 |

**Appendix B. Additional illustrations for results at site scale**


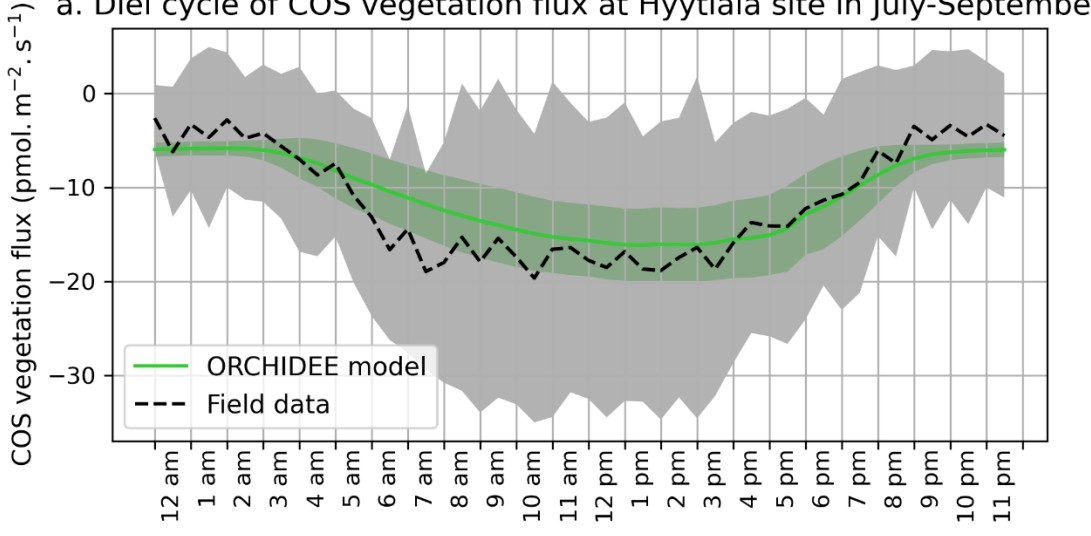

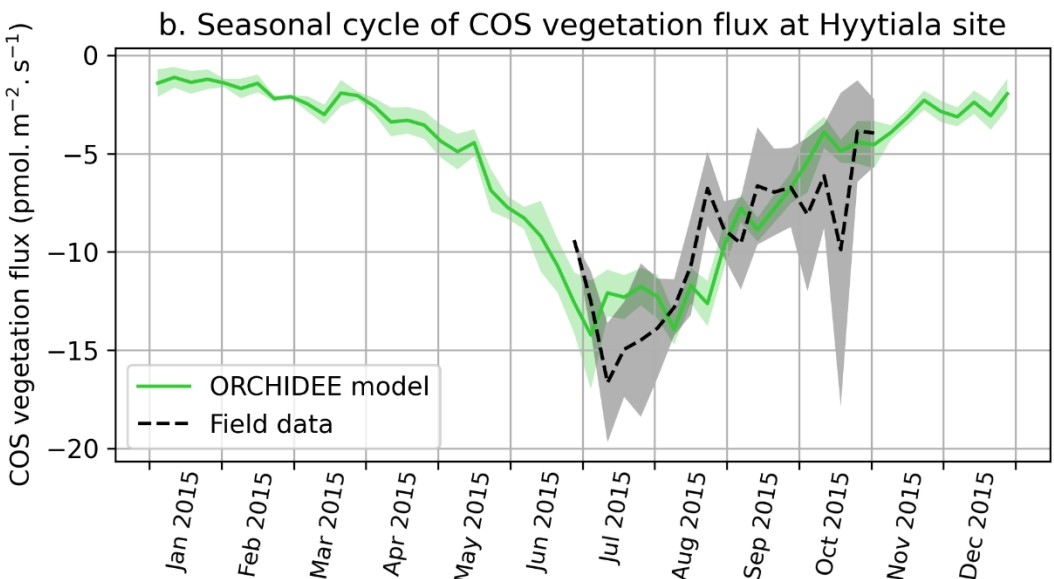

**Figure B1: a. Mean diel cycle of observed vegetation COS flux derived from ecosystem COS flux (Kohonen et al., 2020) and soil COS flux (Sun et al., 2018a), and modelled COS vegetation flux in July-September 2015, at Hyytälä, using an atmospheric convention where an uptake of COS by the ecosystem is negative. The shaded areas above and below each curve represent one standard-deviation of the considered half-hourly values over the July-September period. b. Mean seasonal cycle of simulated and observed weekly average vegetation COS flux in 2015, at Hyytälä. The shaded areas above and below each curve represent one standard-deviation of the daily means within the considered week.**


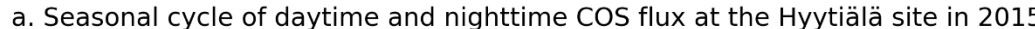

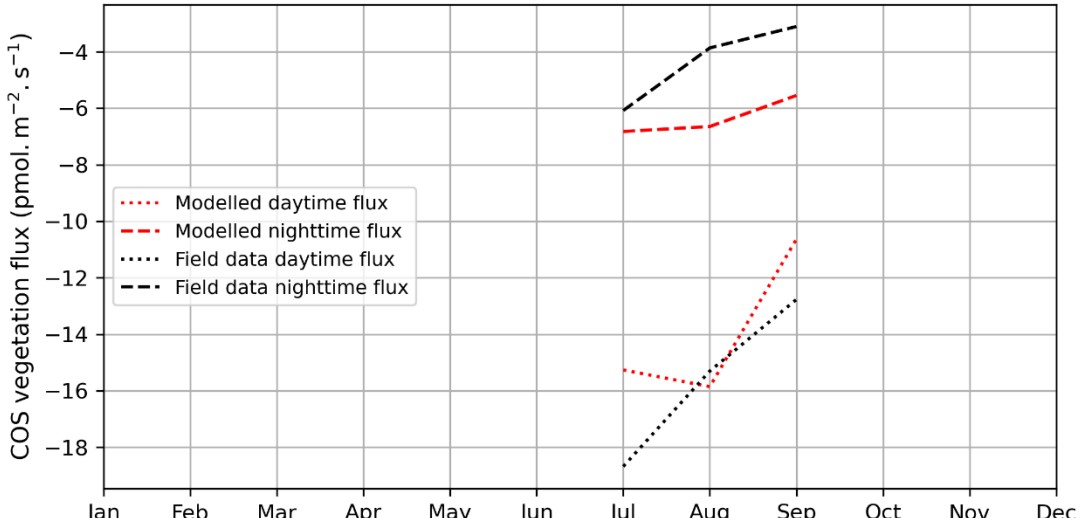

a. Seasonal cycle of daytime and nighttime COS flux at the Hyytiälä site in 2015

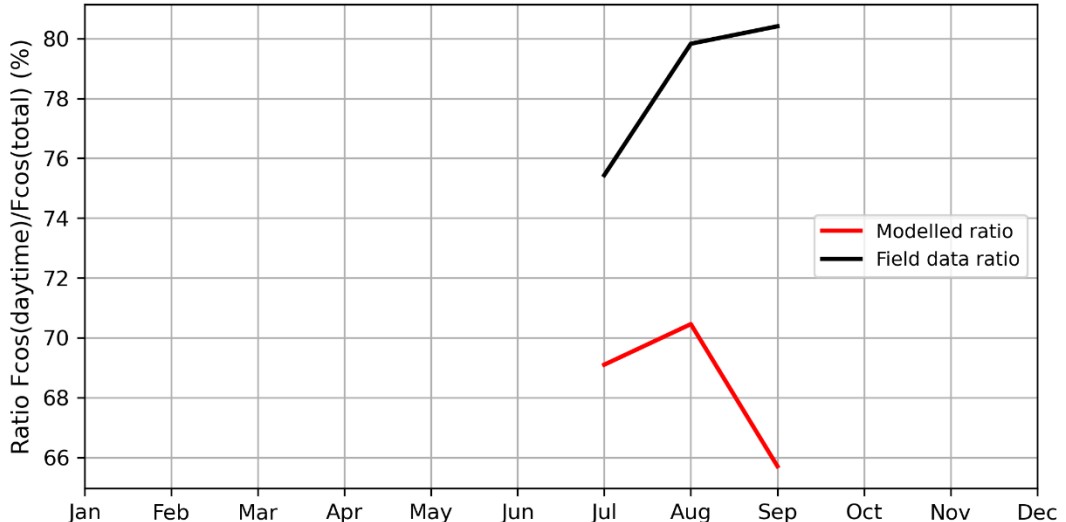

b. Seasonal cycle of percentage of daytime to total flux at the Hyytälä site in 2015

**Figure B2: a. Seasonal cycle of: a. daytime (dotted curve) and nighttime (dashed curve) for observed (black) and modelled (red) vegetation COS fluxes, b. percentage of the daytime to the total flux (solid curve), at the Hyytiälä site in 2015**

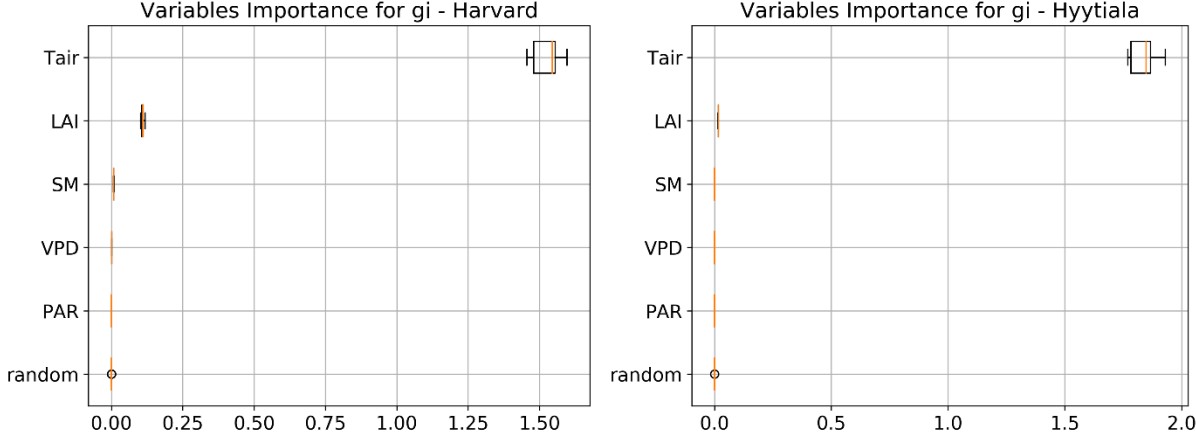

**Figure B3: Variables importance computed using Random Forests for the internal conductance (gi) at the Harvard Forest site in 2012 (left) and at the Hyytiälä site in 2017 (right). The considered predictors are air temperature (Tair),**

 **leaf area index (LAI), soil moisture (SM), vapour pressure deficit (VPD) and photosynthetically active radiation (PAR). A random predictor is added to prevent over-fitting.**

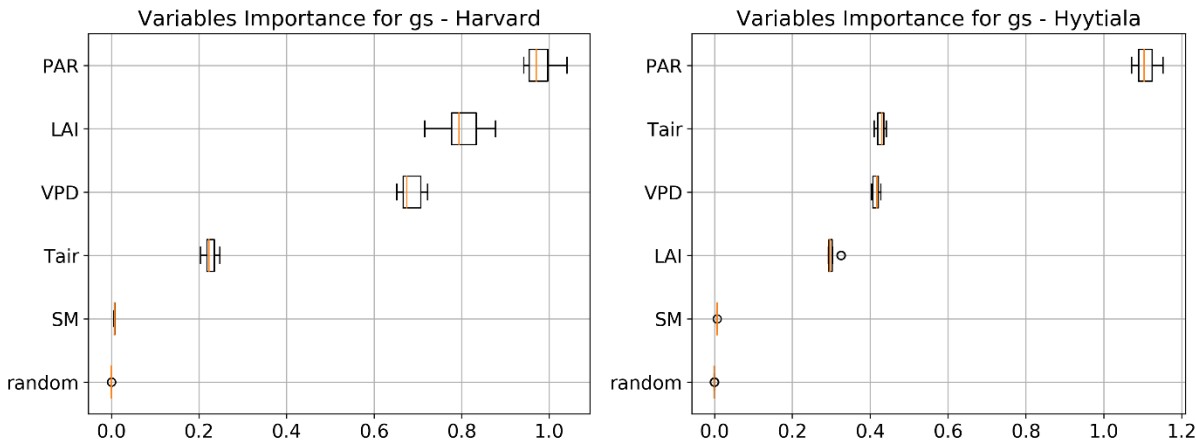

**Figure B4: Same as B3 for the stomatal conductance (gs)**

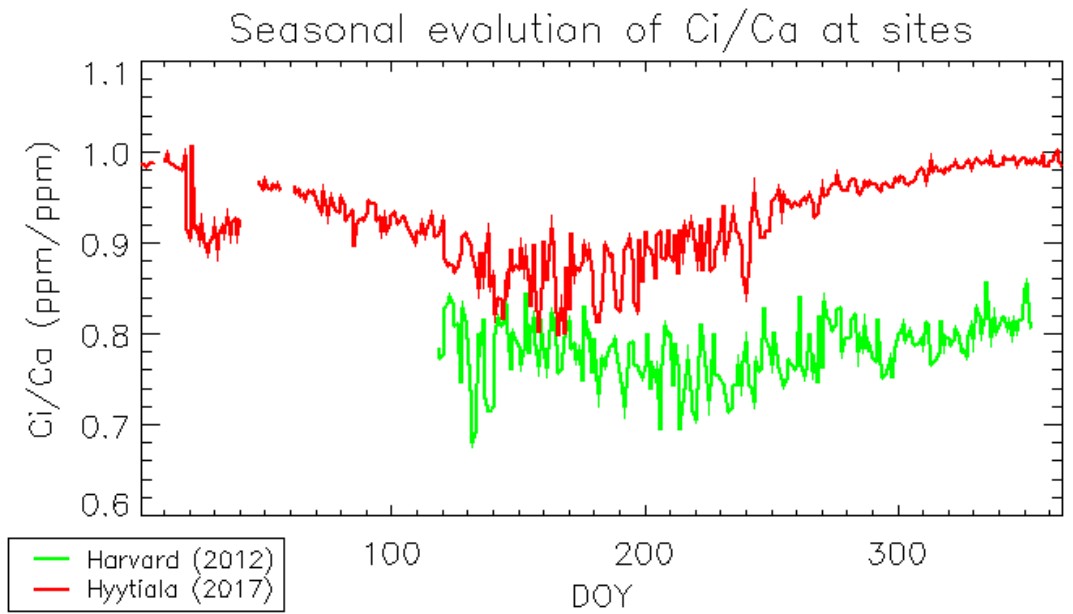

 **Figure B5: Seasonal evolution of the simulated $C_i$ to $C_a$ ratio at the Harvard Forest site in 2012 (green curve) and the Hyytiälä site in 2017 (red curve)**

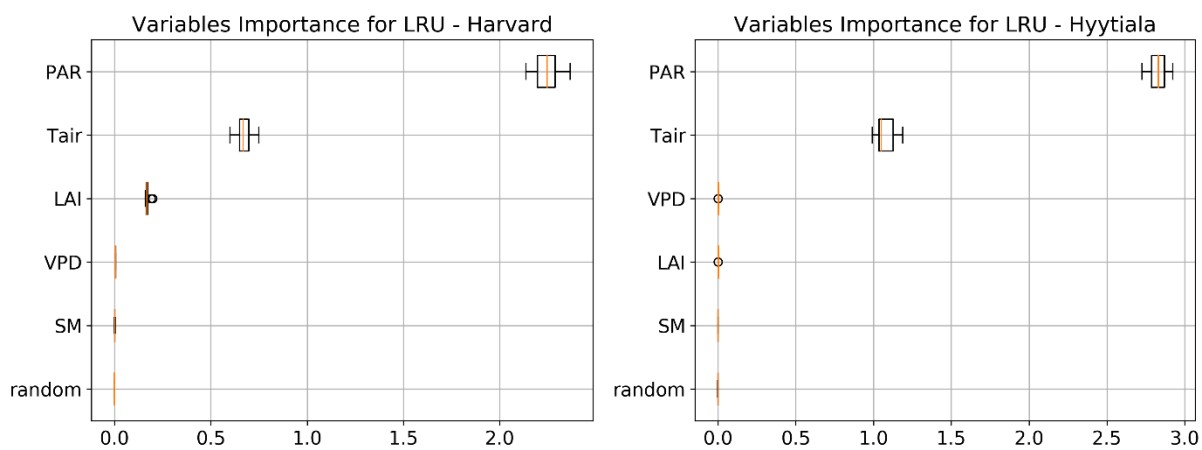

**Figure B6: Same as B3 for the leaf relative uptake (LRU)**

**Appendix C. Additional illustrations for results at global scale**

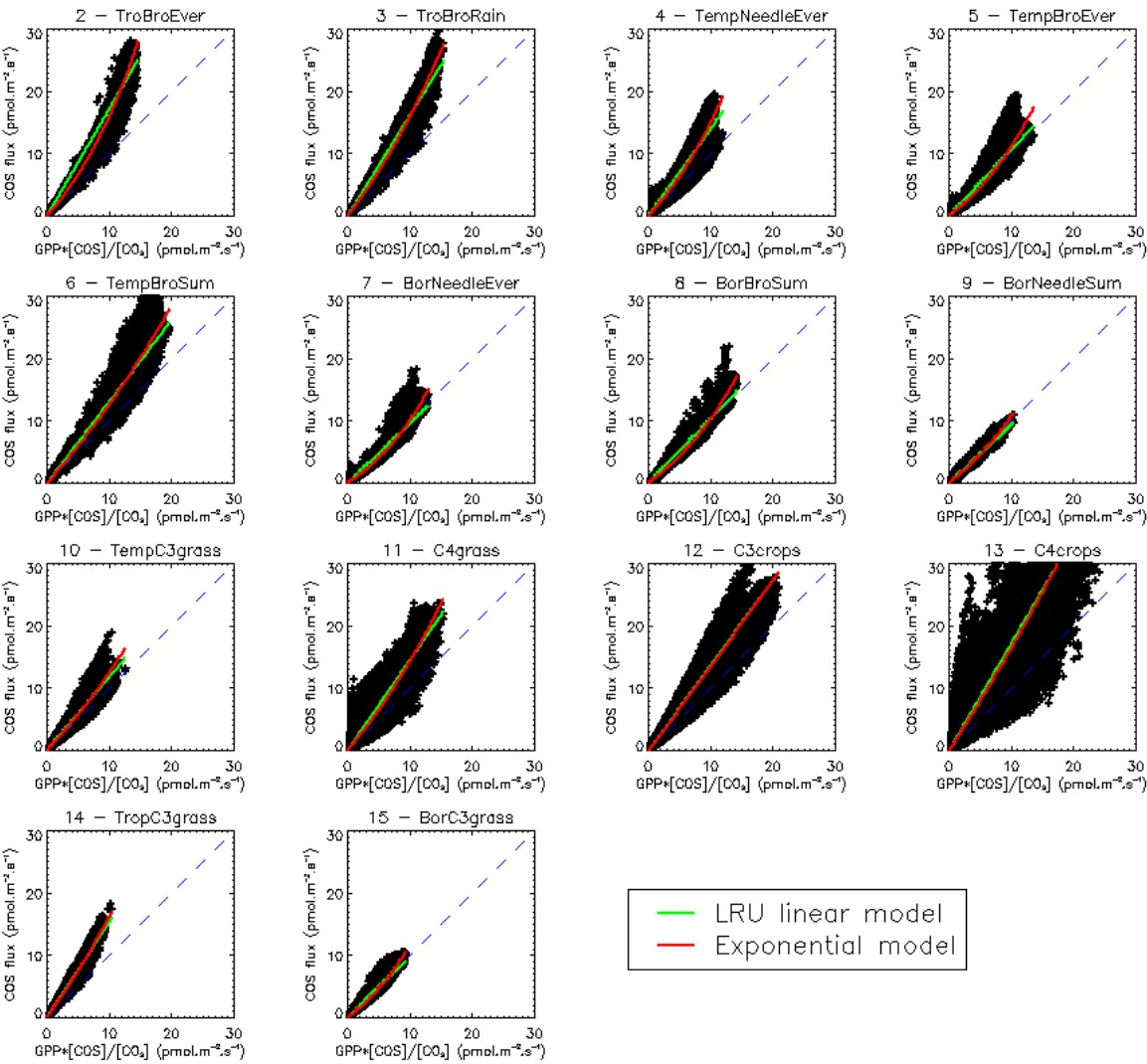

**Figure C1: Scatterplots of COS fluxes against GPP multiplied by the ratio of COS to CO2 concentrations, using a climatology of monthly fluxes over the 2000-2009 period and yearly global averages for $CO_2$ concentrations and a fixed value of 500 ppt for the COS concentration. Each subplot represents one of the 14 vegetated PFTs used in ORCHIDEE. The LRU model in green represents the linear regression, while the exponential model (see text) is represented in red. The blue dashed lines show the 1:1 line.**

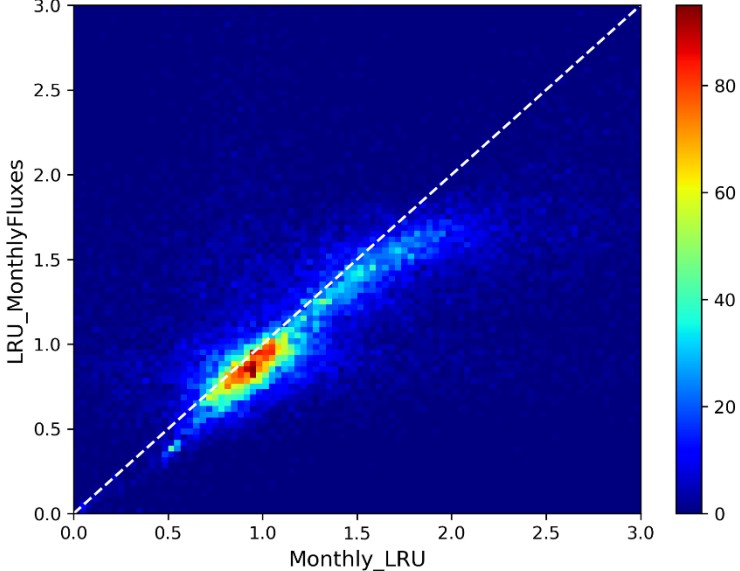

**Figure C2: Bi-dimensionnal histogram of LRU values computed from a climatology of monthly mean fluxes (LRU_MonthlyFluxes) against a climatology of monthly means of LRU computed from original half-hourly values (Monthly_LRU). The colorbar indicates the number of occurrence per bin of 0.1x0.1 size. The white dashed line represents the first bisector.**

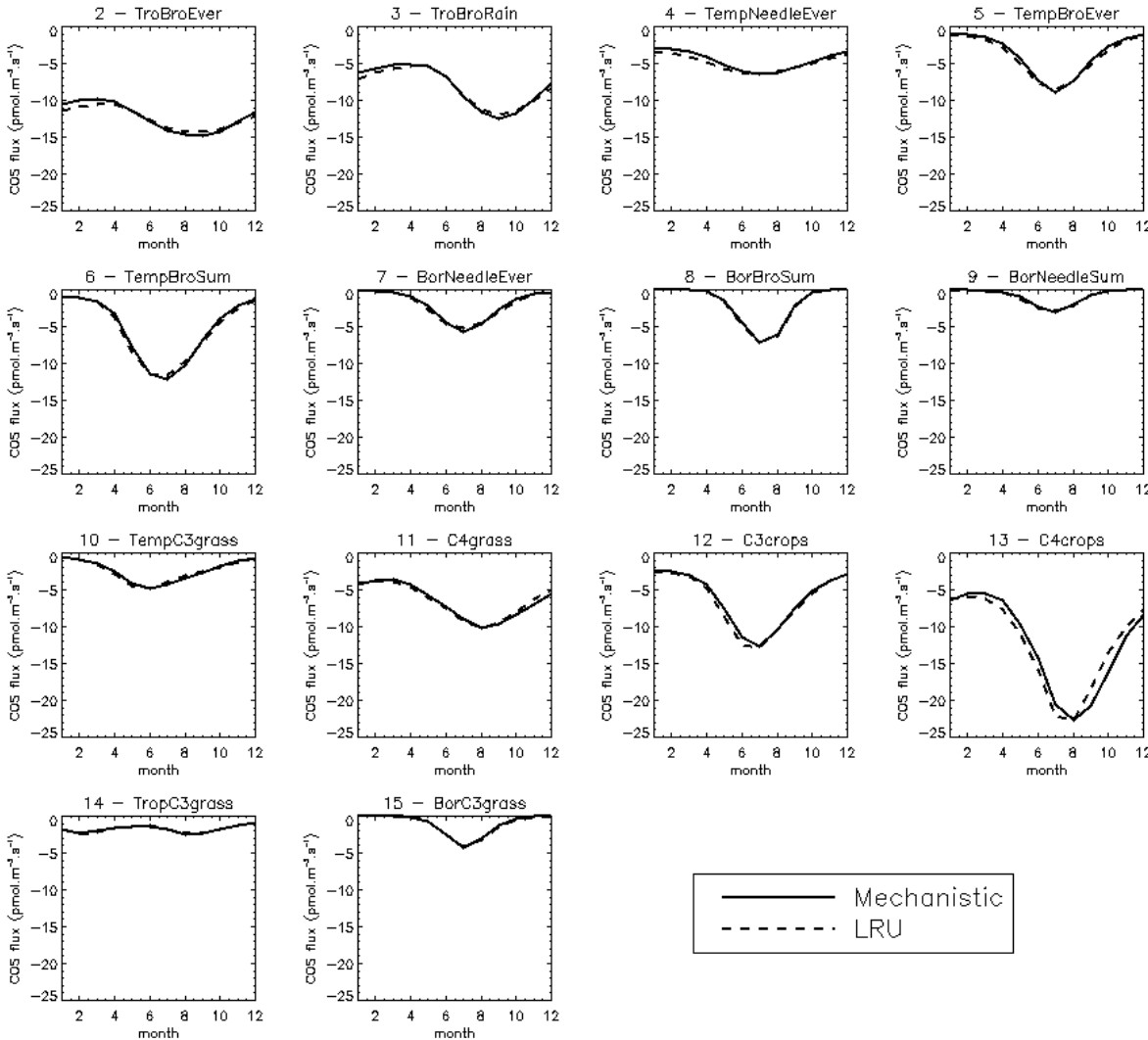

**Figure C3: Mean seasonal cycle (monthly means) of COS for each PFT over the Northern hemisphere for the 2000-2009 period. The solid line represents the mechanistic model, while the dashed line represents the optimal LRU approach.**

## Appendix D. Sensitivity tests for the modelling of atmospheric COS concentrations

### D1. Simulating COS atmospheric concentration at stations: impact of the oceanic emissions

We performed the same experiment as in Sect. 3.4, except that the oceanic fluxes (direct and indirect) are here from Lennartz et al. (2017). In our case, the oceanic emissions (in particular direct oceanic emissions) have more impact than the LRU on the seasonality at surface sites from the NOAA network.

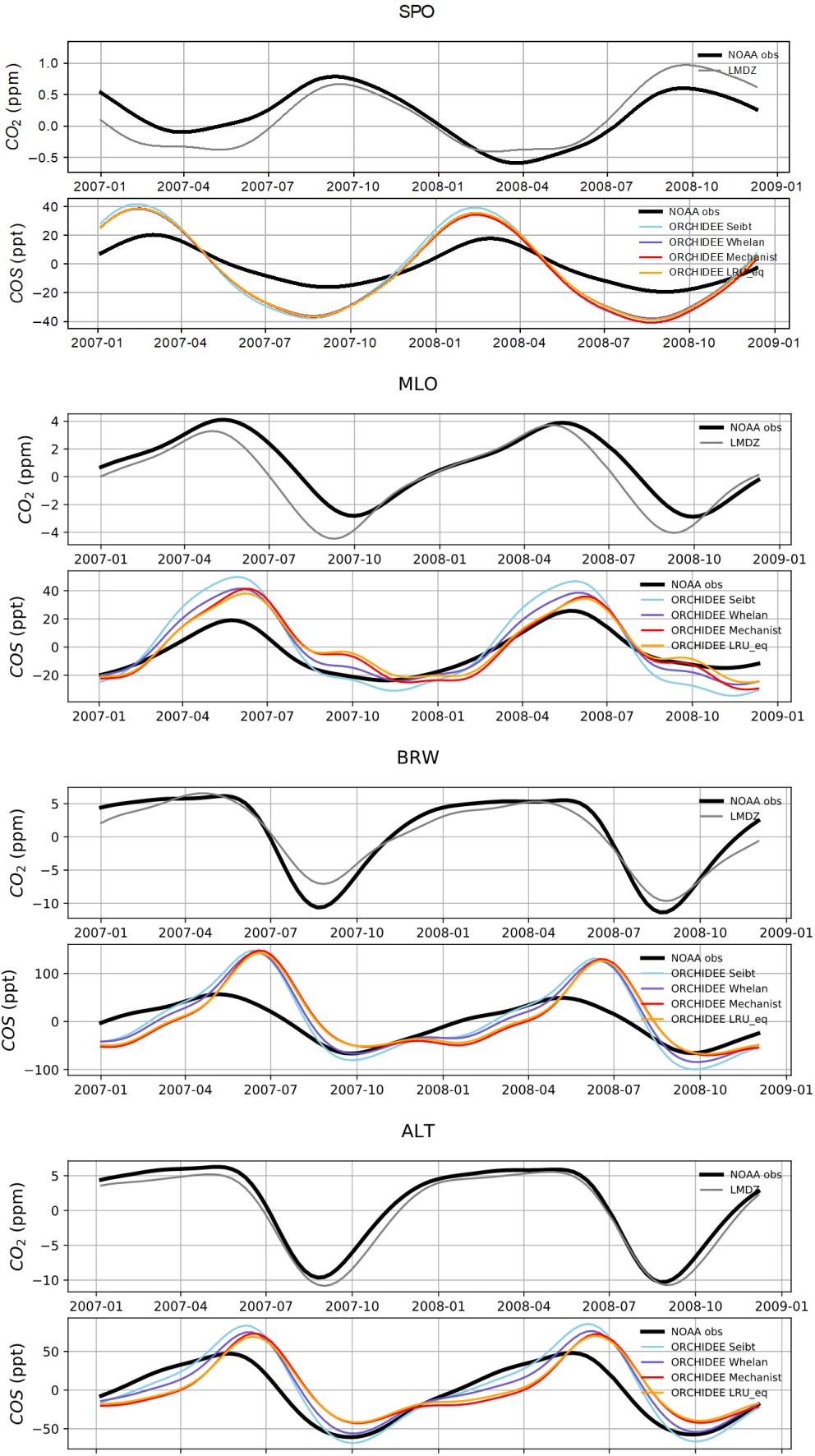

**Figure D1: Detrended temporal evolutions of simulated and observed CO₂ and COS concentrations at four selected sites, for the mechanistic (ORCHIDEE Mechanist) and LRU approaches (ORCHIDEE Seibt, ORCHIDEE Whelan, ORCHIDEE LRU_Opt), simulated with LMDz6 transport between 2007 and 2009. The ORCHIDEE LRU_Opt line (orange) corresponds to the concentrations simulated using the optimal LRU values derived from the mechanistic model. The curves have been detrended beforehand and filtered to remove the synoptic variability (see Sect. 2.2.4).**

**Table D1: Prescribed COS surface fluxes used as model input. Mean magnitudes of different types of fluxes are given for the period 2000-2009**

| Type of COS flux | Temporal resolution | Total (Gg S yr⁻¹) | Data Source |
|---|---|---|---|
| Anthropogenic | Monthly, interannual | 337.3 | Zumkehr et al. (2018) |
| Biomass burning | Monthly, interannual | 56.3 | Stinecipher et al. (2019) |
| Soil | Monthly, climatological | -409.0 | Launois et al. (2015b) |
| Ocean | Monthly, climatological | 344.0 | Lennartz et al. (2017) |
| Vegetation uptake | Monthly, interannual | | This work, including mechanistic and LRU approaches (Seibt et al., 2010; Whelan et al., 2018). |

**D2. DMS sensitivity study**

We further tested the impact of the indirect COS fluxes through DMS on the simulated concentrations at NOAA sites. To do that, we compared the atmospheric concentrations given with and without prescribing indirect oceanic fluxes through DMS using the Launois et al. (2015a) oceanic fluxes. In our case, the removal of the DMS oceanic emissions decreases the seasonal amplitude at SPO and CGO but have very few impacts at other sites. We also performed the same experiment using the Sinikka et al. (2017) fluxes and reported no impact of DMS indirect fluxes on simulated concentrations at NOAA sites.

**Table D2: Normalized standard deviations (NSDs) of the simulated concentrations by the observed concentrations. Within brackets are the Pearson correlation coefficients (R) between simulated and observed COS concentrations for the mechanistic approach including the DMS or not, calculated between 2004 and 2009 at 10 NOAA stations.**

| | SPO | CGO | SMO | KUM | MLO | NWR | LEF | MHD | BRW | ALT |
|---|---|---|---|---|---|---|---|---|---|---|
| ORCHIDEE Mechanist (DMS) | 1.10 (0.97) | 1.01 (0.97) | 0.35 (0.4) | 0.90 (0.95) | 1.05 (0.92) | 1.26 (0.63) | 1.34 (0.94) | 1.09 (0.85) | 0.69 (0.91) | 0.64 (0.96) |
| ORCHIDEE Mechanist (Without DMS) | *0.74* (0.91) | *0.53* (0.94) | 0.38 (0.20) | 0.90 (0.95) | 1.04 (0.91) | 1.31 (0.64) | 1.40 (0.94) | 0.93 (0.94) | 0.74 (0.90) | 0.65 (0.96) |

**Code availability**

The ORCHIDE model is available on request to the authors.

**Author contribution**

FM and PP devised the research. CA and FM coded the ORCHIDEE developments and made the simulations. MR and PP dealt with the transport model. LMJK and KMK provided the Hyytiälä data. RC and RW provided the Harvard Forest data. JEC, SB, SAM, NR, US, YPS, NV, MEW were consulted on their respective expertise. FM, CA and MR analysed the results and wrote the first draft. All authors contributed to the manuscript.

**Competing interests**

The authors declare that they have no conflict of interest.

**Acknowledgments**

The authors thank the reviewers for their constructive and useful comments which helped to further improve this study. The LSCE group thanks the administrative and IT teams for managing the recruitment of CA, and providing the necessary facilities and tools to run the ORCHIDEE model and analyse the outputs. CA, FM and PP have been

mainly supported by the European Commission, Horizon 2020 Framework Programme, 4C (grant no. 821003) and to a smaller extent VERIFY (grant no. 776810). MR was funded by the $CO_2$ Human Emissions (CHE) project which received funding from the European Union's Horizon 2020 research and innovation programme under grant agreement no. 776186. KMK thanks The Vilho, Yrjö and Kalle Väisälä foundation and ICOS-FINLAND (319871) for their financial support. LMJK received funding from the ERC project COS-OCS under grant nr 742798.

Operation of the US-Ha1 site is supported by the AmeriFlux Management Project with funding by the U.S. Department of Energy's Office of Science under Contract No. DE-AC02-05CH11231, and additionally is a part of the Harvard Forest LTER site supported by the National Science Foundation (DEB-1237491).

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
