# Peer review of "Carbonyl Sulfide: Comparing a Mechanistic Representation of the Vegetation Uptake in a Land Surface Model and the Leaf Relative Uptake Approach"

_Biogeosciences, 2020_

## Referee Comment (RC1) · Georg Wohlfahrt (Referee) · 12 Nov 2020

Maignan and co-authors present a simulation study using the ORCHIDEE model comparing a mechanistic formulation of the leaf COS uptake with the established LRU approach. They do so by confronting site-scale simulations with eddy covariance flux measurements, but also simulations at global scale and then compare against measurements of COS concentrations at globally distributed atmospheric measurement sites. While they find that at global and monthly scale, the difference between the two approaches is a minor issue compared to other processes in the global COS cycle,

they provide interesting evidence into the spatio-temporal variability of LRU and the performance of the mechanistic model in reproducing observed physiological behavior at two field sites. The paper is mostly well written (where it is not this can be left to the final copy-editing) and certainly within the scope of BG and presents important progress beyond previous work using similar approaches. I am really intrigued by Fig. 8, which clearly demonstrates that there is substantial variability in LRU which we need to better understand and represent if we are to further develop COS into a defensible proxy for GPP. Most of my detailed comments are probably minor and should be straightforward to deal with.

The one comment that may require more thought and in fact some additional work by the authors relates to text around Fig. 2 and Table 4, where the authors attempt to disentangle the drivers of simulated changes in gs_cos and gi_cos. Given the high non-linearity of the underlying equations I find Fig. 2 to be highly uninformative and wonder how well the simple partial regression analysis presented in Table 4 is able to capture the underlying highly non-linear interactions. In my view this analysis, which is another of the highlights of this study, would require a different approach, something like a Global Sensitivity Analysis or similar (e.g. Pianosi et al. 2015), in order to properly account for the many highly non-linear interactions. Given that ORCHIDEE is quite a complex beast of a model, not sure how feasible such a GSA is given the need to execute a large number of model evaluations.

One more comment, or maybe rather a suggestion, regards the LRU concept, which has to be understood as the simplest possible, but still process-based, closure possibility in order to use measurements of the COS flux to estimate GPP. In choosing some LRU value, implicit assumptions are made about the ratio of conductances, which is dealt with in this paper, but also about the Ci/Ca ratio (Seibt et al. 2010, Wohlfahrt et al. 2012). This has always been a major criticism of the LRU concept, as it amounts to ignoring the demand side of the CO2 diffusion equation. I am thus wondering whether the authors could make a useful contribution here by expanding their analysis to look

into the spatio-temporal Ci/Ca ratio, given that Ci must be calculated by their mechanistic model, and confront these with our understanding of how Ci/Ca varies across the globe and seasonally.

Detailed comments: l. 82-83: fluxes and mixing ratios in $\mu$mol/m2s and $\mu$mol/mol, respectively, result in tiny numbers – why not use units of pmol/m2s and pmol/mol which are typically used in practice? l. 85: ... if Fcos, [CO2]a and [COS]a are available ... l. 85-96: this is all well-known – the real issue in my view is that we are seeing a large spread in LRU values even after normalizing for PAR that we cannot explain; also note that Yang et al. (2018) clearly show that low-light periods, when LRU deviates most, typically contribute little to the daily and longer-term integrated GPP l. 95: and possibly other factors we do not yet understand l. 97-100: this is essentially a technical objective – can the authors come up with an objective that aims at providing an advancement in scientific understanding and/or even some hypotheses of what they believe will result from adopting the mechanistic as opposed to LRU approach? l. 139: "... is generally not produced by plants (but see Gimeno et al. 2017)." l. 140: suggest to remove "to heat and " as this sounds as if the boundary layer conductance to heat would affect the flux of COS l. 164: the seminal paper to nighttime transpiration and residual conductance in my view is Dawson et al. (2007) l. 173: does the model then also calculate H2O gas exchange at night or only COS? l. 195: What does "Vegetation COS flux direct or derived measurements ..." mean? Suggest to reformulate Eq. (4) and (5): what about some statistics which allow evaluating systematic biases? l. 226: this is the first citation of a table and thus table number should be 1 l. 310: wouldn't it make sense to compare modelled and measured canopy-scale COS fluxes before embarking on a detailed analysis of the underlying processes, i.e. move sections 3.1.2.1 and 3.1.2.2 here? Fig. 1: what is the area reference for the units – m2 leaf area or m2 ground area? l. 336-340: alpha values depend on Vmax, which I suppose changes with phenology? If so, would it make sense to also investigate Vmax? l. 346-347: I would turn around the argumentation here, i.e. say that due to the way of how gs is simulated according to Yin and Struik (2009) there is a linear relationship with A ... l.

354-355: but isn't Vmax strongly driven by phenology as well and wouldn't this be the main seasonal driver? Fig. 4: it looks like there is a data gap in May 2013, but the dashed line is not broken and instead interpolates through the gap, compared to 2012 when the dashed line stops during all gaps l. 395: there also appear to be issues in terms of phenology, e.g. in April 2013 l. 396: can the authors estimate the magnitude of the noise (I presume they refer to the random variability of EC flux measurements) and whether this noise could be responsible for the observed variability? l. 403: I would replace "diurnal" with "daytime", which I believe better fits with what the authors mean Fig. 6: what do the model simulations of LRU refer to – the canopy scale? If so what do the PAR values refer to – the PAR incident at the top of the canopy? If so then I would say that there is a scale issue affecting the comparison to the branch-chamber measurements, as the canopy LRU is the integral over all leaves, both sunlit and shaded, which experience very different PAR values; having said this, I however would expect canopy simulated LRU values to lie above the measurements at low PAR, as when PAR incident on the top of the canopy is say 200 $\mu$mol/m2s, a large fraction of the leaves will be in the shade and experience much lower PAR values and should thus actually have a higher LRU; in any case the authors should critically reflect on whether any scale issues affect the interpretation of the comparison between measured branch-chamber and simulated, presumably canopy-scale, LRUs Fig. 7 and 9: appropriate x- and y-axis scale and text needed Table 5: can the authors add the time frame to the table to which the different studies refer to? l. 459: and you use the same parameterization of COS uptake?! Fig. 7: can the authors avoid the figure legend to overlap with the plotted lines? l. 546, 549: isn't Barrow in Alaska/USA? l. 561: to what does the citation Whelan and Seibt refer to? l. 598-599: agreed, but it should be acknowledged that at present the spatial coverage of biomes is very poor (some biomes not having been sampled at all, for others n=1) and also temporal coverage is poor, with the exception of Hyytiälä and Harvard forest most COS flux measurements being confined to less than a year, leaving a big question mark with regard to inter-annual variability l. 603-606: unclear what this sentence tries to say l. 633-634: can you elaborate on which

data would be required to that end? l. 645: nighttime instead of nitghttime l. 666: here the authors might refer to some recently published experimental canopy-scale LRU estimates, such as Spielmann et al. (2019, 2020) who provide estimates for deciduous forest, savanna, C3 crop and C3 grassland l. 669: another issue, mentioned first in Wohlfahrt et al. (2012), is that leaf chambers are typically vigorously ventilated and LRU values inferred from these are thus implicit with the correspondingly large boundary layer conductance; under real-world conditions boundary layer conductances vary with wind speed (temporally but also with depth of the canopy) and transport across the boundary layer may not necessarily be dominated by forced convection as in leaf chambers; that is to say that for this reason leaf-chamber LRUs may not be easily transferable to real-world conditions l. 678: in this section you might cite Spielmann et al. (2020) who show, with a simple linear perturbation analysis, that variability in [COS]a does in fact affect LRU

---

## Referee Comment (RC2) · Anonymous Referee #2 · 17 Dec 2020

Maignan et al., develop a parameterization for including OCS uptake within a well-known land surface model (ORCHIDEE ). Since OCS has been proposed as a proxy for GPP (currently not observable at large scales), this work is an important step towards estimating global GPP. The authors do a very thorough job of combing the OCS literature for published values of leaf relative uptake (LRU) and validating modeled uptake at two temperate sites where OCS flux is measured. I think this work should eventually be published in Biogeosciences. However I am currently a little confused with regard to how the paper is framed. The authors pose their formulation of OCS flux

using a conductance based approach against the well-known LRU based relationship between FOCS and GPP. The LRU approach is also based on conductance, with simple assumptions regarding the relative role of mesophyll and stomatal conductances (Seibt et al., 2010). Thus the differences in flux resulting between the two approaches are not surprising, and don't really manifest themselves once transported and compared against OCS flask measurements from NOAA. This is an interesting finding, but I think the authors must focus on a more scientific question.

The other main comment I have is about modeled gs,ocs and gi,ocs. I agree with the authors that the role of gi is important and often ignored, but I am somewhat skeptical of the large diurnal variation in modeled gi, which is related to temperature. As I understand, gi is estimated from Vmax of Rubisco, but Rubisco response to temperature is not thought to be that large, particularly at temperatures observed at the temperate NH sites (see Sage and Kubien 2007). Moreover, while estimates of diurnal variability of mesophyll conductance hasn't been reported much, a recent study showed that the diurnal variability in gm (which is similar to gi in this study) is much smaller than gs (Strangl et al., 2019; which is also a high latitude coniferous forest, so somewhat similar to Hyytiälä ). This could serve as a mechanism for plants to modify (increase) water use efficiency and therefore continue assimilating carbon even as gs declines due to high VPD commonly observed after mid-day (see also Buckley and Warren, 2014). See also recent work by Gimeno et al., (2020). Thus, I am quite surprised that at Hyytiälä it seems like gas exchange is most often limited by gi and not gs. I believe some sensitivity analyses could be done with regard to the formulation of gi. The temperature dependance of gi is also not uniform across plant species (von Caemmerer and Evans, 2015), with obvious implications for the global formulations presented here. I think a more in-depth discussion of the implications of gas exchange most limited by gi should be added, replacing the current and mostly qualitative discussion currently.

One way to understand the relative roles of gs, gi, and the differences between the mechanistic model and the LRU approach is to look at values of intercellular and chloroplast CO2 concentrations (ci and cc respectively). These should be standard outputs of the model. ORCHIDEE ci and cc could be compared with "inferred" ci and cc using modeled A (GPP) and gi,cos and gs,cos following Seibt et al., 2010. It would be interesting to see if those differences can explain the difference in FOCS flux obtained from those two approaches (for e.g., in Figure 9).

In general, I find the manuscript too long and perhaps some figures (e.g. Figure 2) can be removed. Similarly, perhaps figures 3 and 4 can be combined. I greatly appreciate all the work that has gone in to this, but perhaps the authors ought to split this in two papers. One that describes the modeling framework in ORCHIDEE and another that focuses on the transport modeling? This would allow the authors to delve more deeply in the important findings such as Fig 8. Similarly, I can imagine that Fig 11 discussion can be greatly expanded upon.

I find it a little odd that LMDZ doesn't match the seasonal cycle of CO2 at MLO. Is. this a known issue of the transport model? In summary, I am not entirely convinced of the transport analyses since inferences could be flawed due to erroneous transport. Some quantification of transport error/uncertainty should be presented (perhaps using withheld/independent observations?).

The writing in the results section is often very qualitative and not informative (e.g., line 329: "The conductances drop in the afternoon to reach minimum values at night"). I find most of the discussion unnecessary. It seems like a re-hash of the methodology and results section (e.g., Sec. 4.1.1. and 4.1.2), with inferences that I don't think are always supported by the results (for instance lines 653-656. This is an example where the writing could be improved to make a very compelling argument about the use of OCS flux). Some of the work suggested (e.g in Sec. 4.1.3.) is well within the purview of this study and can be performed (ie., some sensitivity analyses of min gs and gi, and impact on simulated flux.).

The writing overall really needs to be tightened and the conclusions seem a bit weak.

This could be improved with better framing. Thus I believe that with a much clearer presentation of figures and text, this could be a much more compelling manuscript.

Comments about figures and tables: Figure1: Conductance seems to be highest at 12 am ie., midnight? Growing season midday at Harvard Forest is shown to be limited by gi but Wehr et al (2017) measurements show that gs is the limiting conductance to OCS transfer. How do these numbers compare with Wehr and Kooijmans measurements (of gi)? You could also add total conductance to OCS (the quantity multiplied to [OCS]a in eq 3). In general, since you don't seem to be explaining month by month variations, maybe compress these to show 3 month means (e.g. JJA, SON etc)? Currently, it is impossible to see in detail the diurnal variations, specially when one is trying to discern at what times of day gi > gs and vice versa.

Figure 2. I believe this figure should be replaced by scatterplots or a table listing correlations between gs, gi and aux variables (which already exists as Table 4). Why would gi be expected to scale with soil moisture? It is strange that PAR is highest in May at Hyytiälä.

Table 4: gs is more related to Tair than to VPD. Stomatal conductance has been shown to be related to VPD (See for e.g., Oren et al., 1999). This would be worth examining.

Figure 4. Doesn't seem that Harvard Forest 2013 observed fluxes match simulations all that well. For instance, fluxes seem to peak in May-June and Aug-Sep in 2013, but peak in July in the model. There is a mention of "noise" in EC based measurements in the text but these should be quantified or at least described.

Figure 5. It would be easier to view this figure as two separate panels instead of one plot with two y axes.

Minor Comments: Note, I stopped providing minor comments because of the length of the manuscript.

Line 70: remove "then" in COS is then hydrolyses

Line 77: mention that soils can be a sink or a source, and cite appropriate studies (cite appropriate studies e.g., Maseyk et al, 2014; Berkelahmmer et al, 2014; Kitz et al., 2020). Also mention the role of epiphytes in the OCS budget (and cite Kuhn et al., 2000; Gimeno et al., 2017 and Rastogi et al, 2018) .

Lines 78-85: Add some discussion about possible scaling issues for LRU from leaf to canopy and ecoregion scales. I think some of this framework can be found in Wohlfart et al., 2012

Line 90: summer is likely only valid for ecosystems that exhibit seasonality. Explain diurnal and seasonal variability in LRU.

Line 95: What do you mean by 'time'?

Line 112-13: remove these lines.

Line 121: Briefly describe improvements in the Farquhar model.

Lines 208-210: Provide scientific names for these species at Harvard Forest.

Lines 218-220: Awkward phrasing. Also, please elaborate what you mean by "when possible".

Line 339: change 'air surface temperature' to 'air temperature'.

Line 339: Very minor comment: 'modelling' and 'vapor'. The first is a "British" spelling and the second is "American". Please pick one and be consistent throughout.

Line 348: This isn't true based on Fig 2 as PAR peaks in May but gs in June at Hyytiälä.

Lines 356-358: Needs citations.

References:

Berkelhammer, M., Asaf, D., Still, C., Montzka, S., Noone, D., Gupta, M., Provencal, R., Chen, H. and Yakir, D., 2014. Constraining surface carbon fluxes using in situ measurements of carbonyl sulfide and carbon dioxide. Global Biogeochemical Cycles,

28(2), pp.161-179.

Buckley, T.N. and Warren, C.R., 2014. The role of mesophyll conductance in the economics of nitrogen and water use in photosynthesis. Photosynthesis research, 119(1-2), pp.77-88.

Gimeno, T.E., Campany, C.E., Drake, J.E., Barton, C.V., Tjoelker, M.G., Ubierna, N. and Marshall, J.D., 2020. Whole‐tree mesophyll conductance reconciles isotopic and gas‐exchange estimates of water‐use efficiency. New Phytologist.

Gimeno, T.E., Ogée, J., Royles, J., Gibon, Y., West, J.B., Burlett, R., Jones, S.P., Sauze, J., Wohl, S., Benard, C. and Genty, B., 2017. Bryophyte gas‐exchange dynamics along varying hydration status reveal a significant carbonyl sulphide (COS) sink in the dark and COS source in the light. New Phytologist, 215(3), pp.965-976.

Kitz, F., Spielmann, F.M., Hammerle, A., Kolle, O., Migliavacca, M., Moreno, G., Ibrom, A., Krasnov, D., Noe, S.M. and Wohlfahrt, G., 2020. Soil COS exchange: a comparison of three European ecosystems. Global Biogeochemical Cycles, 34(4), p.e2019GB006202.

Maseyk, K., Berry, J.A., Billesbach, D., Campbell, J.E., Torn, M.S., Zahniser, M. and Seibt, U., 2014. Sources and sinks of carbonyl sulfide in an agricultural field in the Southern Great Plains. Proceedings of the National Academy of Sciences, 111(25), pp.9064-9069.

Oren, R., Sperry, J.S., Katul, G.G., Pataki, D.E., Ewers, B.E., Phillips, N. and Schäfer, K.V.R., 1999. Survey and synthesis of intra‐and interspecific variation in stomatal sensitivity to vapour pressure deficit. Plant, Cell & Environment, 22(12), pp.1515-1526.

Rastogi, B., Berkelhammer, M., Wharton, S., Whelan, M.E., Itter, M.S., Leen, J.B., Gupta, M.X., Noone, D. and Still, C.J., 2018. Large uptake of atmospheric OCS observed at a moist old growth forest: Controls and implications for carbon cycle applications. Journal of Geophysical Research: Biogeosciences, 123(11), pp.3424-3438.

Sage, R.F. and Kubien, D.S., 2007. The temperature response of C3 and C4 photo-synthesis. Plant, cell & environment, 30(9), pp.1086-1106.

Seibt, U., Kesselmeier, J., Sandoval-Soto, L., Kuhn, U. and Berry, J.A., 2010. A kinetic analysis of leaf uptake of COS and its relation to transpiration, photosynthesis and carbon isotope fractionation. Biogeosciences, 7(1).

Stangl, Z.R., Tarvainen, L., Wallin, G., Ubierna, N., Räntfors, M. and Marshall, J.D., 2019. Diurnal variation in mesophyll conductance and its influence on modelled water-use efficiency in a mature boreal Pinus sylvestris stand. Photosynthesis research, 141(1), pp.53-63.

von Caemmerer Susanne and Evans, J.R., 2015. Temperature responses of mesophyll conductance differ greatly between species. Plant, Cell & Environment, 38(4), pp.629-637.

Wohlfahrt, G., Brilli, F., Hörtnagl, L., Xu, X., Bingemer, H., Hansel, A. and Loreto, F., 2012. Carbonyl sulfide (COS) as a tracer for canopy photosynthesis, transpiration and stomatal conductance: potential and limitations. Plant, cell & environment, 35(4), pp.657-667.

---

## Author Response (AR1)

*[1] Maignan and co-authors present a simulation study using the ORCHIDEE model comparing a mechanistic formulation of the leaf COS uptake with the established LRU approach. They do so by confronting site-scale simulations with eddy covariance flux measurements, but also simulations at global scale and then compare against measurements of COS concentrations at globally distributed atmospheric measurement sites. While they find that at global and monthly scale, the difference between the two approaches is a minor issue compared to other processes in the global COS cycle, they provide interesting evidence into the spatio-temporal variability of LRU and the performance of the mechanistic model in reproducing observed physiological behavior at two field sites. The paper is mostly well written (where it is not this can be left to the final copy-editing) and certainly within the scope of BG and presents important progress beyond previous work using similar approaches. I am really intrigued by Fig. 8, which clearly demonstrates that there is substantial variability in LRU which we need to better understand and represent if we are to further develop COS into a defensible proxy for GPP. Most of my detailed comments are probably minor and should be straightforward to deal with.*
Answer: We thank the Referee, Dr Wohlfahrt, for his globally positive assessment of our study.

*[2] The one comment that may require more thought and in fact some additional work by the authors relates to text around Fig. 2 and Table 4, where the authors attempt to disentangle the drivers of simulated changes in gs_cos and gi_cos. Given the high non-linearity of the underlying equations I find Fig. 2 to be highly uninformative and wonder how well the simple partial regression analysis presented in Table 4 is able to capture the underlying highly non-linear interactions. In my view this analysis, which is another of the highlights of this study, would require a different approach, something like a Global Sensitivity Analysis or similar (e.g. Pianosi et al. 2015), in order to properly account for the many highly non-linear interactions. Given that ORCHIDEE is quite a complex beast of a model, not sure how feasible such a GSA is given the need to execute a large number of model evaluations.*
Answer: We acknowledge the models for conductances are highly non-linear. As suggested by the Referee, we selected an alternative method to check whether the ranking of our predictors using partial correlations was robust. We used Random Forests (RF) to simulate ORCHIDEE results, and applied a permutation technique on these RF models to rank predictors (Breiman, 2001). RF are well adapted for non-linear problems, they were for example used to rank variables of importance for soil COS fluxes in Spielman et al. (2020).
We used half-hourly ORCHIDEE outputs associated to LRU values between 0 and 8 to train RF models for conductances at the two sites (Hyytiäla in 2017, and Harvard in 2012). As in the initial study with partial correlations, we considered the following predictors: air surface temperature ($T_{air}$), photosynthetically active radiation ($PAR$), vapor pressure deficit ($VPD$) and soil moisture ($SM$). As canopy conductances scale with leaf area index (LAI, see comments [18] and [20]), we added LAI as a predictor. A random predictor was also added to check that the variable importance was correctly estimated. All RF models have an accuracy of at least 96%.

[Figure]

**Figure R1.1: Variables importance computed using Random Forests for the internal conductance (gi) at the Harvard Forest site in 2012 (left) and at the Hyytiälä site in 2017 (right). The considered predictors are air temperature (Tair), leaf area index (LAI), soil moisture (SM), vapour pressure deficit (VPD) and photosynthetically active radiation (PAR). A random predictor is added to prevent over-fitting.**

[Figure]

**Figure R1.2: Same as R1.1 for the stomatal conductance (gs)**

For comparison we updated the former partial correlation study, adding the LAI predictor:

**Table R1.1: Partial correlations linking stomatal and internal conductances to photosynthetically active radiation (PAR), air temperature (Tair), vapour pressure deficit (VPD), soil moisture (SM) and leaf area index (LAI), computed at a half-hourly time step over year 2012 at the Harvard Forest site and 2017 at the Hyytiälä site**

| Conductance | Site | $PAR$ | $T_{air}$ | $VPD$ | $SM$ | LAI |
|---|---|---|---|---|---|---|
| $g_{S\_cos}$ | Harvard | **0.66** | 0.46 | -0.61 | -0.04 | 0.33 |
| | Hyytiälä | **0.59** | 0.49 | -0.47 | -0.03 | 0.25 |
| $g_{I\_cos}$ | Harvard | -0.06 | **0.68** | 0.30 | -0.27 | 0.15 |
| | Hyytiälä | -0.13 | **0.74** | 0.65 | 0.32 | 0.49 |

This section was thus considerably revised, Figure 2 was removed and the presentation of the results has been shortened as follows: "The ranking is different between the two methods (partial correlation vs RF), but they agree that at both sites the main driver for the internal conductance is the air temperature and the main driver for the stomatal conductance is PAR.

As expected, $g_{I\_cos}$ mainly depends on $T_{air}$. This is explained by the fact that $g_{I\_cos}$ is proportional to $V_{max}$, which represents the Rubisco activity for $CO_2$; $V_{max}$ is assumed to be a measure for the mesophyll diffusion and for the CA activity for COS, which are the components of the internal conductance (Berry et al., 2013). $V_{max}$ depends on $T_{air}$, considered here as a proxy of the leaf temperature (Yin and Struik, 2009). This strong link explains why $g_{I\_cos}$ is more limiting in winter, as $T_{air}$ is low with thus lower enzyme activities, and, as soon as $T_{air}$ rises in spring, $g_{I\_cos}$ becomes less limiting, especially at night. $PAR$ is the most important variable for the stomatal conductance at the two sites. Due to the way of how $g_{S\_cos}$ is simulated according to Yin and Struik (2009), there is a linear relationship with the $CO_2$ assimilation, which depends mainly on $PAR$."

We also added a similar study RF for LRU. The predictors rank similarly for the two sites. The main factors for simulated LRU at a half-hourly time step are: $PAR$, $T_{air}$ and LAI (Figure R1.3).

These new results have been added in the site scale results (section 3.1), the figures are provided in the Appendices.

[Figure]

**Figure R1.3: Same as R1.1 for the leaf relative uptake (LRU)**

When trying to apply this RF methodology at monthly and global scales, we could not obtain a satisfying accuracy for LRU whatever the predictors we tried. Given indeed the non-linearity of the problem, the LRU we computed using equation (1) based on the monthly fluxes (called LRU_MonthlyFluxes) are different from the monthly means of the LRU computed at a half-hourly time-step within ORCHIDEE (called Monthly_LRU), as can be seen on the bi-dimensional histogram below.

[Figure]

**Figure R1.4: Bi-dimensionnal histogram of LRU values computed from a climatology of monthly fluxes (LRU_MonthlyFluxes) against a climatology of monthly means of LRU computed from original half-hourly values (Monthly_LRU). The colorbar indicates the number of occurrence per bin of 0.1x0.1 size. The white dashed line represents the first bisector.**

The LRU from monthly fluxes tends to be lower than the monthly mean of the LRU computed at a half-hourly time step. The bias is -0.2 and the correlation is 0.67. This can also be illustrated with an updated version of Figure 8, where we added in blue the distribution of the monthly means of the LRU computed from half-hourly LRU values.

[Figure]

**Figure R1.5: Distributions of the LRU values computed from the mechanistic approach over the 2000-2009 period. Each subplot represents one of the 14 vegetated PFTs used in ORCHIDEE, considering all grid cells where the PFT is present. The x-axis represents the LRU value between 0 and 3, with 0.1 bins. The y-axis represents the occurrences. For each PFT, the black distribution is computed using a monthly climatology of simulated COS and GPP fluxes (LRU_MonthlyFluxes), the blue distribution is computed using the monthly climatology of LRU values estimated at the original half-hourly time step (Monthly_LRU), the red vertical bar represents the median LRU value, the green vertical bar represents the LRU optimal value that minimizes the error between plant COS uptakes estimated at a monthly time step by the mechanistic approach and the LRU approach, for all pixels of the considered PFT.**

We see that the blue distribution yields larger LRU values. This shows that LRU is scale dependent. The values to be used should be coherent with their usage. For example, the optimal values we computed are lower than values estimated from measurements, but they are adapted to make the link with atmospheric COS studies.
We also updated the manuscript correspondingly with these new results at large scale.

Breiman, L.: Random forests, Mach. Learn., 45(1), 5–32, doi:10.1023/A:1010933404324, 2001.

*[3] One more comment, or maybe rather a suggestion, regards the LRU concept, which has to be understood as the simplest possible, but still process-based, closure possibility in order to use measurements of the COS flux to estimate GPP. In choosing some LRU value, implicit assumptions are made about the ratio of conductances, which is dealt with in this paper, but also about the Ci/Ca ratio (Seibt et al. 2010, Wohlfahrt et al. 2012). This has always been a major criticism of the LRU concept, as it amounts to ignoring the demand side of the CO2 diffusion equation. I am thus wondering whether the authors could make a useful contribution here by expanding their analysis to look into the spatio-temporal Ci/Ca ratio, given that Ci must be calculated by their mechanistic model, and confront these with our understanding of how Ci/Ca varies across the globe and seasonally.*
Answer: We first add some precision on how $[CO_2]_a$ is prescribed in ORCHIDEE: "To account for the $CO_2$ fertilization effect we considered global means of $[CO_2]_a$ with yearly varying values, as provided by the TRENDY model inter-comparison project (Sitch et al., 2015). The impact of not taking into account the spatial and temporal variations of $[CO_2]_a$ on GPP has been studied in Lee et al. (2020); while this simplification has indeed no impact at global yearly scale for GPP, this may be less true at site and seasonal scales."
We have also added: "The $CO_2$ assimilation, the stomatal conductance and the intercellular $CO_2$ concentration $C_i$ are computed per LAI layer, provided LAI is higher than 0.01 and the mean monthly temperature is higher than -4°C. The $CO_2$ assimilation and the stomatal are further summed-up over all layers to compute GPP and the total conductance at canopy level.".
We agree with the Referee that, following the equation (8) developed in Seibt et al. (2010), the LRU explicitly depends on only two variables: the $g_{S\_COS}$ to $g_{I\_COS}$ ratio, and the $C_i$ to $[CO_2]_a$ (Ca) ratio.

[Figure]

**Figure R1.6: Top: Seasonal evolution of the simulated $C_i$ to $[CO_2]_a$(Ca) ratio at the Harvard Forest site in 2012 (green curve) and the Hyytiälä site in 2017 (red curve). Bottom: Mean diel cycle over May-September**

The modelled daily mean values for the $C_i$ to Ca ratio computed at the two sites vary between 0.68 and 1.00 (Figure R1.6 top), on the upper part of the values reported by Seibt et al. (2010). The variations are also in agreement with Prentice et al. (2014) who state that the $C_i$ to Ca ratio is pretty stable with only ± 30% variations. The mean diel cycle between May and September has a U-shape with values decreasing during daytime (Figure R1.6 bottom), in coherence with former findings (Tan et al., 2017).

[Figure]

**Figure R1.7: Same as R1.1 for the $C_i$ to Ca ratio**

Random Forest and partial correlations analyses give the same main factors as $VPD$, $PAR$ and $T_{air}$ for both sites (respective partial correlations are -0.81, -0.59 and -0.17 at Harvard and -0.69, -0.52 and 0.35 at Hyytiälä).

These results were added to the site LRU section which was renamed "LRU variability", the Figures were added in the Appendices.

[Figure]

**Figure R1.8: Map of simulated monthly means of $C_i$ to $[CO_2]_a$(Ca) averaged over the 2000-2009 period**

[Figure]

**Figure R1.9: Map of the coefficient of variation of the simulated monthly means of $C_i$ to $[CO_2]_a$(Ca) computed over the 2000-2009 period**

At global scale, most grid-cells have a yearly mean value for valid $C_i$ to Ca ratio between 0.6 and 0.9; in the more arid regions, the mean value can decrease to around 0.3 (Figure R1.8). At a monthly time-scale, the coefficient of variation is lower than 10% for a majority of grid cells (Figure R1.9).

Lee, E., Zeng, F.-W., Koster, R. D., Weir, B., Ott, L. E. and Poulter, B.: The impact of spatiotemporal variability in atmospheric CO2 concentration on global terrestrial carbon fluxes, Biogeosciences, 15(18), 5635–5652, doi:10.5194/bg-15-5635-2018, 2018.

Prentice, I. C., Dong, N., Gleason, S. M., Maire, V. and Wright, I. J.: Balancing the costs of carbon gain and water transport: Testing a new theoretical framework for plant functional ecology, Ecol. Lett., 17(1), 82–91, doi:10.1111/ele.12211, 2014.

Tan, Z. H., Wu, Z. X., Hughes, A. C., Schaefer, D., Zeng, J., Lan, G. Y., Yang, C., Tao, Z. L., Chen, B. Q., Tian, Y. H., Song, L., Jatoi, M. T., Zhao, J. F. and Yang, L. Y.: On the ratio of intercellular to ambient CO2 (ci/ca) derived from ecosystem flux, Int. J. Biometeorol., 61(12), 2059–2071, doi:10.1007/s00484-017-1403-4, 2017.

Sitch, S., Friedlingstein, P., Gruber, N., Jones, S. D., Murray-Tortarolo, G., Ahlström, A., Doney, S. C., Graven, H., Heinze, C., Huntingford, C., Levis, S., Levy, P. E., Lomas, M., Poulter, B., Viovy, N., Zaehle, S., Zeng, N., Arneth, A., Bonan, G., Bopp, L., Canadell, J. G., Chevallier, F., Ciais, P., Ellis, R., Gloor, M., Peylin, P., Piao, S. L., Le Quéré, C., Smith, B., Zhu, Z. and Myneni, R.: Recent trends and drivers of regional sources and sinks of carbon dioxide, Biogeosciences, 12(3), 653–679, doi:10.5194/bg-12-653-2015, 2015.

*Detailed comments:*
*[4] l. 82-83: fluxes and mixing ratios in µmol/m2s and µmol/mol, respectively, result in tiny numbers – why not use units of pmol/m2s and pmol/mol which are typically used in practice?*
Answer: The Referee is right, we have changed the units using pmol for COS variables.

[5] l. 85: . . . if Fcos, [CO2]a and [COS]a are available . . .
Answer: Yes, we have added this complement.

*[6] l. 85-96: this is all well-known – the real issue in my view is that we are seeing a large spread in LRU values even after normalizing for PAR that we cannot explain; also note that Yang et al. (2018) clearly show that low-light periods, when LRU deviates most, typically contribute little to the daily and longer-term integrated GPP.*

Answer: We agree with the Referee. We have added the following sentence: "The diel variation of LRU with light may however be only of second order importance as GPP is very low at low light, and Yang et al. (2018) found that considering sub-daily variations of LRU when computing daily mean GPP values had no importance."

*[7] l. 95: and possibly other factors we do not yet understand*
Answer: This is true. We have added: "We also have to acknowledge that there are still factors that are not accounted for if discrepancies between GPP and COS-based estimations are larger than their estimated respective uncertainties."

*[8] l. 97-100: this is essentially a technical objective – can the authors come up with an objective that aims at providing an advancement in scientific understanding and/or even some hypotheses of what they believe will result from adopting the mechanistic as opposed to LRU approach?*
Answer: We have clarified the context of this study, and hopefully presented our goal in a more scientific and attractive way:
"Before being able to use COS observations to constrain the simulated GPP, Land Surface Models (LSMs) first need to have an accurate model to simulate vegetation COS fluxes. In a former study, Launois et al. (2015b) simply defined the COS uptake by vegetation as the $CO_2$ gross uptake simulated by LSMs, scaled with a constant LRU value for each large vegetation class. **The goal of this study is to now simulate the uptake of atmospheric COS by continental vegetation in a more complex and realistic way using a mechanistic approach within an LSM, and to apply this model to evidence the shortcomings or pertinence of the LRU concept, depending on the studied scales**. To this end:
  i)   We used the state-of-the art ORCHIDEE LSM (Krinner et al., 2015), and implemented in it the vegetation COS uptake model of Berry et al. (2013) to simulate the COS fluxes absorbed at the leaf and canopy levels by the continental vegetation.
  ii)  We evaluated the simulated COS fluxes against measurements at two forest sites, namely the Harvard Forest, United States (Wehr et al., 2017), and Hyytiälä, Finland (Kooijmans et al., 2019; Kohonen et al., 2020; Sun et al., 2018a). **We studied the high frequency behaviour of the modelled conductances over the season and the dependency of the LRU on the environmental and structural conditions.**
  iii) We compared the simulated mechanistic COS fluxes at global scale to former estimates; **we studied LRU values estimated from monthly fluxes, that are pertinent for atmospheric studies, and compared them to monthly means of high-frequency LRU values**.
  iv)  The mechanistic and LRU simulated COS fluxes were used with the atmospheric transport model LMDz (Hourdin et al. 2006), to provide atmospheric COS concentrations that were evaluated against measurements at sites of the NOAA network.

*[9] l. 139: ". . . is generally not produced by plants (but see Gimeno et al. 2017)."*
Answer: Yes, we have added "generally" in this sentence, and cited Gimeno et al. (2017) in the introduction: "It is however to be noted that Gimeno et al. (2017) reported COS emissions by bryophytes during daytime".

[10] l. 140: suggest to remove "to heat and " as this sounds as if the boundary layer conductance to heat would affect the flux of COS
Answer: We agree, this part has been removed.

[11] l. 164: the seminal paper to nighttime transpiration and residual conductance in my view is Dawson et al. (2007)
Answer: Thanks for drawing our attention to this reference, which we are now citing.

*[12] l. 173: does the model then also calculate H2O gas exchange at night or only COS?*
Answer: The model now also calculates H2O gas exchange at night, which should improve the estimation of the transpiration flux. This impact on the modelled transpiration flux at night may be low, as the atmospheric demand is around zero at night, however *VPD* values at night follow an increasing trend (Sadok and Jagadish, 2020). It is to be noted that Barnard and Bauerle (2013) found, based on sensitivity analyses, that the minimum stomatal conductance $g_0$ was the parameter having the largest influence on their modelled transpiration estimates. They also stress that $g_0$ should maybe be seen as an asymptotic minimal value, rather than an offset.
For now, the absolute vegetation COS fluxes at night are slightly overestimated as compared to observed fluxes (updated Figure 4a for Harvard and Figure B1a for Hyytiälä), thus hinting at overestimated nighttime stomatal conductances. This advocates for the optimization of the $g_0$ parameter based on nighttime fluxes measurements. We have slightly updated the section "Exploiting nighttime conductances" based on these comments.

Barnard, D. M. and Bauerle, W. L.: The implications of minimum stomatal conductance on modeling water flux in forest canopies, J. Geophys. Res. Biogeosciences, 118(3), 1322–1333, doi:10.1002/jgrg.20112, 2013.

*[13] l. 195: What does "Vegetation COS flux direct or derived measurements . . ." mean?*
Answer: We rephrased more clearly: "Vegetation COS fluxes can be measured using branch chambers or estimated using the difference between measurements of ecosystem and soil fluxes. Such measurements were available at the Hyytiälä (Finland) and Harvard Forest (United States) FLUXNET sites."

*[14] Suggest to reformulate Eq. (4) and (5): what about some statistics which allow evaluating systematic biases?*
Answer: We have added the computation of the bias, standard deviations and correlation coefficient as equations (6) to (8):

$$bias = \overline{F_{COS}^{Mod}} - \overline{F_{COS}^{Obs}} \tag{6}$$

$$SD^{Mod} = \sqrt{\frac{\sum_{n=1}^{N}\left(F_{COS}^{Mod}(n) - \overline{F_{COS}^{Mod}}\right)^2}{N}}$$

$$SD^{Obs} = \sqrt{\frac{\sum_{n=1}^{N}\left(F_{COS}^{Obs}(n) - \overline{F_{COS}^{Obs}}\right)^2}{N}} \tag{7}$$

$$r = \frac{\sum_{n=1}^{N}\left(F_{COS}^{Obs}(n) - \overline{F_{COS}^{Obs}}\right) \cdot \left(F_{COS}^{Mod}(n) - \overline{F_{COS}^{Mod}}\right)}{N \cdot SD^{Obs} \cdot SD^{Mod}} \tag{8}$$

*[15] l. 226: this is the first citation of a table and thus table number should be 1*
Answer: We have done so and moved the Table in this section.

*[16] l. 310: wouldn't it make sense to compare modelled and measured canopy-scale COS fluxes before embarking on a detailed analysis of the underlying processes, i.e. move sections 3.1.2.1 and 3.1.2.2 here?*
Answer: The authors were initially divided on this topic, we have now swapped the sections order.

*[17] Fig. 1: what is the area reference for the units – m2 leaf area or m2 ground area?*
Answer: The area reference for the units is m$^2$ ground area, we have added this information in the Figure legend.

*[18] l. 336-340: alpha values depend on Vmax, which I suppose changes with phenology? If so, would it make sense to also investigate Vmax?*
Answer: Does the Referee mean $g_{I\_COS}$ (because alpha is a constant)? First, we need to precise how $V_{max}$ is computed in ORCHIDEE. We have thus added a description in the ORCHIDEE model section (2.1.1): "For each PFT, the reference value for the maximum photosynthetic capacity at 25°C, $V_{max,25}$, is derived from literature survey, observation databases, possibly later calibrated using FLUXNET observations (e.g. Kuppel et al., 2012). To compute the maximum photosynthetic capacity at leaf level, $V_{max}$, the reference value is multiplied at a daily time step by the relative photosynthetic efficiency of leaves based on the mean leaf age following Ishida et al. (1999) (see equation A12 and Figure A12 in Krinner et al., 2005). Leaves are very efficient when they are young and stay so till they approach their pre-defined leaf lifespan". We then agree that leaf-level $V_{max}$ indirectly depends on phenology through the leaf age. To illustrate this, we have plotted the seasonal evolution of $V_{max}$ at the Harvard and Hyytiälä sites (Figure R1.10).

[Figure]

**Figure R1.10: Seasonal evolution of the leaf-level maximum photosynthetic capacity at the Harvard Forest site in 2012 (green curve), and at the Hyytiälä site in 2017.**

At the Harvard deciduous forest, the leaf-level $V_{max}$ is zero when there are no leaves, it grows sharply to the maximum value of 50 mmol $CO_2$ m$^{-2}$ s$^{-1}$ at the beginning of the growing season, stays on a plateau, and declines when leaves grow older. At the Hyytiälä evergreen forest, $V_{max}$ stays much stable as leaves are produced and shed continuously.

In the photosynthesis module, a temperature dependency is further applied following Medlyn et al. (2002) and Kattge and Knorr (2007), using a modified Arrhenius function and linear dependences of its parameters for temperature acclimation. The internal conductance $g_{I\_COS}$ is computed as proportional to $V_{max}$, and integrated over LAI. $g_{I\_COS}$ is thus a proxy for canopy-scale $V_{max}$, and its seasonal variations indeed closely follow the leaf phenology.

Medlyn, B. E., Dreyer, E., Ellsworth, D., Forstreuter, M., Harley, P. C., Kirschbaum, M. U. F., Le Roux, X., Montpied, P., Strassemeyer, J., Walcroft, A., Wang, K. and Loustau, D.: Temperature response of parameters of a biochemically based model of photosynthesis. II. A review of experimental data, Plant, Cell Environ., 25(9), 1167–1179, doi:10.1046/j.1365-3040.2002.00891.x, 2002.

*[19] l. 346-347: I would turn around the argumentation here, i.e. say that due to the way of how gs is simulated according to Yin and Struik (2009) there is a linear relationship with A ...*
Answer: We agree with the Referee, and have rephrased: "Due to the way of how $g_{S\_COS}$ is simulated according to Yin and Struik (2009), there is a linear relationship with the $CO_2$ assimilation, which depends mainly on PAR."

*[20] l. 354-355: but isn't Vmax strongly driven by phenology as well and wouldn't this be the main seasonal driver?*
Answer: Given our former answer to comment [18], we agree that the seasonal cycle of LAI is a predictor to be considered, mainly for deciduous forests, and we have added it in our analysis (see answer to comment [1]).

*[21] Fig. 4: it looks like there is a data gap in May 2013, but the dashed line is not broken and instead interpolates through the gap, compared to 2012 when the dashed line stops during all gaps*
Answer: The Referee is right, we formerly computed the standard deviation based on daily means, with a standard deviation being zero for weeks with only one daily mean. We have now imposed a minimum of two daily means to have a representative weekly mean, and Figure 4 and related statistics have been updated accordingly. We added in the legend: "We imposed to have at least observations on two different days to compute the corresponding weekly mean".

[Figure]

**Figure R1.11: Updated Figure 4**

A similar Figure was added in the Appendices for the Hyytiälä site.

[Figure]

**Figure R1.12: a. Mean diel cycle of observed vegetation COS flux derived from ecosystem COS flux (Kohonen et al., 2017) and soil COS flux (Sun et al., 2018a), and modelled COS vegetation flux in July-September 2015 (Hyytälä) using an atmospheric convention where an uptake of COS by the ecosystem is negative. The shaded areas above and below each curve represent one standard-deviation of the considered half-hourly values over the July-September period. b. Mean seasonal cycle of simulated and observed weekly average vegetation COS flux in 2015 (Hyytälä). The shaded areas above and below each curve represent one standard-deviation of the daily means within the considered week.**

*[22] l. 395: there also appear to be issues in terms of phenology, e.g. in April 2013*
Answer: Yes, we have modified this paragraph to compare the signals more accurately: "The simulated weekly seasonal vegetation COS uptake roughly follows the same trend as the observed one (r=0.53, Figure 1b). COS uptake increases in spring when the vegetation growing season starts and decreases in autumn at the end of the forest activity period. Simulated and observed fluxes also take similar values over the two years. There are however differences: in 2013 the start of the season is simulated about two weeks too late in May instead of late April, and measured fluxes peak in May-June and August-September, while the modelled fluxes peak in July".

*[23] l. 396: can the authors estimate the magnitude of the noise (I presume they refer to the random variability of EC flux measurements) and whether this noise could be responsible for the observed variability?*
Answer: We have also clarified our meaning regarding the uncertainty of ecosystem flux measurements: "Kohonen et al. (2020) have quantified the relative uncertainty of weekly-averaged ecosystem COS fluxes at 40%, which is coherent with the large standard-deviation computed for field data (Figure 1b)".

*[24] l. 403: I would replace "diurnal" with "daytime", which I believe better fits with what the authors mean.*
Answer: We have changed "diurnal" and "nocturnal" to "daytime" and "nighttime".

*[25] Fig. 6: what do the model simulations of LRU refer to – the canopy scale? If so what do the PAR values refer to – the PAR incident at the top of the canopy? If so then I would say that there is a scale issue affecting the comparison to the branch-chamber measurements, as the canopy LRU is the integral over all leaves, both sunlit and shaded, which experience very different PAR values; having said this, I however would expect canopy simulated LRU values to lie above the measurements at low PAR, as when PAR incident on the top of the canopy is say 200 μmol/m2s, a large fraction of the leaves will be in the shade and experience much lower PAR values and should thus actually have a higher LRU; in any case the authors should critically reflect on whether any scale issues affect the interpretation of the comparison between measured branch-chamber and simulated, presumably canopy-scale, LRUs*
Answer: The Referee is correct, we use by default the LRU term at canopy scale for the model as we compute it using COS and GPP fluxes integrated at the canopy level. We have now stressed this point in the manuscript, in

the presentation of the ORCHIDEE model: "The canopy is discretized in several layers of growing thickness, the number depending on the actual Leaf Area Index (LAI). All the incoming light is considered to be diffuse, and no distinction is made between sun and shaded leaves. The light is attenuated through the canopy following a simple Beer-Lambert absorption law. The $CO_2$ assimilation, the stomatal conductance and the intercellular $CO_2$ concentration $C_i$ are computed per LAI layer, provided LAI is higher than 0.01 and the mean monthly air temperature is higher than -4°C. The $CO_2$ assimilation, the stomatal conductance are further summed-up over all layers to compute GPP and the total conductance at canopy level". Similarly, in the presentation of the Berry model, we have added: "The vegetation COS flux and related conductances are computed for each LAI layer, and then summed-up to get total values at canopy level. Unless specified otherwise, fluxes, conductances and LRU are further presented and discussed at canopy level".

In the presentation of Figure 6, we have added: "It is to be noted that we compare here LRU values estimated from measurements at branch level to modelled LRU estimated at canopy level. We conducted a similar modelling study considering only the top of canopy level and the associated COS and GPP fluxes, yielding similar results (not shown). This can be linked to the fact that the version of ORCHIDEE we use considers all the incoming light to be diffuse, and does not distinguish between sun and shaded leaves. We thus have similar LRU values at all canopy levels."

*[26] Fig. 7 and 9: appropriate x- and yaxis scale and text needed*
Answer: We have updated Figures 7 and 9, adding latitudes and longitudes.

[Figure]

**Figure R1.13: Updated Figure 7**

[Figure]

**Figure R1.14: Updated Figure 9**

*[27] Table 5: can the authors add the time frame to the table to which the different studies refer to?*
Answer: We have done so.

**Table R1.2: Overview of COS plant uptake per year (Gg S yr⁻¹)**

|  | Kettle et al. (2002) | Montzka et al. (2007) | Suntharalingam et al. (2008) | Berry et al. (2013) | Launois et al. (2015b) | | | This study |
|--|--|--|--|--|--|--|--|--|
|  |  |  |  |  | ORC. | LPJ | CLM4 |  |
| Period study | circa 1990-2000 | 2000-2005 | 2001-2005 | 2002-2005 | 2006-2009 | | | 2000-2009 |
| Uptake by plants | -238 (±30) | -730 to -1500 | -490 (-460 to -530) | -738 | -1335 | -1069 | -930 | -756 |

*[28] l. 459: and you use the same parameterization of COS uptake?!*
Answer: Indeed! We added: "on top of using the same mechanistic model for vegetation COS uptake".

*[29] Fig. 7: can the authors avoid the figure legend to overlap with the plotted lines?*
Answer: We presume the Referee means Figure 11 (above l. 545). The Figure has been remade without the legend overlapping the curves.

[Figure]

**Figure R1.15: Updated Figure 11.**

*[30] l. 546, 549: isn't Barrow in Alaska/USA?*
Answer: Yes, we are sorry for this oversight, we have corrected.

*[31] l. 561: to what does the citation Whelan and Seibt refer to?*
Answer: We clarified the sentence "… than the other LRU approaches with values from Whelan et al. (2018) and Seibt et al. (2010)".

*[32] l. 598-599: agreed, but it should be acknowledged that at present the spatial coverage of biomes is very poor (some biomes not having been sampled at all, for others n=1) and also temporal coverage is poor, with the exception of Hyytiälä and Harvard forest most COS flux measurements being confined to less than a year, leaving a big question mark with regard to inter-annual variability*
Answer: The Reviewer is right. We have added the following sentence: "We acknowledge however the scarcity of available measurements for the time being, with no samples for most biomes, a few sites with less than one year of data, and only Hyytiälä allowing for interannual variability studies".

*[33] l. 603-606: unclear what this sentence tries to say*
Answer: This part was removed from the discussion, and the modelled and observations-based conductances are now compared more precisely in the "Modelled conductances" section: "For the Harvard Forest site, Wehr et al. (2017) computed the stomatal conductance using both a water flux method and a COS flux method, and obtained a close agreement between two different methods; the mesophyll conductance is modelled using an experimental temperature response, and the biochemical conductance, representing CA activity, is modelled using a simple parameter (0.055 mol m$^{-2}$ s$^{-1}$), both scale with LAI to get canopy estimates. Wehr et al. (2017) found similar maximum values around 0.27 mol m$^{-2}$ s$^{-1}$ during daytime, from May to October, for the stomatal conductance and for the biochemical conductance (their Figure 4); adding the slightly larger mesophyll conductance (peaking around 1.0 mol m$^{-2}$ s$^{-1}$) to the biochemical conductance would thus also lead to a more limiting role of the internal conductance (peaking around 0.21 mol m$^{-2}$ s$^{-1}$) during daytime, albeit not as strong as for the modelled one (peaking around 0.13 mol m$^{-2}$ s$^{-1}$); the simulated stomatal conductance exhibits minimum and maximum values similar to the observations-based ones, but peaks more sharply in the morning."

*[34] l. 633-634: can you elaborate on which data would be required to that end?*
Answer: We have added: "A model for the mesophyll conductance is already implemented in ORCHIDEE, with a simple parameter depending on temperature through a multiplication by a modified Arrhenius function following Yin & Struik (2009). Regarding CA activity, we could test the simple model using a constant value presented in Wehr et al. (2017). $^{13}$C discrimination of the isotopic composition of $CO_2$ exchanges allows for a continuous estimation of the mesophyll conductance (Stangl et al., 2019). Measuring CA activity can be done at a coarser frequency, using different techniques (Henry, 1991)."

*[35] l. 645: nighttime instead of nitghttime*
Answer: Thanks, we have corrected.

*[36] l. 666: here the authors might refer to some recently published experimental canopy-scale LRU estimates, such as Spielmann et al. (2019, 2020) who provide estimates for deciduous forest, savanna, C3 crop and C3 grassland*

Answer: We added: "More recently, Spielman et al. (2019) estimated LRU values from ecosystem and soil measurements: 0.89 for an agricultural soybean field, 1.02 for a temperate C3 grassland, 2.22 for a temperate beech forest and 2.27 for a Mediterranean savanna ecosystem; our corresponding PFTs respectively give: 1.37 (C3crops), 1.18 (TempC3grass), 1.31 (TempBroSum) and 1.06 (TempBroEver), with thus higher estimates for herbaceous plants and lower ones for trees."

*[37] l. 669: another issue, mentioned first in Wohlfahrt et al. (2012), is that leaf chambers are typically vigorously ventilated and LRU values inferred from these are thus implicit with the correspondingly large boundary layer conductance; under real-world conditions boundary layer conductances vary with wind speed (temporally but also with depth of the canopy) and transport across the boundary layer may not necessarily be dominated by forced convection as in leaf chambers; that is to say that for this reason leaf-chamber LRUs may not be easily transferable to real-world conditions*

Answer: we have added this concern: "We may also add that LRU values derived from measurements performed in leaf chamber measurements, that are well-ventilated and thus associated with large leaf boundary layer conductances, may not be representative of the real-world transfer processes, where the boundary layer conductances vary with wind speed, temporally and within canopy depth (Wohlfahrt et al., 2012)".

*[38] l. 678: in this section you might cite Spielmann et al. (2020) who show, with a simple linear perturbation analysis, that variability in [COS]a does in fact affect LRU.*

Answer: We agree on the equations derived from the linear perturbation analysis in Spielmann et al. (2020), but in the case of a modelled vegetation COS flux that is proportional to $[COS]_a$, the dependency of LRU to $[COS]_a$ drops. This is the case for the Berry et al. model (2013) and for the Seibt et al. (2010) model where $[COS]_a$ does not appear their equation (8). We can see in the Figure 7b of Spielmann et al. (2020) that in September, before the last grassland cut, the respective impacts of the decreasing COS concentration (in blue) and of the decreasing COS flux (in orange) are of the same magnitude but of opposite signs, thus cancelling each other out.

Biogeosciences Discuss.,
https://doi.org/10.5194/bg-2020-381-RC2, 2020
*[1] Maignan et al., develop a parameterization for including OCS uptake within a well-known land surface model (ORCHIDEE ). Since OCS has been proposed as a proxy for GPP (currently not observable at large scales), this work is an important step towards estimating global GPP. The authors do a very thorough job of combing the OCS literature for published values of leaf relative uptake (LRU) and validating modeled uptake at two temperate sites where OCS flux is measured. I think this work should eventually be published in Biogeosciences. However I am currently a little confused with regard to how the paper is framed. The authors pose their formulation of OCS flux using a conductance based approach against the well-known LRU based relationship between FOCS and GPP. The LRU approach is also based on conductance, with simple assumptions regarding the relative role of mesophyll and stomatal conductances (Seibt et al., 2010). Thus the differences in flux resulting between the two approaches are not surprising, and don't really manifest themselves once transported and compared against OCS flask measurements from NOAA. This is an interesting finding, but I think the authors must focus on a more scientific question.*

Answer: We thank the Referee for this analysis. We must precise here that what we call the "LRU approach" in this study, is based on a former work with ORCHIDEE (Launois et al., 2015b), which does not consider a varying LRU as modelled in Seibt et al. (2010, their equation (8)), but uses constant values for each Plant Functional Type (PFT). We have thus clarified the context of this study and formulated our goal in a more scientific way: "In a former study, Launois et al. (2015b) simply defined the COS uptake by vegetation as the $CO_2$ gross uptake simulated by LSMs, scaled with a constant LRU value for each large vegetation class. The goal of this study is to now simulate the uptake of atmospheric COS by continental vegetation in a more complex and realistic way using a mechanistic approach within an LSM, and apply this model to evidence the shortcomings or pertinence of the LRU concept, depending on the studied scales". In the revised manuscript we have thus emphasized the study of the LRU variability both at hourly and site scales, and at monthly and global scales. This can be seen in more details through our answers to both Referees' comments.

*[2] The other main comment I have is about modeled gs,ocs and gi,ocs. I agree with the authors that the role of gi is important and often ignored, but I am somewhat skeptical of the large diurnal variation in modeled gi, which is related to temperature. As I understand, gi is estimated from Vmax of Rubisco, but Rubisco response to temperature is not thought to be that large, particularly at temperatures observed at the temperate NH sites (see Sage and Kubien 2007). Moreover, while estimates of diurnal variability of mesophyll conductance hasn't been reported much, a recent study showed that the diurnal variability in gm (which is similar to gi in this study) is much smaller than gs (Strangl et al., 2019; which is also a high latitude coniferous forest, so somewhat similar to Hyytiälä ). This could serve as a mechanism for plants to modify (increase) water use efficiency and therefore continue assimilating carbon even as gs declines due to high VPD commonly observed after mid-day (see also Buckley and Warren, 2014). See also recent work by Gimeno et al., (2020). Thus, I am quite surprised that at Hyytiälä it seems like gas exchange is most often limited by gi and not gs. I believe some sensitivity analyses could be done with regard to the formulation of gi. The temperature dependance of gi is also not uniform across plant species (von Caemmerer and Evans, 2015), with obvious implications for the global formulations presented here. I think a more in-depth discussion of the implications of gas exchange most limited by gi should be added, replacing the current and mostly qualitative discussion currently.*

Answer: We first need to say that we have corrected a problem in the computation of the total internal conductance (which was wrongly systematically scaled with the maximum LAI of 12 instead of the actual LAI, but did not affect the computation of fluxes) and have redrawn Figure 1.

[Figure]

[Figure]

[Figure]

**Figure R2.1: Updated Figure 1**

The amplitude of the diel variation of the internal conductance is then much reduced (Figure R2.1). The diel variation of the $V_{max}$ variable is based on a modified Arrhenius function calibrated on experimental data (Medlyn et al., 2002).

We thank the Referee for drawing our attention to the Stangl et al. (2019) study, and recall that the internal conductance includes both the mesophyll conductance and the biochemical CA activity. Stangl et al. (2019) studied the diurnal variations of stomatal and mesophyll conductances on mature *Pinus Sylvestris* trees in northern Sweden. Interestingly they found that the stomatal and mesophyll conductances estimated from measurements in June and July 2017 have significant diurnal variations (their Figure 2), with the stomatal conductance peaking between 9 and 10 am, earlier in the day than the mesophyll conductance. Albeit we don't know about the biochemical conductance, this is similar to what we see at Hyyiälä in the June-August period, and Kooijmans et al. (2019) found indications from branch measurements that the internal conductance may be limiting COS uptake during daytime.

The cited references encourage us to have a closer look at the mesophyll conductance, and we have updated our discussion on limiting conductances: "We have to acknowledge the large uncertainty regarding the modelling of the internal conductance. In parallel to optimizing the parameters of the internal conductance, an improvement could thus also be to replace it by the two factors it represents, i.e. the mesophyll conductance and CA activity. A model for the mesophyll conductance is already implemented in ORCHIDEE, with a simple parameter depending on temperature through a multiplication by a modified Arrhenius function following Medlyn and al. (2002) and Yin & Struik (2009). The impact of mesophyll conductance on photosynthesis and water use efficiency is now more studied (e.g. Buckley and Warren, 2014), even if its modelling remains challenging too: the temperature response has notably been reported as highly variable between plant species (von Caemmerer and Evans, 2015), which would imply having PFT-dependent parameters. Regarding measurements, $^{13}$C discrimination of the isotopic composition of $CO_2$ exchanges allows for an estimation of the mesophyll conductance (Stangl et al., 2019)."

We agree a sensitivity analysis is required too, but this is a further large effort to be led in a separate study (see answer to comments [6] and (7)).

Medlyn, B. E., Dreyer, E., Ellsworth, D., Forstreuter, M., Harley, P. C., Kirschbaum, M. U. F., Le Roux, X., Montpied, P., Strassemeyer, J., Walcroft, A., Wang, K. and Loustau, D.: Temperature response of parameters of a biochemically based model of photosynthesis. II. A review of experimental data, Plant, Cell Environ., 25(9), 1167–1179, doi:10.1046/j.1365-3040.2002.00891.x, 2002.

*[3] One way to understand the relative roles of gs, gi, and the differences between the mechanistic model and the LRU approach is to look at values of intercellular and chloro-plast CO2 concentrations (ci and cc respectively). These should be standard outputs of the model. ORCHIDEE ci and cc could be compared with "inferred" ci and cc using modeled A (GPP) and gi,cos and gs,cos following Seibt et al., 2010. It would be interesting to see if those differences can explain the difference in FOCS flux obtained from those two approaches (for e.g., in Figure 9).*
Answer: We again clarify that what we called the "LRU approach" in this study was using a constant LRU value per large vegetation class. The equation (8) derived in Seibt et al. (2010) describing the variation of LRU as a function of the $g_{S\_COS}$ to $g_{I\_COS}$ ratio, and the $C_i$ to $[CO_2]_a$ (Ca) ratio, is also valid in our mechanistic framework. So yes, the LRU variability can be mostly explained by these two ratios (the leaf boundary conductance has been neglected). The partial correlations analysis between the LRU and these two ratios yields similar values larger than 0.6 in absolute values at both sites, the Random Forest analysis also shows that both are equally important (not shown). We now indeed look at the $C_i$ to Ca ratio (but not at $C_c$, as our modelled internal conductance includes both the mesophyll conductance and the biochemical CA activity). Considering a constant LRU thus may be seen as neglecting the variability of the $C_i$ to Ca ratio.

[Figure]

**Figure R2.2: Top: Seasonal evolution of the simulated $C_i$ to $[CO_2]_a$(Ca) ratio at the Harvard Forest site in 2012 (green curve) and the Hyytiälä site in 2017 (red curve). Bottom: Mean diel cycle over May-September**

The modelled daily mean values for the $C_i$ to Ca ratio computed at the two sites vary between 0.68 and 1.00 (Figure R2.2 top), on the upper part of the values reported by Seibt et al. (2010). The variations are also in agreement with Prentice et al. (2014) who state that the $C_i$ to Ca ratio is pretty stable with only ± 30% variations. The mean diel cycle between May and September has a U-shape with values decreasing during daytime (Figure R2.6 bottom), in coherence with former findings (Tan et al., 2017).
Given the non-linearity of the problem, the link between the $C_i$ to Ca ratio and the LRU is less strong when dealing with monthly means at global scale, the partial correlation for all grid-cell-PFT drops to 0.3.

Prentice, I. C., Dong, N., Gleason, S. M., Maire, V. and Wright, I. J.: Balancing the costs of carbon gain and water transport: Testing a new theoretical framework for plant functional ecology, Ecol. Lett., 17(1), 82–91, doi:10.1111/ele.12211, 2014.

Tan, Z. H., Wu, Z. X., Hughes, A. C., Schaefer, D., Zeng, J., Lan, G. Y., Yang, C., Tao, Z. L., Chen, B. Q., Tian, Y. H., Song, L., Jatoi, M. T., Zhao, J. F. and Yang, L. Y.: On the ratio of intercellular to ambient $CO_2$ (ci/ca) derived from ecosystem flux, Int. J. Biometeorol., 61(12), 2059–2071, doi:10.1007/s00484-017-1403-4, 2017.

*[4] In general, I find the manuscript too long and perhaps some figures (e.g. Figure 2) can be removed. Similarly, perhaps figures 3 and 4 can be combined. I greatly appreciate all the work that has gone in to this, but perhaps the authors ought to split this in two papers. One that describes the modeling framework in ORCHIDEE and another that focuses on the transport modeling? This would allow the authors to delve more deeply in the important findings such as Fig 8. Similarly, I can imagine that Fig 11 discussion can be greatly expanded upon.*

Answer: As suggested by the Referee, we removed Figure 2 and combined Figures 3 and 4. We also moved Figure 10 in the Appendices.

[Figure]

**Figure R2.3 Combination of former Figures 3 and 4 for the Harvard Forest site**

We acknowledge the fact that the manuscript is rich, but we would prefer to not split it. The transport part is not the focal part in this study, and Figure 11 is here to show that, at large scales, the mechanistic and LRU approaches provide similar results as evaluated by atmospheric COS concentrations. The result is important when it comes to use the LRU approach to estimate the GPP through inverse modelling of atmospheric measurements. However, we agree that there is much to say regarding transport and concentrations, and the transport model errors for COS will be assessed at NOAA sites by performing an intercomparison experiment with several transport models in a separate study; the other components of the COS budget (ocean, soil…) will be transported and a more complete comparison between observed and simulated concentrations will be made.

*[5] I find it a little odd that LMDZ doesn't match the seasonal cycle of CO2 at MLO. Is. this a known issue of the transport model? In summary, I am not entirely convinced of the transport analyses since inferences could be flawed due to erroneous transport. Some quantification of transport error/uncertainty should be presented (perhaps using withheld/independent observations?).*

Answer: A preliminary analysis between the TM5 and LMDz transport models suggests that transport errors are of second importance on the seasonal cycle at the NOAA stations compared to the impact of COS fluxes (Figure R2.4). The LMDz transport model is nudged toward ERA5 wind reanalysis, which prevents the large-scale advection of tracers from diverging. Note that the NOAA surface observations are independent as they have not been used to calibrate the ORCHIDEE land surface parameters. Note also that the time lag in the $CO_2$ seasonal

cycle at MLO appears when considering the land use change, harvest terms of the respiration. These terms can be poorly represented in land surface models and explain the seasonal lag.

[Figure]

[Figure]

**Figure R2.4: Detrended temporal evolutions of simulated and observed (purple curve) COS concentrations at two selected sites, for the mechanistic model ORCHIDEE, simulated with the LMDz3 (orange curve), LMDz6 (cyan curve) and TM5 (green curve) transport between 2010 and 2016. Left: Mauna Loa station (MLO, Hawaii), right: Barrow station (BRW, Alaska). The curves have been detrended beforehand and filtered to remove the synoptic variability (see Sect. 2.2.4 of the paper).**

*[6] The writing in the results section is often very qualitative and not informative (e.g., line 329: "The conductances drop in the afternoon to reach minimum values at night"). I find most of the discussion unnecessary. It seems like a re-hash of the methodology and results section (e.g., Sec. 4.1.1. and 4.1.2), with inferences that I don't think are always supported by the results (for instance lines 653-656. This is an example where the writing could be improved to make a very compelling argument about the use of OCS flux). Some of the work suggested (e.g in Sec. 4.1.3.) is well within the purview of this study and can be performed (ie., some sensitivity analyses of min gs and gi, and impact on simulated flux.).*

Answer: We usually describe what is seen on the Figures, as readers may not all be COS experts. Also, we provide numerous statistics (including RMSD, correlation coefficient, bias) to quantify model-data discrepancies. However, we tried to tighten the manuscript, where we thought it was possible without jeopardising the understanding.

We disagree with the Referee regarding the Discussion. The discussion is oriented on how to best exploit the results of this study in the optic of a further data assimilation to optimize parameters, and really use COS information to improve the simulated GPP.

We agree a sensitivity study of the fluxes to the parameters, including minimal stomatal conductance, is further needed, but as stressed by the Referee, the manuscript is already long and this sensitivity study will be led in a subsequent study, associated with the optimisation of the most significant parameters identified in the sensitivity study.

*[7] The writing overall really needs to be tightened and the conclusions seem a bit weak. This could be improved with better framing. Thus I believe that with a much clearer presentation of figures and text, this could be a much more compelling manuscript.*

Answer: The number of Figures has been reduced from 11 to 8. The manuscript is now more clearly focused on the evaluation of the mechanistic model, and its use to study the LRU variability. We have rephrased and tightened the beginning of the "Conclusions and Outlooks" section to better emphasize our main findings:

"We have implemented inside the ORCHIDEE land surface model the mechanistic model of Berry et al. (2013) for COS uptake by the continental vegetation. Modelled COS fluxes were compared at site scale against measurements at the Harvard temperate deciduous broadleaf forest (USA) and at the Hyytiälä Scots pine forest (Finland), yielding relative RMSDs of around 40% at both diel and seasonal scales. **We found that the mechanistic model yields a lower and thus more limiting internal conductance as compared to former works (Seibt et al., 2010; Wehr et al., 2017)**. The next step is to perform a sensitivity analysis (Morris, 1991; Sobol, 2001) and to optimize the most sensitive parameters related to the modelled fluxes and conductances, to get a better agreement with observations.

Our global estimate of COS uptake by continental vegetation of -756 Gg S yr-1 is in the lower range of former studies. **An important finding is that the LRU computed from monthly values of the COS and GPP fluxes yields values lower than monthly means of high-frequency LRU values**. This has consequences for atmospheric studies where COS concentrations integrate influences from fluxes at large spatial and temporal scales."

Morris, M. D.: Factorial Sampling Plans for Preliminary Computational Experiments, Technometrics, 33(2), 161, doi:10.2307/1269043, 1991.

Sobol, I. M.: Global sensitivity indices for nonlinear mathematical models and their Monte Carlo estimates., 2001.

*[8] Comments about figures and tables: Figure1: Conductance seems to be highest at 12 am ie., midnight? Growing season midday at Harvard Forest is shown to be limited by gi but Wehr et al (2017) measurements show that gs is the limiting conductance to OCS transfer. How do these numbers compare with Wehr and Kooijmans measurements (of gi)? You could also add total conductance to OCS (the quantity multiplied to [OCS]a in eq 3). In general, since you don't seem to be explaining month by month variations, maybe compress these to show 3 month means (e.g. JJA, SON etc)? Currently, it is impossible to see in detail the diurnal variations, specially when one is trying to discern at what times of day gi > gs and vice versa.*

Answer: We indeed made a mistake and have corrected '12 am' into '12 pm'. We have now grouped the months per season, and added the total conductance, as suggested by the Referee (see new Figure in comment [2]).

We have to remember that the modelled internal conductance gathers both the mesophyll conductance and the biochemical conductance representing the Carbonic Anhydrase (CA) activity. We have now detailed more precisely the comparison with Wehr et al. (2017) in the "Modelled conductances" section: "For the Harvard Forest site, Wehr et al. (2017) computed the stomatal conductance using both a water flux method and a COS flux method, and obtained a close agreement between two different methods; the mesophyll conductance is modelled using an experimental temperature response, and the biochemical conductance, representing CA activity, is modelled using a simple parameter (0.055 mol $m^{-2}$ $s^{-1}$), both scale with LAI to get canopy estimates. Wehr et al. (2017) found similar maximum values around 0.27 mol $m^{-2}$ $s^{-1}$ during daytime, from May to October, for the stomatal conductance and for the biochemical conductance (their Figure 4); adding the slightly larger mesophyll conductance (peaking around 1.0 mol $m^{-2}$ $s^{-1}$) to the biochemical conductance would thus also lead to a more limiting role of the internal conductance (peaking around 0.21 mol $m^{-2}$ $s^{-1}$) during daytime, albeit not as strong as for the modelled one (peaking around 0.13 mol $m^{-2}$ $s^{-1}$); the simulated stomatal conductance exhibits minimum and maximum values similar to the observations-based ones, but peaks more sharply in the morning". It is more difficult to compare absolute levels of conductances for Hyytiälä as the observations are made at branch level.

*[9] Figure 2. I believe this figure should be replaced by scatterplots or a table listing correlations between gs, gi and aux variables (which already exists as Table 4). Why would gi be expected to scale with soil moisture? It is strange that PAR is highest in May at Hyytiälä.*

Answer: We have removed Figure 2, updated Table 4 and added a Random Forest analysis (see our answer to comment [10] below). We double-checked for $PAR$ and found the same maximum in May. It can be observed in Koiijmans et al. (2019), their Figure S1 shows a long series of high $PAR$ in May, followed by a series of much more varying $PAR$. This is not always the case, but in 2017 the other summer months were very cloudy and rainy in Southern Finland, reducing $PAR$ (both midday value and monthly average); in May midday $PAR$ values can indeed be very high and in recent years have been higher than June/July midday PAR in at least 2013, 2016 and 2017 (K.M. Kohonen, personal communication). We looked at soil moisture because $g_{I\_COS}$ is linearly related to $V_{max}$, which is modulated by a water stress factor depending on the soil moisture (de Rosnay and Polcher, 1998). However, both sites did not experience a strong water stress during the examined years (Table R1.1).

*[10] Table 4: gs is more related to Tair than to VPD. Stomatal conductance has been shown to be related to VPD (See for e.g., Oren et al., 1999). This would be worth examining.*

Answer: Indeed equations (15) and (15a) in Yin and Struik (2009) give the explicit dependency of the stomatal conductance on $VPD$, but $g_{S\_COS}$ also depends on $CO_2$ assimilation (A), which depends primarily on $PAR$, and on air temperature. Table 4 was updated following the clarification that we were discussing canopy-scale conductances, which should then also depend on LAI (Table R2.1).

**Table R2.1: Partial correlations linking stomatal and internal conductances to photosynthetically active radiation (PAR), air temperature (Tair), vapour pressure deficit (VPD), soil moisture (SM) and leaf area index (LAI), computed at a half-hourly time step over year 2012 at the Harvard Forest site and 2017 at the Hyytiälä site**

| Conductance | Site | $PAR$ | $T_{air}$ | $VPD$ | $SM$ | LAI |
|---|---|---|---|---|---|---|
| $g_{S\_COS}$ | Harvard | **0.66** | 0.46 | -0.61 | -0.04 | 0.33 |
| | Hyytiälä | **0.59** | 0.49 | -0.47 | -0.03 | 0.25 |
| $g_{I\_COS}$ | Harvard | -0.06 | **0.68** | 0.30 | -0.27 | 0.15 |
| | Hyytiälä | -0.13 | **0.74** | 0.65 | 0.32 | 0.49 |

For stomatal conductance, this updated Table gives the second role to $VPD$ at Harvard and similar second roles to $T_{air}$ and $VPD$ at Hyytiälä. The Random Forest analysis, which we added given the non-linearity of the problem

stressed by Referee 1, confirms the main role of $PAR$, and ranks $VPD$ as the third most important variable at both sites (Figure R2.5).

[Figure]

**Figure R2.5: Variables importance computed using Random Forests for the stomatal conductance (gs) at the Harvard Forest site in 2012 (left) and at the Hyytiälä site in 2017 (right). The considered predictors are air temperature (Tair), leaf area index (LAI), soil moisture (SM), vapour pressure deficit (VPD) and photosynthetically active radiation (PAR). A random predictor is added to prevent over-fitting.**

*[11] Figure 4. Doesn't seem that Harvard Forest 2013 observed fluxes match simulations all that well. For instance, fluxes seem to peak in May-June and Aug-Sep in 2013, but peak in July in the model. There is a mention of "noise" in EC based measurements in the text but these should be quantified or at least described.*

Answer: We have detailed a bit the description following the Referee's comment: "The simulated weekly seasonal vegetation COS uptake roughly follows the same trend as the observed one ($r$=0.53). COS uptake increases in spring when the vegetation growing season starts and decreases in autumn at the end of the forest activity period. Simulated and observed fluxes also take similar values over the two years. There are however differences: in 2013 the start of the season is simulated about two weeks too late in May instead of late April, and measured fluxes peak in May-June and August-September, while the modelled fluxes peak in July". We have also clarified our meaning regarding the uncertainty of ecosystem flux measurements: "Kohonen et al. (2020) have quantified the relative uncertainty of weekly-averaged ecosystem COS fluxes at 40%, which is coherent with the large standard-deviation computed for field data".

*[12] Figure 5. It would be easier to view this figure as two separate panels instead of one plot with two y axes.*
Answer: We have split the Figure in two separate panels.

[Figure]

[Figure]

**Figure R2.6 Updated Figure 5**

*[14] Minor Comments: Note, I stopped providing minor comments because of the length of the manuscript.*
Answer: Thanks anyway for providing them.

*[15] Line 70: remove "then" in COS is then hydrolyses*
Answer: We have removed it.

*[16] Line 77: mention that soils can be a sink or a source, and cite appropriate studies (cite appropriate studies e.g., Maseyk et al, 2014; Berkelahmmer et al, 2014; Kitz et al., 2020). Also mention the role of epiphytes in the OCS budget (and cite Kuhn et al., 2000; Gimeno et al., 2017 and Rastogi et al, 2018) .*
Answer: Regarding soils and epiphytes, we have expanded the presentation on LRU estimation: "LRU can be estimated experimentally, and then used as a scaling factor for estimating GPP, if $F_{COS}$, $[COS]_a$ and $[CO_2]_a$ are available. Measurements can be made at leaf level using branch chambers (Seibt et al., 2010; Kooijmans et al., 2019); LRU can also be estimated at ecosystem level: eddy-covariance flux towers measure the ecosystem total COS flux (Kohonen et al., 2020), removing the soil contribution gives access to the vegetation part (Wohlfahrt et al., 2012; Wehr et al., 2017). Soil can absorb and emit COS (Whelan et al., 2016; Kitz et al., 2020), the magnitude of their flux being generally much lower than that of vegetation fluxes (Berkelhammer et al., 2014; Maseyk et al., 2014; Wehr et al., 2017; Whelan et al., 2018). Epiphytes (lichen, mosses) could also have a significant contribution to the ecosystem COS budget (Kuhn and Kesselmeier, 2000; Rastogi et al., 2018)". Gimeno et al. (2017) was cited earlier in the introduction, following a comment by the other Referee: "It is however to be noted that Gimeno et al. (2017) reported COS emissions by bryophytes during daytime".

*[17] Lines 78-85: Add some discussion about possible scaling issues for LRU from leaf to canopy and ecoregion scales. I think some of this framework can be found in Wohlfart et al., 2012*
Answer: Regarding the scaling issue in measurements, see in the above comment [16] the modifications that were made in the manuscript. We have also clarified how we integrate fluxes and conductances at canopy and ecoregion scales in ORCHIDEE. In the presentation of ORCHIDEE, we have added: "The canopy is discretized in several layers of growing thickness, the number depending on the actual Leaf Area Index (LAI). The $CO_2$ assimilation, the stomatal conductance and the intercellular $CO_2$ concentration $C_i$ are computed per LAI layer, provided LAI is higher than 0.01 and the mean monthly temperature is higher than -4°C. The $CO_2$ assimilation, the stomatal conductance are further summed-up over all layers to compute GPP and the total conductance at canopy level. The scaling to the grid cell is made using means weighted by the Plant Functional Types fractions". Similarly, in the presentation of the Berry model, we have added: "The vegetation COS flux and related conductances are computed

for each LAI layer, and then summed-up to get total values at canopy level. Unless specified otherwise, fluxes, conductances and LRU are further presented and discussed at canopy level".

*[18] Line 90: summer is likely only valid for ecosystems that exhibit seasonality. Explain diurnal and seasonal variability in LRU.*
Answer: Yes, we have clarified the sentences: "Because of these different responses of COS and $CO_2$ uptake in leaves, LRU varies with light conditions, and decreases sharply with PAR increase (Stimler et al., 2010, 2011; Maseyk et al., 2014; Commane et al., 2015; Wehr et al., 2017; Yang et al., 2018). Consequently, LRU values are smaller at midday or in seasons with high incoming light (Kooijmans et al., 2019)".

*[19] Line 95: What do you mean by 'time'?*
Answer: We refer here to variability of LRU over the day and season inferred by changes in light-conditions. We have changed the sentence ''The variability of LRU with plant type, light, and time should therefore…'' in "The variability of LRU with plant type and over a day and season (inferred by changes in light-conditions) should therefore…''.

*[20] Line 112-13: remove these lines.*
Answer: Albeit this is the classical way to present an article structure in many journals, we removed these lines in our effort to tighten the narration.

*[21] Line 121: Briefly describe improvements in the Farquhar model.*
Answer: We added: "A main novelty is the introduction of a mesophyll conductance linking the $CO_2$ concentration at the carboxylation sites, $C_c$, to the intracellular concentration, $C_i$.". Later is also specified: "The temperature-dependence of the maximum photosynthetic capacity follows Medlyn et al. (2002) and Kattge and Knorr (2007).

*[22] Lines 208-210: Provide scientific names for these species at Harvard Forest.*
Answer: Yes, we have added them: "red oak (*Quercus rubra*), red maple (*Acer rubrum*) and hemlock (*Tsuga canadensis*)".

*[23] Lines 218-220: Awkward phrasing. Also, please elaborate what you mean by "when possible".*
Answer: We were referring to the uncertainties that are later inferred from using different LRU datasets (70% line 472) or different GPP datasets (40% line 464). We removed the part "evidencing some uncertainties when possible" for the sake of simplicity.

*[24] Line 339: change 'air surface temperature' to 'air temperature'.*
Answer: We made the change.

*[25] Line 339: Very minor comment: 'modelling' and 'vapor'. The first is a "British" spelling and the second is American". Please pick one and be consistent throughout.*
Answer: Thanks, the British spelling has been selected, we corrected "vapor" into "vapour".

*[26] Line 348: This isn't true based on Fig 2 as PAR peaks in May but gs in June at Hyytiälä.*
Answer: The Referee is correct. Regarding the $PAR$ seasonal cycle, some additional information was provided in our answer to comment [9]. Following comments by both Referees, this section was largely modified: Figure 2 was removed and a Random Forest analysis was added to the initial partial correlations. The text for $g_{S\_cos}$ and $PAR$ now simply reads: "$PAR$ is the most important variable for the stomatal conductance at the two sites. Due to the way of how $g_{S\_cos}$ is simulated according to Yin and Struik (2009), there is a linear relationship with the $CO_2$ assimilation, which depends mainly on $PAR$."

*[27] Lines 356-358: Needs citations.*
Answer: We added Berry et al. (2013) regarding the modelled internal conductance, and Yin and Struik (2009) for the dependency of $V_{max}$ on air temperature.

*References:*
*Berkelhammer, M., Asaf, D., Still, C., Montzka, S., Noone, D., Gupta, M., Provencal,*
*R., Chen, H. and Yakir, D., 2014. Constraining surface carbon fluxes using in situ*
*measurements of carbonyl sulfide and carbon dioxide. Global Biogeochemical Cycles,*
*28(2), pp.161-179.*

*Buckley, T.N. and Warren, C.R., 2014. The role of mesophyll conductance in the economics*

*of nitrogen and water use in photosynthesis. Photosynthesis research, 119(1-2), pp.77-88.*

*Gimeno, T.E., Campany, C.E., Drake, J.E., Barton, C.V., Tjoelker, M.G., Ubierna, N. and Marshall, J.D., 2020. Wholeˇ ARˇ tree mesophyll conductance reconciles isotopic and gasˇ A ˇ Rexchange estimates of waterˇ ARˇ use efficiency. New Phytologist.*

*Gimeno, T.E., Ogée, J., Royles, J., Gibon, Y.,West, J.B., Burlett, R., Jones, S.P., Sauze, J., Wohl, S., Benard, C. and Genty, B., 2017. Bryophyte gasˇ ARˇ exchange dynamics along varying hydration status reveal a significant carbonyl sulphide (COS) sink in the dark and COS source in the light. New Phytologist, 215(3), pp.965-976.*

*Kitz, F., Spielmann, F.M., Hammerle, A., Kolle, O., Migliavacca, M., Moreno, G., Ibrom, A., Krasnov, D., Noe, S.M. and Wohlfahrt, G., 2020. Soil COS exchange: a comparison of three European ecosystems. Global Biogeochemical Cycles, 34(4), p.e2019GB006202.*

*Maseyk, K., Berry, J.A., Billesbach, D., Campbell, J.E., Torn, M.S., Zahniser, M. and Seibt, U., 2014. Sources and sinks of carbonyl sulfide in an agricultural field in the Southern Great Plains. Proceedings of the National Academy of Sciences, 111(25), pp.9064-9069.*

*Oren, R., Sperry, J.S., Katul, G.G., Pataki, D.E., Ewers, B.E., Phillips, N. and Schäfer, K.V.R., 1999. Survey and synthesis of intraˇ ARˇ and interspecific variation in stomatal sensitivity to vapour pressure deficit. Plant, Cell & Environment, 22(12), pp.1515-1526.*

*Rastogi, B., Berkelhammer, M., Wharton, S., Whelan, M.E., Itter, M.S., Leen, J.B., Gupta, M.X., Noone, D. and Still, C.J., 2018. Large uptake of atmospheric OCS observed at a moist old growth forest: Controls and implications for carbon cycle applications. Journal of Geophysical Research: Biogeosciences, 123(11), pp.3424-3438.*

*Sage, R.F. and Kubien, D.S., 2007. The temperature response of C3 and C4 photosynthesis. Plant, cell & environment, 30(9), pp.1086-1106.*

*Seibt, U., Kesselmeier, J., Sandoval-Soto, L., Kuhn, U. and Berry, J.A., 2010. A kinetic analysis of leaf uptake of COS and its relation to transpiration, photosynthesis and carbon isotope fractionation. Biogeosciences, 7(1).*

*Stangl, Z.R., Tarvainen, L., Wallin, G., Ubierna, N., Räntfors, M. and Marshall, J.D., 2019. Diurnal variation in mesophyll conductance and its influence on modelled wateruse efficiency in a mature boreal Pinus sylvestris stand. Photosynthesis research, 141(1), pp.53-63.*

*von Caemmerer Susanne and Evans, J.R., 2015. Temperature responses of mesophyll conductance differ greatly between species. Plant, Cell & Environment, 38(4), pp.629-637.*

*Wohlfahrt, G., Brilli, F., Hörtnagl, L., Xu, X., Bingemer, H., Hansel, A. and Loreto, F., 2012. Carbonyl sulfide (COS) as a tracer for canopy photosynthesis, transpiration and stomatal conductance: potential and limitations. Plant, cell & environment, 35(4), pp.657-667.*